# PCLR: Progressively Compressed LoRA for Multimodal Continual Instruction Tuning

**Weicheng Meng**[1,2], **Jingyang Qiao**[2,3], **Shaohui Liu**[1,2†], **Zhizhong Zhang**[3,4†], **Yuan Xie**[2,3‡]

[1]Harbin Institute of Technology, Harbin, China
[2]Shanghai Innovation Institute, Shanghai, China
[3]East China Normal University, Shanghai, China
[4]Shanghai Key Laboratory of Computer Software Evaluating and Testing, Shanghai, China
24b903022@stu.hit.edu.cn, 52275901010@stu.ecnu.edu.cn, zzzhang@cs.ecnu.edu.cn,
shliu@hit.edu.cn, yxie@cs.ecnu.edu.cn

## Abstract

Continual Instruction Tuning (CIT) enables Large Multimodal Models (LMMs) to rapidly adapt to new tasks without retraining, but it suffers from the catastrophic forgetting problem. By adding new branches, model extension provides a great idea to accommodate novel knowledge while causing huge memory consumption. To jointly address forgetting and memory explosion, we propose the Compression–Integration–Learning (CIL) pipeline, which draws on the memory consolidation processes during human sleep. Compression streamlines old parameters to release capacity. Integration merges knowledge from similar tasks to restore the performance loss due to compression. For example, based on LLaVA-7B, the forgetting is reduced from 11.29 to 5.09. Learning reallocates released capacity for new task-relevant parameters. Next, based on the characteristics of LMMs at different learning stages, we establish the progressive learning process, further reducing forgetting from 5.09 to 3.39. Moreover, to adapt this process, we decompose LoRA into a set of rank vectors and introduce an extremely fine-grained architecture, LoRA Rank Pool (LRP), with the goal of flexible knowledge employment and editing. Finally, we combine all components, and yield **P**rogressively **C**ompressed **L**o**R**A (PCLR). Extensive experiments demonstrate that PCLR owns a memory budget close to non-extension methods while outperforming extension methods in performance. The implementation code is available at `https://github.com/SII-HITclearlove777/PCLR`.

## 1 Introduction

Large Multimodal Models (LMMs) have gained widespread adoption due to their exceptional cross-modal comprehension and generation capabilities (Lu et al., 2024; Zheng et al., 2023). The training process follows a two-stage paradigm: Pre-Training (PT) and Supervised Fine-Tuning (SFT) (Bai et al., 2025; Chen et al., 2024c). Within SFT, instruction tuning markedly improves the ability of models to follow human intent and become an industry standard practice (Dai et al., 2023; Achiam et al., 2023). However, with continuously evolving data sources and task requirements, the need for frequent retraining is costly and impractical (Scialom et al., 2022; Luo et al., 2023). Thus, Continual Instruction Tuning (CIT) becomes a promising way to learn evolving knowledge for LMMs (Chen et al., 2024a; Xie et al., 2025).

CIT methods include static-structure (non-extension) and model extension (extension) (Yu et al., 2024a). Static methods alleviate forgetting by constraining parameter updates, but they suffer from the trade-off between stability and plasticity (Li & Hoiem, 2017; Aljundi et al., 2018; Zhu et al., 2024). Extension methods append task-specific modules to isolate interference (Wang et al., 2022a; Yu et al., 2024b), but this incurs unbounded memory with tasks growing.

---

[1]†Corresponding Authors
[2]‡Project Leader

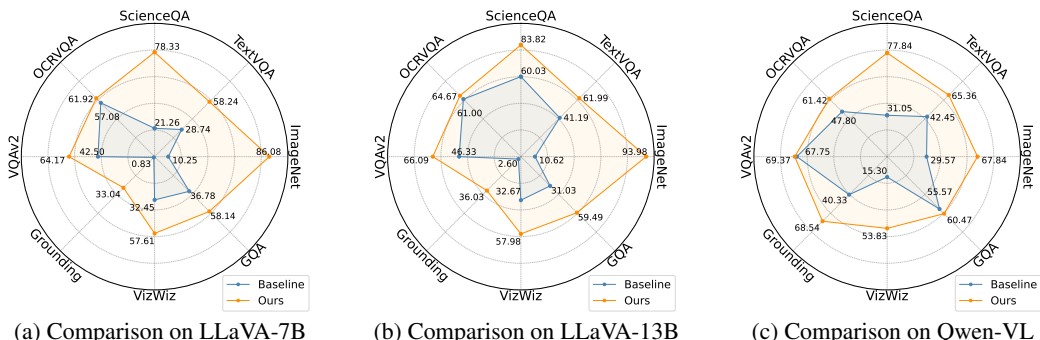

(a) Comparison on LLaVA-7B     (b) Comparison on LLaVA-13B     (c) Comparison on Qwen-VL

Figure 1: Radar chart of comparisons on Final Accuracy between baseline (LoRA) and ours.

To mitigate memory growth, recent methods focus on conditional extension, assuming task interference correlates with feature distribution differences. Before training, feature distribution similarities between previous tasks and the new task are measured. If no similar previous tasks, an independent parameter group is instantiated to avoid interference; otherwise, the task is assigned to the cluster whose feature distribution is closest to the new task and the corresponding parameter group is trained within a regularization constrained update scheme (He et al., 2023; Qiao et al., 2025a). With treating parameter groups as routing experts, these frameworks mitigate high-interference tasks and reduce the need for extension. Nevertheless, they only implicitly postpone structural extension, and memory will still undergo unbounded growth with a long sequence of tasks.

The memory overhead of the model extension is in fact unnecessary. During continual learning, the LoRA parameters from distinct tasks contain linearly dependent rank vectors (as shown in Figure 2), and these rank vectors are compressible. However, current extension methods ignore fine-grained rank-level relationships, which constitute the key driver of memory explosion. To address the problem, we develop a fine-grained Mixture-of-Experts structure, LoRA Rank Pool (LRP), which is inspired by AdaLoRA (Zhang et al., 2023) and L2P series (Wang et al., 2022c;b; Smith et al., 2023b) (Appendix M). LRP decomposes LoRA into rank experts, providing maximal flexibility for knowledge employment and editing. This design reduces storage overhead by minimizing parameter extension during training and pruning parameters post-training.

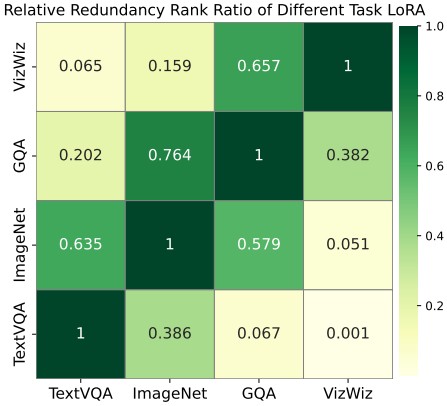

Figure 2: From TextVQA to VizWiz, $\langle \text{VizWiz}, \text{GQA} \rangle$ represents $38.2\%$ ranks in VizWiz's LoRA are redundant versus GQA. Details are shown in Appendix N.

To save memory, we introduce **Compression** after extension: pruning redundant rank experts in LRP to release capacity. Inspired by hippocampal reactivation during sleep that consolidates memory (Wilson & McNaughton, 1994; Stickgold, 2005; Srinivasan et al., 2025), we propose off-learning-phase **Integration**: using knowledge distillation to reactivate learned experts and fuse their internal knowledge, thereby mitigating the performance degradation induced by compression. Combined with **Learning** on new tasks, this yields the **Compression-Integration-Learning (CIL)** pipeline. The memory saved by compression is set equal to that added by learning, aiming to fundamentally eliminate the issue of unbounded extension. Finally, we establish the progressive learning process, effectively mitigating catastrophic forgetting in long-term CIT. Different from previous work, our approach allows all experts to contribute to new task learning, which promotes knowledge transfer and integration. In our method, task-specific experts that support a single task are progressively transformed into mixture-of-experts that support multiple tasks, during continual learning. The evolution is visualized in Appendix F.

We evaluate the proposed method on LLaVA-1.5 (Liu et al., 2024) and Qwen-VL (Bai et al., 2023b) by using the Continual Instruction Tuning benchmark (Chen et al., 2024a). Furthermore, we extend our evaluation to a long-term multimodal CIT benchmark, Continual-NExT. Experimental results demonstrate that PCLR significantly enhances the continual learning capabilities of LMMs. In summary, the contributions of this work are as follows:

- We develop the Compression–Integration–Learning (CIL) pipeline to balance stability, plasticity, and memory efficiency in CIT. Additionally, it can be combined with other regularization strategies to further improve the long-term CIT performance.

- We propose an extremely fine-grained Mixture-of-Experts (MoE) structure, LoRA Rank Pool (LRP), enabling maximal freedom to employ and edit knowledge.

- To the best of our knowledge, PCLR achieves state-of-the-art performance with significant baseline improvements (as shown in Figure 1). In addition, it also owns the superior performance on Continual-NExT (15 tasks), the longest known multimodal CIT benchmark.

## 2 RELATED WORK

**Large Multimodal Models:** Benefiting from the advanced understanding and generation capabilities of LLMs (Bai et al., 2023a; Touvron et al., 2023b; Dubey et al., 2024), LMMs achieve rapid development (Zhan et al., 2024; Wang et al., 2024; Team, 2024; Ge et al., 2024), particularly in Visual Large Language Models (VLLMs) (Chen et al., 2024b; Bai et al., 2025; Li et al., 2024a). VLLMs employ linear projection layers (Touvron et al., 2023a; Bai et al., 2023a) and Q-Former (Liu et al., 2023; Zhu et al., 2023) as cross-modal bridge modules to connect visual encoders to LLM backbones (Touvron et al., 2023a; Bai et al., 2023a), enhancing multimodal reasoning.

**Multimodal Continual Instruction Tuning:** CIT methods can be categorized into three paradigms. Regularization methods constrain gradients or parameters during training (Smith et al., 2023a; Zhu et al., 2024). Structure-based methods enhance performance through architectural modification and dynamic extension (Yan et al., 2021; Jha et al., 2024; Yu et al., 2024b). Replay methods mitigate catastrophic forgetting by replaying high-quality historical samples (Chaudhry et al., 2018; 2019; Yoon et al., 2021; Zhang et al., 2024). To support multimodal CIT research, the CoIN (Chen et al., 2024a) benchmark is introduced. Representative methods: Model Tailor (Zhu et al., 2024) updates critical parameters via sparse masking. Eproj (He et al., 2023) and LCIA (Qiao et al., 2025a) implement grouped parameter learning based on task similarity. Specifically, Eproj introduces similarity-driven dynamic regularization, while LCIA incorporates the dynamic exponential moving average.

## 3 PRELIMINARY

**Problem Definition:** Given the LMM $M$, a series of tasks $T = \{T_1, T_2, \cdots, T_n\}$ and the corresponding instruction datasets $D = \{D_1, D_2, \cdots, D_n\}$, multimodal continual instruction tuning refers to sequentially training $M$ on each new task $T_t$ ($1 \leq t \leq n$) with access only to $D_t$ (or limited access to previous datasets), where $D_t = \{V_i^t, M_i^t, I_i^t\}_{i=1}^{n_t}$, $n_t$ denotes the size of the training set for the $t$-th task, and $V_i^t, M_i^t, I_i^t$ denote the visual inputs, the textual messages, and the instruction for sample $i$, respectively. The objective is to acquire new knowledge while preserving performance on previous tasks during continual learning.

**LoRA Tuning:** LoRA directly interacts with the frozen weights of the original model and has strong performance on complex tasks (Hu et al., 2022). It uses matrix factorisation $\Delta W = \beta A B^T$ to represent weight updates during fine-tuning, where $\beta$ is a scaling factor, $\Delta W \in \mathbb{R}^{d_{in} \times d_{out}}$, $A \in \mathbb{R}^{d_{in} \times r}$, $B \in \mathbb{R}^{d_{out} \times r}$, and $d_{in}, d_{out}, r$ represent the input dimension, output dimension, and latent dimension, respectively. The forward process is defined as: $y = xW + \beta x A B^T$.

**MoE for CIT:** The Mixture-of-Experts (MoE) architecture inherently excels in multi-task learning due to its sparse activation mechanism (Guo et al., 2025). Recent methods leverage data features as queries for sequence-level routing: Eproj (He et al., 2023) encodes visual-textual features to assign the task-specific routing expert, while LCIA (Qiao et al., 2025a) identifies instruction patterns to allocate experts. The general formulation can be shown as:

$$y = xW + \beta \sum_{i=1}^{t} gate(s(x), k)_i x A_i B_i^T, \tag{1}$$

where $s(x) \in \mathbb{R}^t$ denotes the expert routing scores, $t$ is the number of total experts, $k$ is the number of activated experts, and $A_i, B_i$ are the LoRA weights of the $i$-th expert. The operator $gate(v, k)$ selects the top-$k$ entries of vector $v$, setting those positions to 1 and others to 0. Moreover, to

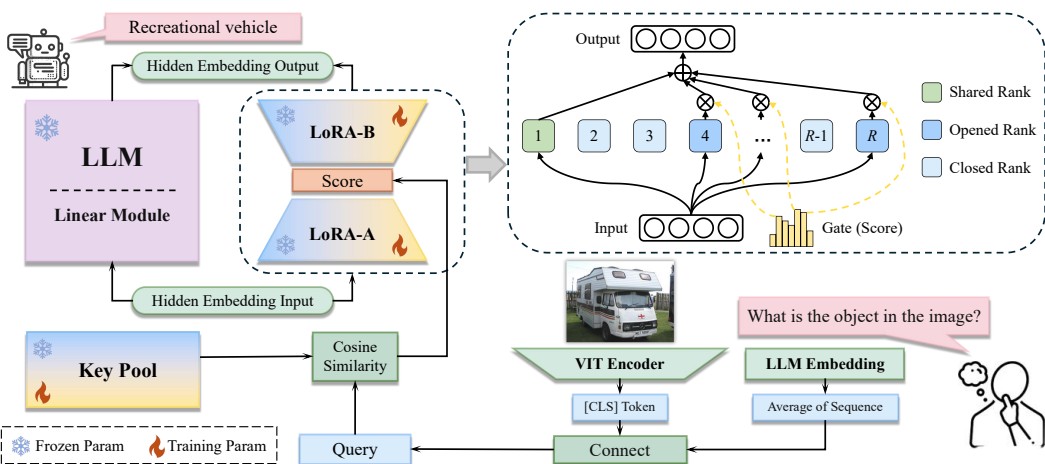

Figure 3: The pipeline of LRP. First, extract the query from inputs. Next, compute similarity scores with the key pools and gate the scores. Then, apply scores jointly with LRP weights in the forward pass. LRP is semi-frozen during training: new parameters are trained while previous ones are frozen.

accommodate continuously growing knowledge, the MoE can be extended by adding new experts, and we refer to this as Dynamic MoE in this paper.

## 4 METHOD

### 4.1 OVERVIEW

Our method comprises two coupled components: the LoRA Rank Pool (LRP) architecture and the Compression–Integration–Learning (CIL) pipeline. LRP factorizes LoRA adapters into rank vectors, each paired with a learnable key, forming an atomic expert that is inserted into linear layers. LRP is similar to Dynamic MoE, which uses static features to activate experts and preserves past knowledge during training by freezing parameters (as shown in Figure 3). For the learning paradigm, we design CIL to emulate the memory cycle of human lifelong learning (as shown in Figure 4). Compression, prunes experts at a preset retention rate to release capacity. Integration, applies an improved distillation algorithm to align the post-compression LRP with its original state, thereby compensating for compression-induced loss by merging similar experts. Learning, trains new experts with the released capacity, and initializes them with similar previous experts to enhance forward transfer. Finally, we build a progressive synergy mechanism combining the CIL pipeline and the CIT process to enhance long-term CIT performance (the algorithm is shown in Appendix U.1).

### 4.2 LORA RANK POOL

Although Dynamic MoE achieves outstanding performance on CIT, its experts tend to acquire overlapping (redundant) knowledge. To address this problem, we decompose experts to the atomic by splitting the LoRA weight $A$ into a column vector set $\{a_1, a_2, \cdots, a_n\}$, and the weight $B^T$ into a row vector set $\{b_1^T, b_2^T, \cdots, b_n^T\}$. Then, we introduce globally shared components $A_s, B_s$ to accumulate global knowledge. Thus, we obtain the LoRA Rank Pool formulation from Eq.(1):

$$y = xW + \beta_s x A_s B_s^T + \beta_m \sum_{i=1}^{n} gate(s(x), r)_i x a_i b_i^T, \tag{2}$$

where $n$ is the number of total rank experts, $r$ is the number of activated rank experts, $\beta_s$ is the factor of the shared ranks part, $\beta_m$ is the factor of the mixture-of-experts part.

Next, we define the score function $s(x) = Kq$, where $K \in \mathbb{R}^{n \times d}, q \in \mathbb{R}^d$ are the L2-regularized key pool and query. For each input $x$, we construct $q$ (query) by concatenating the mean-pooling of text embeddings with the visual output ([CLS] token or mean-pooling). Each rank expert in the LRP is assigned a learnable key, and all keys form $K$ (key pool). We use cosine similarity scores and gating to select top-$r$ relevant experts. This enables the model to employ distinct experts to acquire different knowledge and selectively activate them during inference.

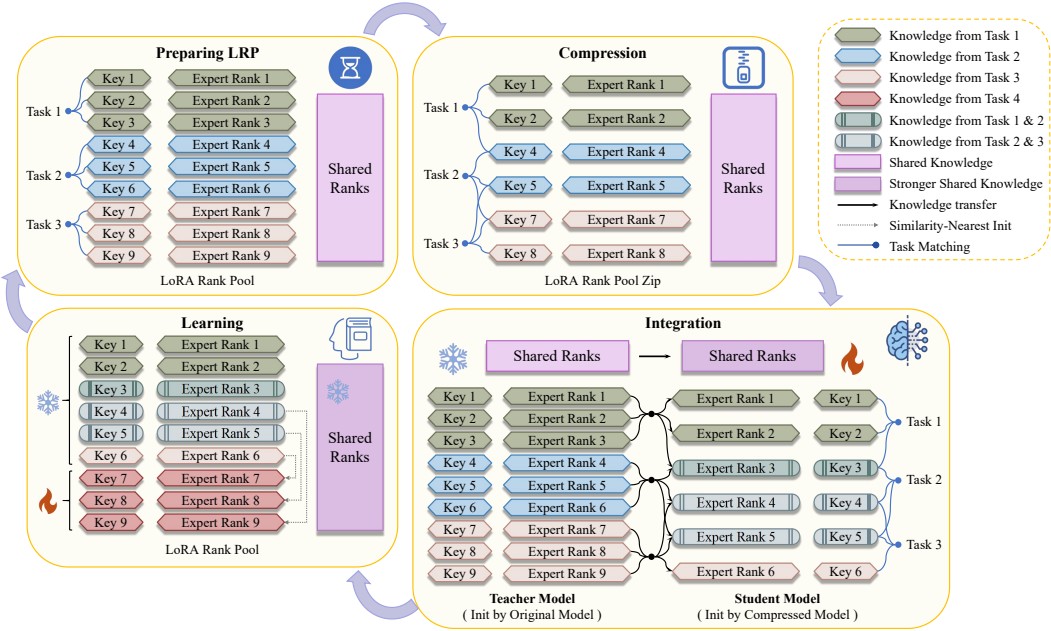

Figure 4: Compression-Integration-Learning (CIL) pipeline. Compression is a training-free process that streamlines the original LRP. Integration is a distillation process that aligns the compressed LRP with the original LRP. Learning is a supervised fine-tuning process that initializes the LRP from the integration-processed LRP and then adapts to the new task.

In Eq.(2), the mixture-of-experts part forms an $n$-term matrix polynomial. Sequential execution of all terms would cause significant computational overhead, making parallelization necessary. We aggregate the discrete ranks into $A_m$ and $B_m$. We broadcast the scoring vector $s(x)$ to match the dimensions of $xA_m$ and derive a parallelized formulation:

$$y = xW + \beta_s xA_s B_s^T + \beta_m F\left(xA_m, gate\left(Kq, r\right)^T\right) B_m^T,$$
(3)

where the operator $F\left(U, v\right)$ broadcasts $v \in \mathbb{R}^b$ to match the shape of $U \in \mathbb{R}^{a \times b}$, resulting in two matrices with identical dimensions, followed by an element-wise multiplication.

To jointly improve downstream task performance and learn the keys of labeled experts within the LRP, we combine the cross-entropy loss with the query–key cosine similarity loss:

$$\mathcal{L}\left(\theta\right) = -\frac{1}{T}\sum_{t=1}^{T}\log P_\theta\left(y_t \mid x, y_{<t}, q\right) + \frac{\lambda}{l}\sum_{i=1}^{l}\left\|\mathbf{1}_r - top_r\left(K_i\right)q\right\|_1,$$
(4)

where $\lambda$ is a balancing hyperparameter, and $l$ denotes the number of key pools, which equals the LMMs layer count. To preserve previous knowledge, we freeze all previous keys and ranks while optimizing only the newly added ones.

### 4.3 COMPRESSION-INTEGRATION-LEARNING

Conventional learning faces the dilemma of whether to learn within the previous parameter space or to allocate a new one. The former suffers from the trade-off between stability and plasticity, whereas the latter leads to unbounded memory growth. To address this problem, we propose the Compression-Integration-Learning (CIL) pipeline, which compresses previous knowledge when learning new tasks. We decompose each CIT task into three main phases: compressing, integrating past knowledge, and learning new knowledge (as shown in Figure 4). Notice that CIL is initiated once the total rank reaches a predefined value.

**Compression:** At this phase, we introduce a compression retention rate $\alpha \in (0, 1]$, discarding a portion of redundant rank experts and their corresponding keys. As a result, the pruned LRP maintains its rank and key count at $\alpha$ times the values of the original one. Although compression may lead to performance loss, it is crucial for alleviating the memory pressure of the learning system.

**Integration:** Removing key-value pairs will degrade knowledge from previously learned tasks. To compensate for this loss, we perform integration on the current dataset to adapt the LRP using the distillation loss (Hinton et al., 2015; Huang & Wang, 2017; Gou et al., 2021). Specifically, we formulate the Query-based Kullback-Leibler Divergence Loss (QKLD Loss) to align the compressed LRP with the original one. The process freezes the teacher model (LRP before compression) and updates the student model (LRP after compression). Through distillation, rank experts in LRP absorb the knowledge from pruned experts, and the common representations across different tasks flow into the shared knowledge space, ultimately mitigating the knowledge loss caused by compression. The pseudo algorithm of compression and integration can be referred to in Appendix U.3.

LRP activates specific rank experts based on the query of the input. However, the datasets from previous tasks are not accessible during integration. Considering that we optimize the query–key cosine similarity during learning, and its optimum is achieved when each new key equals the mean of all queries for the corresponding task. Therefore, the learned keys can serve as surrogates for the queries from previous tasks, and we define them as fake queries. During integration, we compute Kullback-Leibler Divergence (KLD) to quantify the performance loss associated with the fake query:

$$D_{KL}(q) = \frac{1}{T}\sum_{t=1}^{T} P_{\theta_{ori}}(x_t \mid x_{<t}, q) \log\left(\frac{P_{\theta_{ori}}(x_t \mid x_{<t}, q)}{P_{\theta_{zip}}(x_t \mid x_{<t}, q)}\right), \tag{5}$$

where $\theta_{ori}$ is the original LRP, $\theta_{zip}$ is the compressed LRP, $x$ is the input. To emphasize optimization for tasks with larger performance degradation, we define the sampling probability of fake query:

$$P(q) = \frac{\sqrt{D_{KL}(q)}}{\sum_{q \in K_{ori}} \sqrt{D_{KL}(q)}}, \tag{6}$$

where $K_{ori}$ is the original key pool. Finally, we define the QKLD Loss:

$$\mathcal{L}(\theta_{zip}) = \mathbb{E}_{q \sim P}[D_{KL}(q)] = \sum_{q \in K} P(q) D_{KL}(q). \tag{7}$$

**Learning:** During the learning phase, shared and previous ranks are frozen to preserve past knowledge, while new key–value pairs (rank experts and their corresponding keys) are trained using the capacity released through compression to optimize the new task. By decomposing LoRA into rank vectors, we enhance forward transfer: new parameters are initialized using ranks from similar tasks. This is achieved by computing the queries of a subset of training samples before learning, taking their mean as an identifier to match to key-value pairs in the LRP, and initializing the new rank components with the values corresponding to the keys that exhibit the top-$r$ cosine similarity scores. The pseudo algorithm of learning can be referred to in Appendix U.2.

### 4.4 PROGRESSIVE LEARNING PROCESS

In this section, we introduce the progressive learning process to further improve the continual learning performance of LMMs. In the early CIT, the rank space has not attained its allotted capacity, rendering compression an unnecessary expense. Thus, we disable compression until the capacity is filled. This optimization reduces training cost and avoids unnecessary performance degradation without altering the final number of rank experts and the model memory.

As the number of tasks increases, LMMs accumulate more knowledge and strengthen their capabilities. This means that when learning new tasks, the number of additional trainable parameters progressively decreases, because much of the relevant knowledge has already been encountered. Consequently, the learning focus shifts from acquiring entirely new knowledge to organizing and consolidating acquired knowledge. In the CIL pipeline, overlap between new tasks and earlier knowledge (reusable knowledge) increases. Meanwhile, internal representation becomes increasingly compact and resistant to further compression (high knowledge density).

Accordingly, in the later CIT, we allocate fewer new ranks and employ a lighter, high-retention compression scheme to minimize performance degradation. This encourages LMMs to rely more on frozen knowledge during the late learning phase and has negligible impact on new task performance. The progressive learning process, through deliberate planning of the learning path for LMMs, further enhances stability, plasticity, memory efficiency, and temporal efficiency of its evolution.

Table 1: Comparisons between ours and baselines on LLaVA-1.5-7B, CoIN benchmark.

| Method | Venue | Accuracy on Each Task | | | | | | | | Overall Results | | |
|---|---|---|---|---|---|---|---|---|---|---|---|---|
| | | 1-ScienceQA | 2-TextVQA | 3-ImageNet | 4-GQA | 5-VizWiz | 6-Grounding | 7-VQAv2 | 8-OCRVQA | Avg.ACC(↑) | Forgetting(↓) | New.ACC(↑) |
| Zero-shot | - | 49.91 | 2.88 | 0.33 | 2.08 | 0.90 | 0.00 | 0.68 | 0.17 | 7.12 | - | - |
| Multi-Task | - | 56.77 | 49.35 | 95.55 | 56.65 | 53.90 | 30.09 | 59.50 | 55.65 | 57.18 | - | - |
| LoRA (Hu et al., 2022) | ICLR'22 | 21.26 | 28.74 | 10.25 | 36.78 | 32.45 | 0.83 | 42.50 | 57.08 | 28.74 | 37.29 | 61.36 |
| LwF (Li & Hoiem, 2017) | TPAMI'16 | 63.14 | 39.60 | 8.90 | 34.83 | 14.53 | 2.48 | 40.67 | 62.35 | 33.31 | 22.32 | 52.58 |
| EWC (Kirkpatrick et al., 2017) | PNAS'17 | 67.41 | 40.41 | 8.18 | 35.05 | 37.88 | 2.67 | 41.27 | 61.02 | 36.74 | 20.51 | 54.68 |
| MoELoRA (Chen et al., 2024a) | NIPS'24 | 58.92 | 38.59 | 8.85 | 37.10 | 44.25 | 2.45 | 41.40 | 55.35 | 35.86 | 25.71 | 58.36 |
| AdaLoRA (Zhang et al., 2023) | ICLR'23 | 73.40 | 51.29 | 35.47 | 44.53 | 46.75 | 0.93 | 55.86 | 62.03 | 46.28 | 23.99 | 63.27 |
| MT (Zhu et al., 2024) | ICML'24 | 79.63 | 55.47 | 35.64 | 58.70 | 44.37 | 32.20 | 62.21 | 61.59 | 53.73 | 14.03 | 66.00 |
| PGP (Qiao et al., 2025b) | ICLR'24 | 85.17 | 56.85 | 32.26 | 61.74 | 49.43 | 32.74 | 65.74 | 62.20 | 55.77 | 12.94 | 67.09 |
| CIA* (Qiao et al., 2025a) | ICML'25 | 75.63 | 54.47 | 43.64 | 60.70 | 43.37 | 36.00 | 65.21 | 63.59 | 55.33 | 7.04 | 61.49 |
| SEFE (Chen et al., 2025) | ICML'25 | 75.35 | 58.66 | 83.10 | 54.25 | 48.85 | 16.75 | 65.35 | 66.25 | 58.57 | 11.94 | 69.02 |
| ProgLoRA (Yu et al., 2025) | ACL'25 | 74.84 | 51.83 | 83.90 | 49.93 | 53.87 | 31.19 | 62.71 | 64.44 | 59.09 | 7.53 | 65.68 |
| EProj (He et al., 2023) | ArXiv'23 | 78.51 | 57.53 | 92.35 | 55.93 | 44.67 | 36.59 | 63.74 | 57.00 | 60.79 | 5.42 | 65.54 |
| **PCLR** | - | **78.33** | **58.24** | **86.08** | **58.14** | **57.61** | **33.04** | **64.17** | **61.92** | **62.19** | **3.39** | **65.16** |

## 4.5 Integration with Regularization Methods

To further enhance the performance of our proposed PCLR on long-term CIT while minimizing unnecessary performance degradation and training overhead, we integrate it with regularization methods. Specifically, the long sequence of tasks is partitioned into groups of contiguous tasks. Within each group we perform continual learning in a unified rank space combined with regularization methods, while across groups we retain the CIL process, thereby avoiding unnecessary extension and compression. This optimization enhances the long-term CIT performance of PCLR while markedly reducing training overhead. We present cases of the integration of PCLR with LwF (Li & Hoiem, 2017) in the experiments, and the pseudo algorithm can be referred to in Appendix U.4.

## 5 Experiments

### 5.1 Experimental Setup

**CoIN Benchmark:** To verify the performance of our method in challenging scenarios, we adopt LLaVA-1.5 (Liu et al., 2023)/Qwen-VL (Bai et al., 2023b) as the base model, and insert the PCLR module into the linear layers of the LLM backbone network and the cross-modal bridge. We use the CoIN (Chen et al., 2024a) benchmark and keep the order of ScienceQA (Lu et al., 2022), TextVQA (Singh et al., 2019), ImageNet (Russakovsky et al., 2015), GQA (Hudson & Manning, 2019), VizWiz (Gurari et al., 2018), Grounding (Mao et al., 2016), VQAv2 (Goyal et al., 2017), and OCRVQA (Mishra et al., 2019). These visual language datasets encompass a diverse range of task types, including selection, classification, grounding, and open-ended question answering.

**Continual-NExT Benchmark:** In order to further verify our method in the long-term CIT setting, we use the Continual-NExT (Xie et al., 2025) benchmark, and keep the order of ArXivQA (Li et al., 2024b), GeoChat (Kuckreja et al., 2024), IconQA (Lu et al., 2021), ClevrMath (Lindström & Abraham, 2022), CodeQA (Liu & Wan, 2021), ImageNet (Russakovsky et al., 2015), Flickr30k (Plummer et al., 2015), DocVQA (Mathew et al., 2021), TextVQA (Singh et al., 2019), MathQA (Amini et al., 2019), ChartQA (Masry et al., 2022), PathVQA (He et al., 2020), Grounding (Mao et al., 2016), ScienceQA (Lu et al., 2022), and WikiQA (Yang et al., 2015). These visual language and pure language datasets cover multiple fields such as coding, mathematics, remote sensing and medical images. The experimental setup is shown in Appendix B.

**Comparison Methods:** Compared baselines include LoRA (Hu et al., 2022), zero-shot and multitask; regularization methods: (1) LWF (Li & Hoiem, 2017), (2) EWC (Kirkpatrick et al., 2017), (3) GEM (Lopez-Paz & Ranzato, 2017), (4) MT (Zhu et al., 2024), (5) PGP (Qiao et al., 2025b), (6) CIA* (w/o Instruction Grouping) (Qiao et al., 2025a), (7) SEFE (Chen et al., 2025); static-structure method: (1) MoELoRA (Chen et al., 2024a), (2) AdaLoRA (Zhang et al., 2023); extension methods: (1) Eproj (He et al., 2023), (2) CIA (Qiao et al., 2025a), (3) ProgLoRA (Yu et al., 2025); replay method: (1) Experience Replay (Rolnick et al., 2019). The experiments all adopt the setting of training each task for 1 epoch. The method details are provided in Appendix S.

**Evaluation Metrics:** We utilize three popular evaluation metrics: the average accuracy (Avg.ACC), the forgetting (Forgetting) and the new accuracy (New.ACC), which are shown in Appendix E.

### 5.2 Main Results

As shown in Table 1, we discover that PCLR achieves the highest overall performance on the CoIN benchmark. Compared to the former best regularization method SEFE, the Avg.ACC improves by

Table 2: Comparisons between ours and baselines on LLaVA-1.5-hf, Continual-NExT benchmark.

| Method | Venue | Accuracy on Each Task & Overall Results | | | | | | | | |
|---|---|---|---|---|---|---|---|---|---|---|
| | | 1-ArxivQA | 2-GeoChat | 3-IconQA | 4-ClevrMath | 5-CodeQA | 6-ImageNet | 7-Flickr30k | 8-DocVQA | 9-TextVQA |
| Zero-shot | - | 36.99 | 67.67 | 18.77 | 20.27 | 0.26 | 18.10 | 17.27 | 14.58 | 57.39 |
| Multi-Task | - | 64.08 | 96.40 | 60.60 | 70.50 | 10.81 | 97.06 | 20.52 | 23.18 | 65.44 |
| LoRA (Hu et al., 2022) | ICLR'22 | 53.99 | 92.23 | 47.23 | 44.86 | 4.36 | 67.84 | 17.16 | 16.47 | 47.70 |
| LwF (Li & Hoiem, 2017) | TPAMI'16 | 51.04 | 87.33 | 30.97 | 39.20 | 4.74 | 84.89 | 16.26 | 16.56 | 54.09 |
| EWC (Kirkpatrick et al., 2017) | PNAS'17 | 55.16 | 91.73 | 47.17 | 49.30 | 4.38 | 82.03 | 16.71 | 16.88 | 51.73 |
| GEM (Lopez-Paz & Ranzato, 2017) | NIPS'17 | 55.30 | 91.03 | 49.13 | 48.30 | 4.76 | 76.20 | 16.21 | 15.85 | 51.33 |
| Replay-100 (Rolnick et al., 2019) | NIPS'19 | 54.85 | 94.40 | 51.73 | 40.07 | 4.48 | 94.61 | 9.36 | 14.65 | 54.70 |
| MoELoRA (Chen et al., 2024a) | NIPS'24 | 56.00 | 91.36 | 48.76 | 48.90 | 3.82 | 82.19 | 17.77 | 16.33 | 59.51 |
| CIA* (Qiao et al., 2025a) | ICML'25 | 56.35 | 93.41 | 48.76 | 48.20 | 4.23 | 83.00 | 16.54 | 16.98 | 51.32 |
| **PCLR** | - | **59.00** | **78.20** | **51.30** | **37.17** | **7.86** | **85.19** | **18.68** | **20.00** | **64.10** |
| **PCLR-LwF (PCLR variant)** | - | **61.16** | **96.37** | **62.40** | **64.70** | **8.68** | **97.46** | **20.63** | **19.84** | **63.97** |
| | | 10-MathQA | 11-ChartQA | 12-PathVQA | 13-Grounding | 14-ScienceQA | 15-WikiQA | Avg.ACC(↑) | Forgetting(↓) | New.ACC(↑) |
| Zero-shot | - | 0.44 | 9.60 | 33.29 | 28.28 | 66.19 | 17.54 | 27.11 | - | - |
| Multi-Task | - | 36.01 | 20.76 | 58.61 | 72.03 | 86.21 | 23.38 | 53.71 | - | - |
| LoRA (Hu et al., 2022) | ICLR'22 | 33.80 | 18.04 | 50.98 | 69.52 | 89.46 | 22.27 | 45.06 | 11.62 | 55.91 |
| LwF (Li & Hoiem, 2017) | TPAMI'16 | 30.05 | 18.64 | 52.79 | 64.11 | 87.95 | 24.96 | 44.24 | 12.29 | 55.70 |
| EWC (Kirkpatrick et al., 2017) | PNAS'17 | 35.41 | 19.00 | 50.92 | 69.92 | 89.51 | 24.17 | 46.93 | 9.72 | 56.01 |
| GEM (Lopez-Paz & Ranzato, 2017) | NIPS'17 | 35.28 | 17.68 | 51.38 | 67.23 | 89.86 | 23.85 | 46.23 | 10.19 | 55.74 |
| Replay-100 (Rolnick et al., 2019) | NIPS'19 | 31.42 | 14.40 | 49.64 | 56.98 | 85.62 | 23.85 | 45.38 | 11.41 | 56.03 |
| MoELoRA (Chen et al., 2024a) | NIPS'24 | 34.17 | 18.52 | 49.04 | 67.65 | 88.28 | 22.59 | 46.99 | 8.06 | 54.51 |
| CIA* (Qiao et al., 2025a) | ICML'25 | 32.56 | 17.62 | 50.47 | 69.86 | 89.37 | 23.54 | 46.81 | 8.30 | 54.55 |
| **PCLR** | - | **36.42** | **20.96** | **58.48** | **69.76** | **89.65** | **22.27** | **47.94** | **7.71** | **55.14** |
| **PCLR-LwF (PCLR variant)** | - | **37.22** | **19.84** | **58.42** | **62.83** | **83.09** | **33.49** | **52.67** | **4.58** | **56.89** |

3.62 and the Forgetting decreases by 8.55, demonstrating strong continual learning ability. Additionally, some early approaches (e.g., LwF and EWC) tend to overemphasize the mitigation of forgetting, while they suffer from severe plasticity reduction. Recent methods (such as MT and EProj) place greater importance on maintaining plasticity, contributing to significant improvements in New.ACC. However, these methods still struggle to achieve an optimal balance between New.ACC and Forgetting. Notably, our proposed method, PCLR, achieves the highest New.ACC and the lowest Forgetting among all compared approaches, demonstrating its superior performance. Furthermore, PCLR surpasses both the previous best extension method (EProj) and the leading dynamic update method (CIA*). To validate PCLR's stability and scalability on larger models and diverse architectures, we conduct evaluations on LLaVA-1.5-13B and Qwen-VL. Detailed results are provided in Appendix A. Additionally, we present several visualizations in Appendix C.

As shown in Table 2, our method exceeds the best of other methods (MoELoRA), improving Avg.ACC by 0.95 and reducing Forgetting by 0.35, demonstrating that PCLR sustains strong performance on the long-term Continual-NExT benchmark. To further enhance PCLR's efficiency and performance, we merge it with LwF to form PCLR-LwF. Specifically, we group adjacent tasks and perform LwF fine-tuning within each group (details are shown in Appendix B and Appendix K). This integration with a simple baseline yields notable gains over the original PCLR, improving Avg.ACC by 4.73 and reducing Forgetting by 3.13. Additionally, the results of Qwen2.5-VL-Instruct (Bai et al., 2025) are provided in Appendix A. The visualization is shown in Appendix G.

## 5.3 ROBUST EXPERIMENTS

To verify that PCLR exhibits strong robustness under different CIT settings, we conduct two sets of experiments on instruction templates and learning orders. Note that, in double-row tables, the upper row denotes the immediate accuracy (evaluate after the current task), and the lower row denotes the final accuracy (evaluate after the final task).

Table 3: Results of LLaVA-1.5-7B on **different instruction templates**.

| Type | Accuracy on Each Task | | | | | | | | Overall Results | | |
|---|---|---|---|---|---|---|---|---|---|---|---|
| | 1-ScienceQA | 2-TextVQA | 3-ImageNet | 4-GQA | 5-VizWiz | 6-Grounding | 7-VQAV2 | 8-OCRVQA | Avg.ACC | Forgetting | New.ACC |
| Origin | 83.47 | 61.29 | 96.50 | 59.97 | 58.32 | 34.02 | 65.75 | 61.92 | 62.19 | 3.39 | 65.16 |
| | 78.33 | 58.24 | 86.08 | 58.14 | 57.61 | 33.04 | 64.17 | 61.92 | | | |
| Diverse | 83.47 | 61.36 | 96.57 | 60.03 | 58.53 | 34.23 | 65.83 | 61.96 | 62.00 | 3.71 | 65.25 |
| | 76.92 | 56.46 | 86.36 | 57.93 | 57.54 | 33.06 | 65.75 | 61.96 | | | |
| 10Type | 84.58 | 60.89 | 96.40 | 59.59 | 58.21 | 34.14 | 65.67 | 60.99 | 61.98 | 3.52 | 65.06 |
| | 80.24 | 58.62 | 84.81 | 57.85 | 56.59 | 32.29 | 64.44 | 60.99 | | | |

**Different Instruction Templates:** As part of the text input, different instructions can influence query encoding and current task learning. We select three instruction templates (details are shown in Appendix I) for comparison, as shown in Table 3. We observe that changes in the instructions incur only a negligible effect, and our method can be adapted to different instruction templates.

**Different Learning Order:** In CIT, interfering tasks suffer from severe catastrophic forgetting when learned in adjacent order without parameter grouping. PCLR smoothly transitions task-specific experts to mixture-of-experts through the CIL pipeline, which can resist this interference phenomenon

Table 4: Results of LLaVA-1.5-7B on **different task orders**.

| Order | Accuracy on Each Task | | | | | | | | Overall Results | | |
|---|---|---|---|---|---|---|---|---|---|---|---|
| | 1-ScienceQA | 2-TextVQA | 3-ImageNet | 4-GQA | 5-VizWiz | 6-Grounding | 7-VQAV2 | 8-OCRVQA | Avg.ACC | Forgetting | New.ACC |
| Origin | 83.47 | 61.29 | 96.50 | 59.97 | 58.32 | 34.02 | 65.75 | 61.92 | 62.19 | 3.39 | 65.16 |
| | 78.33 | 58.24 | 86.08 | 58.14 | 57.61 | 33.04 | 64.17 | 61.92 | | | |
| | 1-OCRVQA | 2-VQAV2 | 3-Grounding | 4-VizWiz | 5-GQA | 6-ImageNet | 7-TextVQA | 8-ScienceQA | Avg.ACC | Forgetting | New.ACC |
| Reverse | 58.97 | 65.06 | 34.40 | 58.42 | 61.08 | 96.61 | 60.40 | 83.68 | 62.18 | 3.03 | 64.83 |
| | 56.78 | 62.66 | 26.68 | 55.85 | 57.00 | 94.02 | 60.74 | 83.68 | | | |
| | 1-GQA | 2-Grounding | 3-ImageNet | 4-OCRVQA | 5-ScienceQA | 6-TextVQA | 7-VizWiz | 8-VQAV2 | Avg.ACC | Forgetting | New.ACC |
| Alphabet | 60.14 | 33.10 | 96.69 | 61.49 | 83.97 | 60.52 | 57.10 | 64.34 | 60.62 | 4.63 | 64.67 |
| | 59.70 | 27.32 | 83.88 | 50.15 | 82.72 | 59.94 | 56.89 | 64.34 | | | |

to a certain extent. In order to verify that PCLR has strong adaptability to different CIT orders, we compare three settings with different task orders. As shown in Table 4, task order changes induce slight knowledge conflicts, and have minor impact on the overall continual learning performance.

## 5.4 ABLATION STUDY

To evaluate each PCLR component, we start from the LoRA baseline, add components incrementally, and compare continual instruction tuning performance. Results are shown in Table 5. The experimental results demonstrate that each proposed component is effective in enhancing accuracy and reducing forgetting of LMMs. To validate the superiority of the LRP architecture, we compare the LoRA baseline with the LRP (which only performs compression and learning).

We can observe that the introduction of the LRP architecture increases Avg.ACC (+25.92) and reduces Forgetting (-26). In addition, we retain LRP and compare the Compression-Integration-Learning (CIL) and Compression-Learning (CL) pro-

Table 5: Ablation study results.

| Method | Avg.ACC(↑) | Forgetting(↓) | New.ACC(↑) |
|---|---|---|---|
| LoRA(Baseline) | 28.74 | 37.29 | 61.36 |
| w/o Integration Process | 54.66 | 11.29 | 64.54 |
| w/o Progressive Process | 60.78 | 5.09 | 65.23 |
| PCLR(Ours) | **62.19** | **3.39** | **65.16** |

cesses to validate the effectiveness of the integration process, and the results show that integration increases Avg.ACC (+6.12) and reduces Forgetting (-6.2). Moreover, we observe that the introduction of the progressive process further increases Avg.ACC (+1.41) and reduces Forgetting (-1.7). In summary, compared with the baseline, PCLR increases Avg.ACC (+33.45) and New.ACC (+3.8), reduces Forgetting (-33.9), and achieves an optimal balance between plasticity and stability.

## 5.5 THE IMPACT OF PROGRESSIVE LEARNING

To investigate the impact of different compression strategies on CIT, we designed five methods: Aggressive (non-progressive), Conservative (non-progressive), Reverse Progressive, Centralized Compression, and Progressive Compression (Ours). These methods differ only in their compression strategies while maintaining the same memory usage (details are shown in Appendix J).

Table 6: Results of LLaVA-1.5-7B on **different compression strategies**.

| Strategy | Final Accuracy on Each Task | | | | | | | | Overall Results | | |
|---|---|---|---|---|---|---|---|---|---|---|---|
| | 1-ScienceQA | 2-TextVQA | 3-ImageNet | 4-GQA | 5-VizWiz | 6-Grounding | 7-VQAV2 | 8-OCRVQA | Avg.ACC | Forgetting | New.ACC |
| Aggressive | 76.02 | 57.03 | 82.61 | 55.84 | 57.24 | 30.71 | 63.86 | 62.92 | 60.78 | 5.09 | 65.23 |
| Conservative | 81.37 | 59.23 | 92.28 | 58.77 | 54.55 | 25.88 | 64.16 | 59.03 | 61.91 | 1.77 | 63.43 |
| Reverse | 79.13 | 58.79 | 87.41 | 58.51 | 53.39 | 25.67 | 64.44 | 62.76 | 61.26 | 3.21 | 64.07 |
| Centralized | 77.72 | 53.78 | 88.93 | 60.05 | 55.71 | 28.71 | 62.93 | 60.16 | 61.00 | 3.38 | 64.37 |
| **Ours** | **78.33** | **58.24** | **86.08** | **58.14** | **57.61** | **33.04** | **64.17** | **61.92** | **62.19** | **3.39** | **65.16** |

As shown in Table 6, Aggressive Compression improves performance on new tasks but lacks the ability to consolidate old knowledge (stability), with Forgetting increased (+1.7). Conservative Compression reduces forgetting but limits the ability to learn new tasks (plasticity), with New.ACC decreased (-1.73). Reverse Progressive Compression exhibits subpar performance in both stability and plasticity, with Avg.ACC decreased (-0.93). Centralized Compression also demonstrates overall lower performance, with Avg.ACC decreased (-1.19). In contrast, Progressive Compression dynamically adjusts the compression retention rate, outperforming other strategies in reducing forgetting (Forgetting = 3.39) and enhancing new task learning ability (New.ACC = 65.16), achieving the best overall performance (Avg.ACC = 62.19). By optimizing the learning trajectory based on changes in task knowledge density, Progressive Compression strikes a balance between stability and plasticity, providing an effective design solution for our system development.

Table 7: The results of LLaVA-1.5-13B on **different integration data usage**. For the double-row table, the upper row denotes the final accuracy, and the lower row denotes the integration cost.

| IDU | Accuracy and Integration Cost on Each Task | | | | | | | | Evaluation Metrics | | |
|---|---|---|---|---|---|---|---|---|---|---|---|
| | 1-ScienceQA | 2-TextVQA | 3-ImageNet | 4-GQA | 5-VizWiz | 6-Grounding | 7-VQAV2 | 8-OCRVQA | Avg.ACC | Forgetting | Avg.ICost |
| 0 | 81.63 | 60.84 | 56.55 | 55.83 | 52.95 | 34.51 | 67.64 | 65.27 | 59.40 | 8.86 | - |
| | - | - | - | - | - | - | - | - | | | |
| 5k | 83.49 | 61.33 | 91.09 | 58.11 | 54.41 | 35.92 | 67.47 | 64.17 | 64.50 | 3.11 | 11 min |
| | - | - | - | 14 min | 7 min | 12 min | 10 min | - | | | |
| 10k | 83.33 | 61.35 | 93.15 | 58.39 | 55.94 | 35.76 | 67.29 | 64.12 | 64.92 | 2.61 | 20 min |
| | - | - | - | 26 min | 13 min | 23 min | 18 min | - | | | |
| 20k | 83.57 | 61.47 | 94.34 | 58.90 | 56.82 | 34.31 | 66.88 | 64.82 | 65.14 | 2.27 | 38 min |
| | - | - | - | 49 min | 24 min | 45 min | 35 min | - | | | |
| Origin | 83.82 | 61.99 | 93.98 | 59.49 | 57.98 | 36.03 | 66.09 | 64.67 | 65.51 | 2.08 | 115 min |
| | - | - | - | 173 min | 24 min | 121 min | 141 min | - | | | |

## 5.6 EFFICIENCY ANALYSIS

All LRP computations are fully parallelizable with the training cost close to the MoELoRA with one expert (Appendix L). For PCLR, the primary additional cost arises from the integration. We control the integration strength by varying the amount of integration data usage. Specifically, Integration Data Usage (IDU) is set to 5 levels: 0, 5k, 10k, 20k samples, and Origin. Here, **0** denotes no integration, and **Origin** refers to the full dataset for the current task. Experiments are conducted on LLaVA-1.5-13B. Results are reported in Table 7 (Avg.ICost denotes the average integration-phase cost in minutes). In this paper, we use the **Origin** level to obtain the best performance.

In conclusion, PCLR does not incur significant time cost, and appropriately reducing the IDU will not cause serious performance degradation. When computing resources are sufficient, the IDU can be increased to pursue the best final performance.

## 6 CONCLUSION

In this work, we introduce PCLR, which provides an extremely fine-grained LoRA Rank Pool (LRP) with a Compression–Integration–Learning (CIL) pipeline to balance stability, plasticity, and memory efficiency during CIT. LRP provides maximal flexibility for expert employment and editing. CIL progressively compresses LRP, trading off a minor performance drop to eliminate unlimited model extension. On CoIN and Continual-NExT benchmarks across multiple LMMs, it delivers superior overall performance to former regularization methods while remaining competitive with extension methods. Robustness studies demonstrate the stability of the method. Ablation experiments confirm the necessity of each component. Efficiency analyses guide our selection of the optimal amount of integration data usage, balancing performance and speed. Owing to computing resource limits, we currently focus on image–text CIT. In the future, we aim to extend our method to modality extension scenarios. Additionally, we will design effective spatial structures in a block-wise manner to further improve memory efficiency (feasibility can be found in Appendix D).

### ACKNOWLEDGMENTS

This work is supported by the National Natural Science Foundation of China (Grant No. 62476090, 62302167, U23A20343, 62502159), Science and Technology Commission of Shanghai Municipality (Grant No. 25511102700), Natural Science Foundation of Shanghai (Grant No. 25ZR1402135), Natural Science Foundation of Chongqing (Grant No. CSTB2023NSCQ-JQX0007, CSTB2025NSCQ-GPX0445), Open Project Program of the State Key Laboratory of CAD&CG (Grant No. A2501), Zhejiang University, Open Research Fund of Key Laboratory of Advanced Theory and Application in Statistics and Data Science-MOE, ECNU.

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

# A SUPPLEMENTARY RESULTS OF CONTINUAL INSTRUCTION TUNING

Table 8: Comparisons with baselines on LLaVA-1.5-13B, CoIN benchmark.

| Method | Venue | Accuracy on Each Task | | | | | | | | Overall Results | | |
|---|---|---|---|---|---|---|---|---|---|---|---|---|
| | | 1-ScienceQA | 2-TextVQA | 3-ImageNet | 4-GQA | 5-VizWiz | 6-Grounding | 7-VQAv2 | 8-OCRVQA | Avg.ACC(↑) | Forgetting(↓) | New.ACC(↑) |
| LoRA (Hu et al., 2022) | ICLR'22 | 60.03 | 41.19 | 10.62 | 31.03 | 32.67 | 2.60 | 46.33 | 61.00 | 35.68 | 32.90 | 64.47 |
| MT (Zhu et al., 2024) | ICML'24 | 80.43 | 60.72 | 46.70 | 60.35 | 49.19 | 33.16 | 63.74 | 65.44 | 57.47 | 11.26 | 67.32 |
| PGP (Qiao et al., 2025b) | ICLR'24 | 82.50 | 60.64 | 49.15 | 62.53 | 49.43 | 37.37 | 65.57 | 65.82 | 59.13 | 10.11 | 67.98 |
| EProj (He et al., 2023) | ArXiv'23 | 77.65 | 58.93 | 92.31 | 60.22 | 38.27 | 33.77 | 64.39 | 65.80 | 61.42 | 5.84 | 66.53 |
| CIA (Qiao et al., 2025a) | ICLR'24 | 83.94 | 61.40 | 97.05 | 62.61 | 43.99 | 39.72 | 66.29 | 65.78 | 65.10 | 2.31 | 67.12 |
| **Ours** | - | **83.82** | **61.99** | **93.98** | **59.49** | **57.98** | **36.03** | **66.09** | **64.67** | **65.51** | **2.08** | **67.32** |

Table 9: Comparisons with baselines on Qwen-VL, CoIN benchmark.

| Method | Venue | Accuracy on Each Task | | | | | | | | Overall Results | | |
|---|---|---|---|---|---|---|---|---|---|---|---|---|
| | | 1-ScienceQA | 2-TextVQA | 3-ImageNet | 4-GQA | 5-VizWiz | 6-Grounding | 7-VQAv2 | 8-OCRVQA | Avg.ACC(↑) | Forgetting(↓) | New.ACC(↑) |
| LoRA (Hu et al., 2022) | ICLR'22 | 31.05 | 42.45 | 29.57 | 55.57 | 15.30 | 40.33 | 67.75 | 47.80 | 41.23 | 19.36 | 58.17 |
| EWC (Kirkpatrick et al., 2017) | PNAS'17 | 64.30 | 58.67 | 44.04 | 57.73 | 38.16 | 48.04 | 66.98 | 41.76 | 52.46 | 8.68 | 60.06 |
| PGP (Qiao et al., 2025b) | ICLR'24 | 66.42 | 41.33 | 32.16 | 49.83 | 36.05 | 24.22 | 58.60 | 43.96 | 44.07 | 5.90 | 48.30 |
| **Ours** | - | **77.84** | **65.36** | **67.84** | **60.47** | **53.83** | **68.54** | **69.37** | **61.42** | **65.58** | **4.21** | **69.27** |

Table 10: Comparisons with baselines on Qwen2.5-VL-Instruct, Continual-NExT benchmark.

| Method | Venue | Accuracy on Each Task & Overall Results | | | | | | | | |
|---|---|---|---|---|---|---|---|---|---|---|
| | | 1-ArxivQA | 2-GeoChat | 3-IconQA | 4-ClevrMath | 5-CodeQA | 6-ImageNet | 7-Flickr30k | 8-DocVQA | 9-TextVQA |
| Zero-shot | - | 70.23 | 67.53 | 28.60 | 81.93 | 3.63 | 35.62 | 8.97 | 82.09 | 68.94 |
| LoRA (Hu et al., 2022) | ICLR'22 | 71.39 | 89.00 | 73.40 | 96.80 | 2.05 | 78.77 | 20.92 | 85.04 | 68.94 |
| LwF (Li & Hoiem, 2017) | TPAMI'16 | 72.60 | 92.33 | 58.56 | 95.40 | 3.49 | 76.53 | 21.42 | 83.77 | 71.86 |
| EWC (Kirkpatrick et al., 2017) | PNAS'17 | 73.28 | 92.00 | 67.87 | 89.20 | 3.53 | 76.06 | 20.53 | 80.61 | 53.82 |
| Replay (Rolnick et al., 2019) | NIPS'19 | 71.37 | 87.57 | 78.37 | 98.83 | 2.45 | 83.29 | 20.81 | 86.32 | 71.28 |
| MoELoRA (Chen et al., 2024a) | NIPS'24 | 72.19 | 91.53 | 74.36 | 97.66 | 2.65 | 90.75 | 21.08 | 85.51 | 79.55 |
| **PCLR** | - | **74.72** | **92.13** | **88.10** | **97.27** | **8.45** | **74.06** | **21.93** | **86.27** | **80.63** |
| **PCLR-LwF (PCLR variant)** | - | **74.43** | **92.23** | **88.60** | **97.80** | **8.93** | **84.42** | **20.75** | **86.19** | **80.38** |
| | | 10-MathQA | 11-ChartQA | 12-PathVQA | 13-Grounding | 14-ScienceQA | 15-WikiQA | Avg.ACC(↑) | Forgetting(↓) | New.ACC(↑) |
| Zero-shot | - | 0.03 | 75.00 | 33.22 | 72.23 | 82.95 | 2.90 | 47.59 | - | - |
| LoRA (Hu et al., 2022) | ICLR'22 | 34.81 | 73.88 | 7.96 | 86.60 | 80.08 | 4.87 | 58.30 | 9.36 | 67.04 |
| LwF (Li & Hoiem, 2017) | TPAMI'16 | 50.72 | 67.28 | 7.57 | 87.60 | 91.44 | 16.27 | 59.79 | 10.98 | 70.04 |
| EWC (Kirkpatrick et al., 2017) | PNAS'17 | 48.31 | 67.36 | 13.26 | 86.99 | 87.05 | 15.64 | 58.37 | 11.89 | 69.47 |
| Replay (Rolnick et al., 2019) | NIPS'19 | 45.50 | 74.72 | 8.66 | 86.11 | 90.90 | 16.11 | 61.49 | 6.40 | 67.46 |
| MoELoRA (Chen et al., 2024a) | NIPS'24 | 2.14 | 74.80 | 6.35 | 87.68 | 80.90 | 16.90 | 58.94 | 11.81 | 69.96 |
| **PCLR** | - | **42.85** | **76.76** | **39.44** | **86.24** | **87.27** | **7.74** | **64.26** | **5.18** | **69.08** |
| **PCLR-LwF (PCLR variant)** | - | **45.16** | **76.08** | **44.43** | **85.53** | **88.38** | **9.32** | **65.78** | **3.74** | **69.26** |

As shown in Table 8, we discover that PCLR achieves the highest overall performance on the CoIN benchmark, LLaVA-1.5-13B. Compared to the former best regularization method PGP (Qiao et al., 2025b), the Avg.ACC improves by 6.38 and the Forgetting decreases by 8.03, demonstrating strong continual learning ability. Furthermore, our method outperforms the extension methods Eproj (He et al., 2023) and CIA (Qiao et al., 2025a). Experimental results show that our method is robust across model scales and maintains strong performance on larger and powerful models.

As shown in Table 9, we discover that PCLR achieves the highest overall performance on the CoIN benchmark, Qwen-VL. Compared to the former best regularization method EWC (Kirkpatrick et al., 2017), the Avg.ACC improves by 13.12 and the Forgetting decreases by 4.47, demonstrating strong continual learning ability. Experimental results show that our method is robust across models of different architectures and maintains strong performance even when the base architecture varies.

As shown in Table 10, our method exceeds the best of other methods (Replay), improving Avg.ACC by 2.77 and reducing Forgetting by 1.22, which demonstrates that PCLR sustains strong performance on the long-term Continual-NExT benchmark, Qwen2.5-VL-Instruct. Similar to the main paper, we introduce PCLR-LwF, which divides the whole task sequence into five consecutive groups, and applies LwF in each group. This integration with simple baseline yields notable gains by improving Avg.ACC by 1.52 and reducing Forgetting by 1.44, while preserving the same parameter number as PCLR.

In summary, extensive experiments on Large Multimodal Models (LMMs), including LLaVA-1.5-7B, LLaVA-1.5-13B and Qwen-VL on the CoIN benchmark, LLaVA-1.5-hf and Qwen2.5-VL-Instruct on the Continual-NExT benchmark, demonstrate the superiority of PCLR. The method achieves an effective balance between plasticity, stability and memory efficiency. Importantly, PCLR prevents unbounded parameter growth via the Compression–Integration–Learning (CIL) pipeline. Moreover, its modular design enables seamless integration with regularization methods such as LwF, as demonstrated by the PCLR-LwF variant, which further improves training efficiency without sacrificing performance. These results establish PCLR as a scalable, efficient and robust framework for multimodal continual instruction tuning in LMMs.

## B   DETAILS OF EXPERIMENTAL SETTING

**Overall Setting:** For each dataset, we set the training epoch to 1. The LLM backbone and cross-modal modules are incorporated into the PCLR framework. Throughout all phases, only the inserted PCLR modules are trainable, and the LMM remains frozen. During learning, the learning rate is set to $2 \times 10^{-4}$ for LLaVA series, $2 \times 10^{-5}$ for Qwen-VL, $1 \times 10^{-4}$ for Qwen2.5-VL-Instruct. During integration, only data from the current task is used, the learning rate is set to $5 \times 10^{-5}$ for LLaVA-1.5, $1 \times 10^{-5}$ for Qwen-VL, while $2 \times 10^{-6}$ for LLaVA-1.5-hf and Qwen2.5-VL-Instruct. Weight decay is set to 0. The maximum input embedding length is fixed as 2048. We employ gradient checkpointing and mixed-precision training using TF32 and BF16. We use DeepSpeed ZeRO-2 for distributed training. All experiments are conducted on $4 \times 80$-GB GPUs. During learning, we employ a rank-64 extension and later switch to a rank-32 extension. During compression, we adopt a retention rate of $100\%$ (no compression) for the first three tasks, then set it to $75\%$, and finally adjust it to $87.5\%$.

**LoRA Rank Configuration**: In the main paper, all baselines are fine-tuned using LoRA with rank 128. However, due to the MoE architecture of PCLR, it is infeasible to ensure a fair comparison of both total and activated ranks simultaneously. To balance parameter efficiency and memory cost while preserving the MoE structure, we adopt: total rank 256, activated rank 64 (activated during forward passes in both training and inference), and optional shared rank 32 (activated during forward passes in both training and inference, trainable during Integration and frozen during Learning). This setting balances total rank and activated rank to maximize overall fairness.

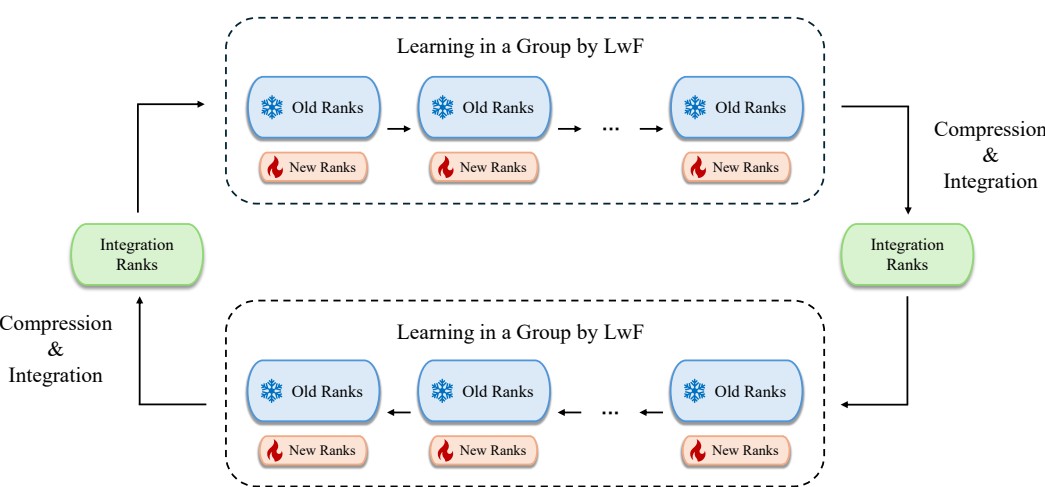

Figure 5: PCLR-LwF Overview. In long-term CIT, the task sequence is divided into several groups, with the tasks in each group being contiguous in the original sequence. Within each group, we apply LwF for learning. Between the groups, the CIL process is adopted.

**Combination with LwF**: In the main paper, we propose an optimization for long-term continual instruction tuning: instead of running CIL after each task, we perform constrained updates in the newly extended rank space using regularization methods such as LwF (Li & Hoiem, 2017), for a subset of tasks. As described in the main paper, on the Continual-NExT benchmark, we group consecutive tasks into a cluster and apply LwF for constrained updates within each cluster, with a weight of 0.1. This approach reduces training cost and improves performance while maintaining the same total rank as the standard PCLR. The PCLR-LwF pipeline is shown in figure 5.

**Equivalent Setting for Fake Query Sampling**: The main paper states that fake queries are sampled from the key pool. However, during continual compression, the number of keys from earlier tasks decays exponentially. This produces unfair sampling of synthetic task-related queries. After completing each task, we average its newly generated keys and place the resulting mean vector into a global task key pool. Each task thus retains a single representative fake query, which keeps cross-task sampling balanced during the integration.

## C  THE CASES OF CONTINUAL INSTRUCTION TUNING

In this section, we present evaluation cases after completing the CoIN (Chen et al., 2024a) benchmark by using Pretrain, MoELoRA, PCLR, and Ground-Truth.

**Model after continual tuning, test on ScienceQA**

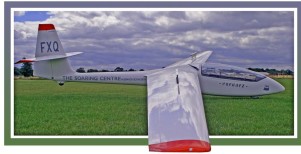

Q: Which ocean is highlighted?
A.the Atlantic Ocean
B.the Indian Ocean
C.the Arctic Ocean
D.the Southern Ocean
Answer with the option's letter from the given choices directly.

A — Answer from Pretrain
The correct answer is C. the Arctic Ocean.

A — Answer from MoELoRA
A

A — Answer from PCLR
C

A — Answer from Ground-Truth
C

---

**Model after continual tuning, test on TextVQA**

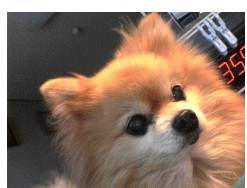

Q: Which centre is featured on the plane?
Reference OCR token: FXQ, THE, SOARING, CENTRE, HUSBANDS, BOSWORTH, PuCHACZ
Answer the question using a single word or phrase.

A — Answer from Pretrain
The plane features the FXQ centre.

A — Answer from MoELoRA
Soaring Centre Husband & Bosworth

A — Answer from PCLR
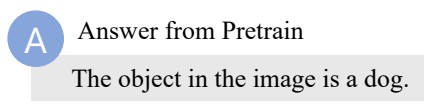
soaring centre

A — Answer from Ground-Truth
the soaring centre

---

**Model after continual tuning, test on ImageNet**

A — Answer from Pretrain
The object in the image is a dog.

A — Answer from MoELoRA
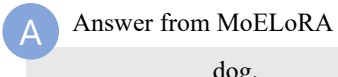
dog.

A — Answer from PCLR
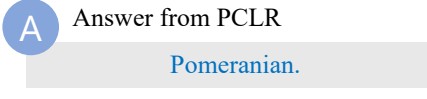
Pomeranian.

A — Answer from Ground-Truth
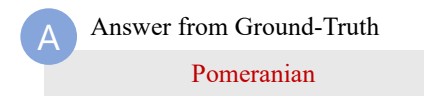
Pomeranian

Q: What is the object in the image?
Give the object in the image in the form of a single word or a concise phrase.

Figure 6: From top to bottom, these are cases on ScienceQA, TextVQA, and ImageNet after learning the final task of the CoIN benchmark.

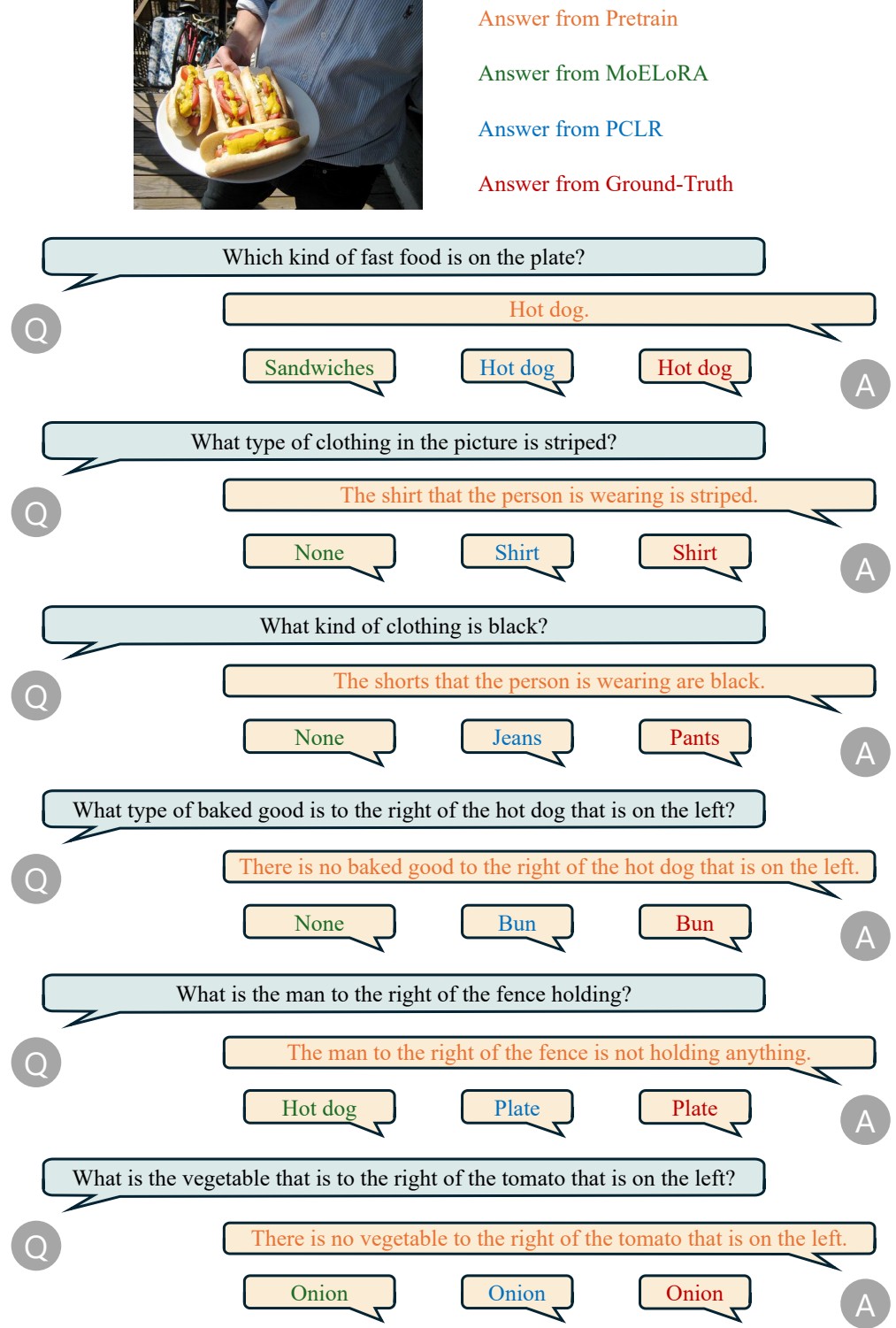

Figure 7: The case on GQA after learning the final task of the CoIN benchmark. This is a case of multiple rounds of dialogue.

**Model after continual tuning, test on VizWiz**

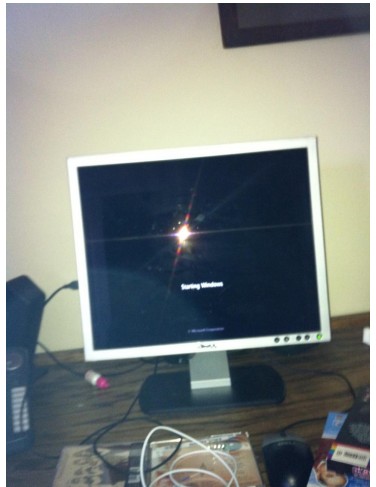

**Question1:** Can you tell me what the error message is on the Windows screen?
When the provided information is insufficient, respond with 'unanswerable'.
Answer the question using a single word or phrase.

**Pretrain:** unanswerable
**MoELoRA:** unanswerable
**PCLR: starting windows**
**Ground-Truth:** starting windows

**Question2:** What is the title of this book?

**Pretrain:** I'm sorry, but I cannot answer your question as there is no book visible in the provided image. Please provide a different image or question.
**MoELoRA:** Windows 7: The Missing Manual
**PCLR: unanswerable**
**Ground-Truth:** unanswerable

**Model after continual tuning, test on Grounding**

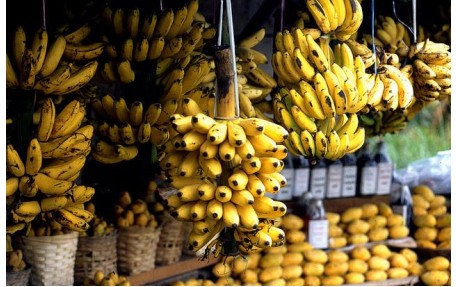

**Question1:** Identify and provide the coordinates of the bounding box described in the sentence: bananas in the middle.

**Pretrain:** I'm sorry, but I cannot provide the bounding box coordinates for the sentence \"bananas in the middle\" as it is not clear what region or object is being described. Please provide more context or a specific sentence for me to assist you.
**MoELoRA:** [0.4,0.52,0.61,0.69]
**PCLR: [0.32,0.39,0.67,0.76]**
**Ground-Truth:** [0.35,0.4,0.64,0.76]

**Question2:** Give the bounding box coordinates as described in the sentence: top left bunch.

**Pretrain:** I'm sorry, but I cannot provide the bounding box coordinates for the sentence \"top left bunch\" as it is not clear what region or object is being described. Please provide more context or a specific sentence for me to assist you.
**MoELoRA:** [0.4,0.52,0.61,0.69]
**PCLR: [0.0,0.24,0.25,0.49]**
**Ground-Truth:** [0.0,0.22,0.24,0.4]

Figure 8: From top to bottom, these are cases on VizWiz and Grounding after learning the final task of the CoIN benchmark.

# D    LAYER SENSITIVITY ANALYSIS IN PCLR

In this section, we investigate the sensitivity of different LMM layers to compression and integration in PCLR for continual learning. Based on the 40-layer LLaVA-1.5-13B Transformer architecture, we divide it into 5 groups of 8 consecutive layers (Lower: layers 1–8, Middle-Lower: layers 9–16, Middle: layers 17–24, Middle-Upper: layers 25–32, Upper: layers 33–40). In each experiment, we skip the Compression and Integration stages for one group (opened group), while keeping the other four groups unchanged. The results in Table 11 show that task-specific sensitivity varies across layers, with the highest performance degradation observed in upper layers compression (-0.53@Avg.ACC) and the lowest in mid-level layers compression (-0.15@Avg.ACC). This observation highlights the layer-specific characteristics in PCLR.

Table 11: The results of LLaVA-1.5-13B on **different opened groups**.

| Layers | Accuracy on Each Task | | | | | | | | Evaluation Metrics | |
|---|---|---|---|---|---|---|---|---|---|---|
| | ScienceQA | TextVQA | ImageNet | GQA | VizWiz | Grounding | VQAV2 | OCRVQA | Avg.ACC | Forgetting |
| Origin | 83.82 | 61.99 | 93.98 | 59.49 | 57.98 | 36.03 | 66.09 | 64.67 | 65.51 | 2.08 |
| Lower | 86.02 | 62.28 | 94.28 | 60.40 | 56.80 | 35.33 | 66.51 | 63.99 | 65.70 | 1.56 |
| Middle-Lower | 86.02 | 63.18 | 94.59 | 60.43 | 56.54 | 36.21 | 66.05 | 62.93 | 65.74 | 1.36 |
| Middle | 83.99 | 62.49 | 94.89 | 59.87 | 58.76 | 34.93 | 66.14 | 64.19 | 65.66 | 1.73 |
| Middle-Upper | 83.80 | 62.04 | 95.07 | 59.58 | 58.44 | 36.70 | 66.16 | 64.31 | 65.76 | 1.67 |
| Upper | 83.85 | 61.89 | 96.46 | 60.02 | 59.23 | 36.13 | 66.15 | 64.56 | 66.04 | 1.32 |

We hypothesize that lower layers focus on input comprehension, mid-level layers focus on reasoning, and upper layers focus on instruction-following output generation. Since reasoning processes across tasks often share a high degree of similarity (general logic patterns), mid-level compression causes minimal interference. However, upper layers face significant task-specific divergence in output generation (single-choice answers in ScienceQA *v.s.* classification answers in ImageNet *v.s.* bounding-box answers in Grounding), leading to pronounced conflicts during compression-integration. Notably, the compression of lower layers is especially sensitive to ScienceQA because its inputs mix text-only and image–text samples, unlike tasks that are exclusively image–text. This indicates that heterogeneous modality distributions can amplify compression-induced performance drops in the lower layers.

Our findings motivate a **layer-aware compression strategy** for future CIL optimization: applying maximal compression retention ratio to upper layers (high conflict), moderate compression retention ratio to lower layers (moderate conflict), and minimal compression retention ratio to mid-level layers (low conflict). This optimization is expected to achieve further performance improvements while maintaining a similar memory budget.

# E    EVALUATION METRICS

We emphasize that our evaluation of prediction accuracy is based on a comparison between the outputs of LMMs and the corresponding ground-truth annotations. This evaluation protocol in Chen et al. (2024a), provides a rigorous and consistent criterion. For short textual outputs we apply **Truth Alignment**, and for longer textual outputs we compute semantic similarity using Sentence-Transformers (Reimers & Gurevych, 2019) and accept when the similarity score is at least 0.8.

We adopt 3 primary metrics to comprehensively evaluate continual instruction tuning performance:

**Average Accuracy (Avg.ACC)** measures the average test accuracy across all datasets, reflecting the overall performance of models throughout the continual learning process.

**Forgetting (FOR)** quantifies the decline in performance on previously learned datasets after training on new datasets. It serves as an indicator of **stability** (retaining previous knowledge).

**New Accuracy (New.ACC)** computes the average test accuracy on newly introduced datasets. It serves as an indicator of **plasticity** (adapting to new tasks).

Overall, these metrics are generally defined as follows:

$$\text{Avg.ACC} = \frac{1}{T} \sum_{i=1}^{T} A_{T,i}, \tag{8}$$

$$\text{FOR} = \frac{1}{T-1} \sum_{i=1}^{T-1} \left( \max_{j \in [i,T]} A_{j,i} - A_{T,i} \right), \tag{9}$$

$$\text{New.ACC} = \frac{1}{T} \sum_{i=1}^{T} A_{i,i}, \tag{10}$$

where $T$ denotes the total number of datasets, $A_{T,i}$ represents the accuracy of the $i$-th dataset evaluated on the model after training on the $T$-th (final) dataset, $A_{j,i}$ is the accuracy of the $i$-th dataset on the model after training on the $j$-th dataset, and $A_{i,i}$ is the accuracy of the $i$-th dataset evaluated immediately after its own training.

## F  VISUALIZATION OF EXPERT TRANSITION DURING CIT

In this section, we visualize the expert transition process during continual instruction tuning (CIT) within the PCLR method. We follow the task sequence defined by the CoIN benchmark: ScienceQA → TextVQA → ImageNet → GQA → VizWiz → Grounding → VQAv2 → OCRVQA. We present the rank experts activation patterns on the last five tasks. Specifically, the invocation rate of the rank experts on the current task and previously learned tasks is shown as below.

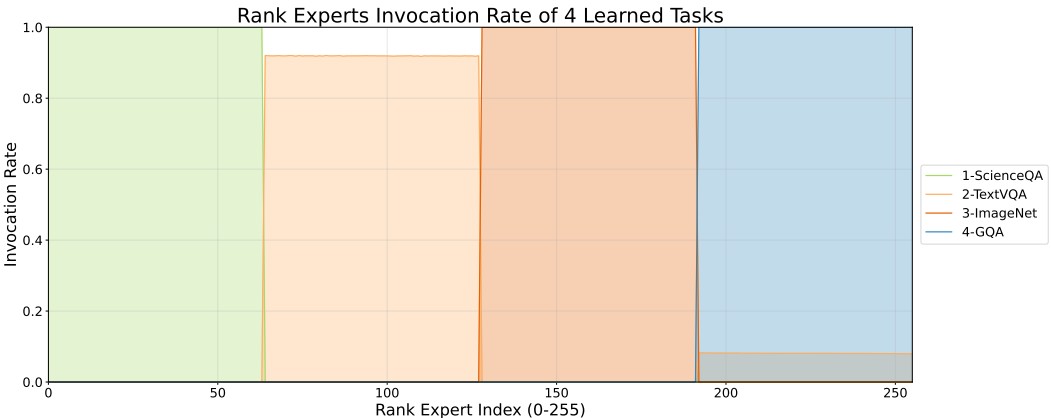

Figure 9: The invocation rate of rank experts after learning the 4-th task GQA.

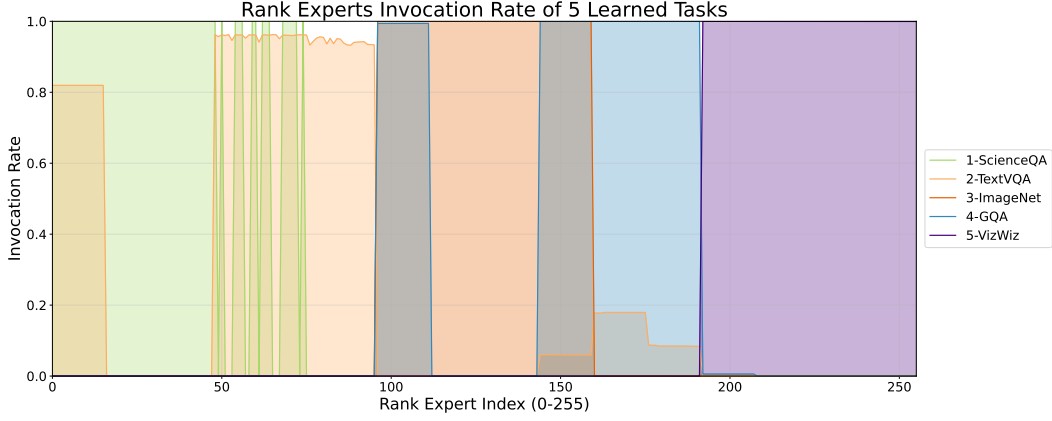

Figure 10: The invocation rate of rank experts after learning the 5-th task VizWiz.

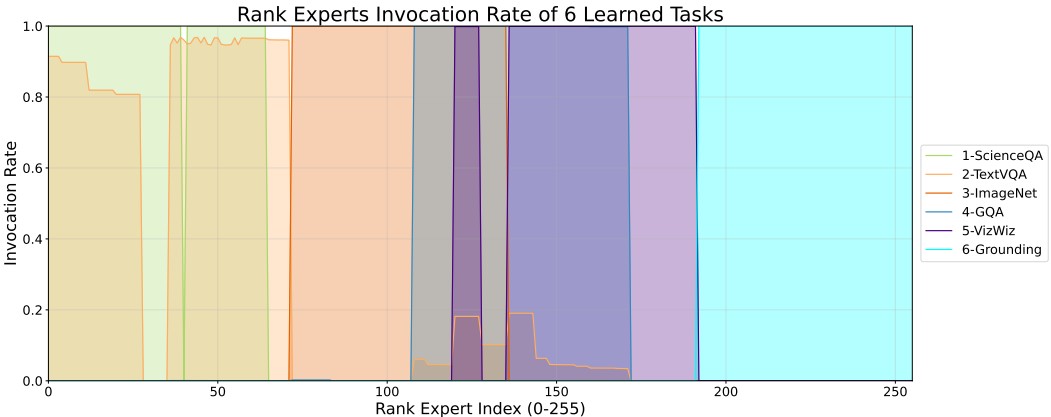

Figure 11: The invocation rate of rank experts after learning the 6-th task Grounding.

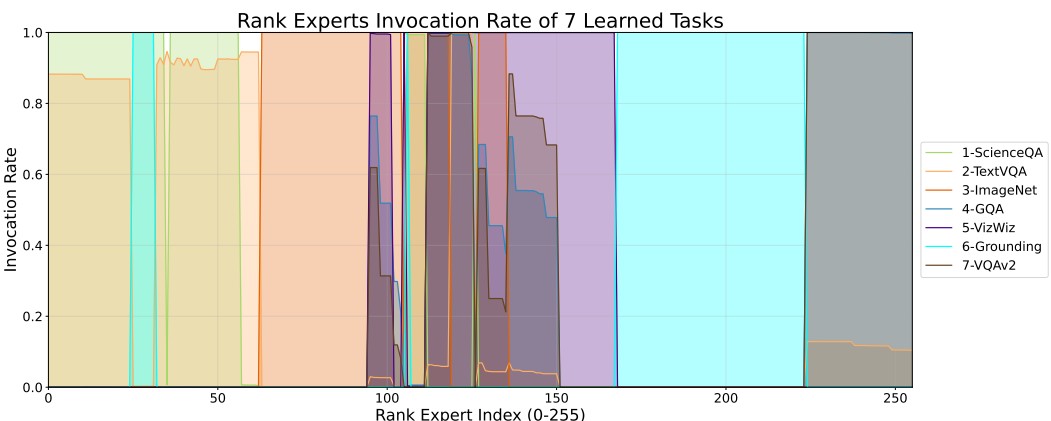

Figure 12: The invocation rate of rank experts after learning the 7-th task VQAv2.

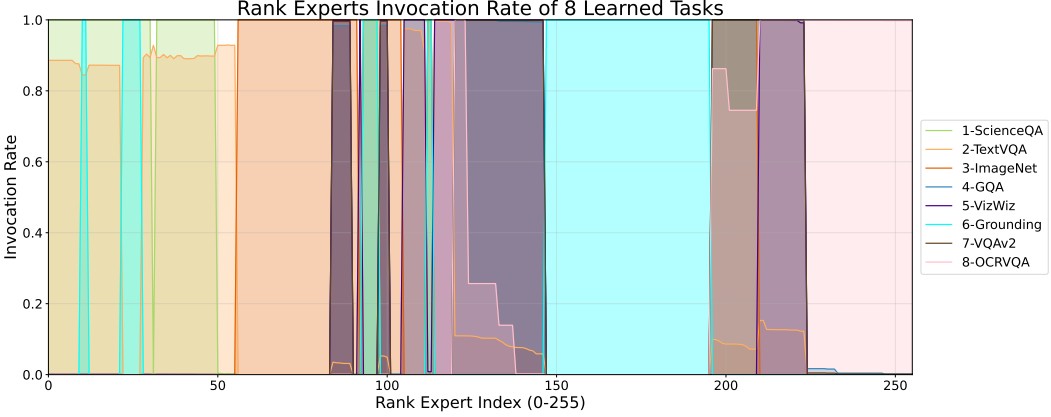

Figure 13: The invocation rate of rank experts after learning the 8-th task OCRVQA.

## G  THE RESULTS OF CONTINUAL-NEXT BENCHMARK

We present the performance of PCLR and its variant PCLR-LwF on the Continual-NExT bench-mark, and we visualize the final accuracies of two representative models (LLaVA-1.5-hf and

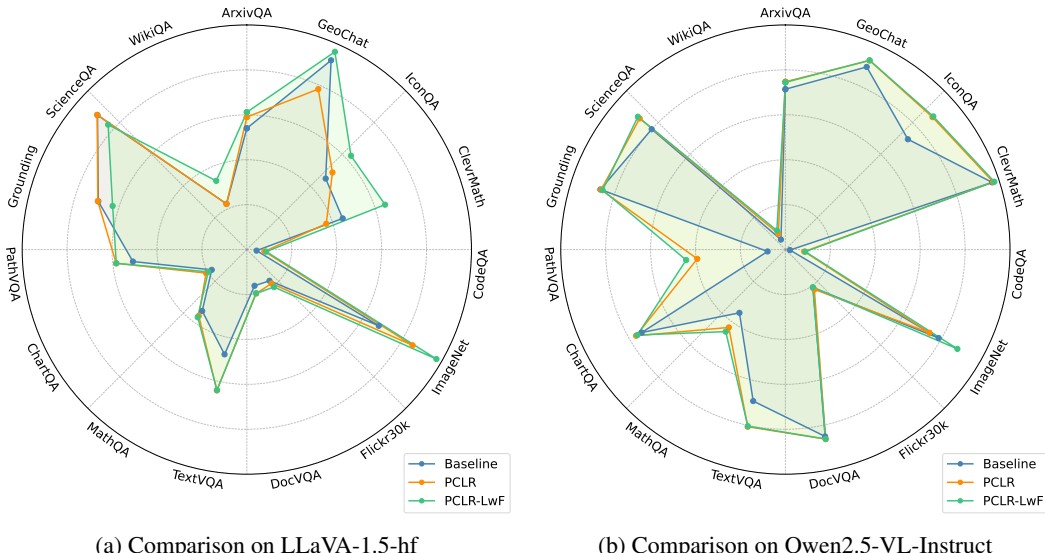

(a) Comparison on LLaVA-1.5-hf  (b) Comparison on Qwen2.5-VL-Instruct

Figure 14: Radar chart of comparisons on Accuracy between baseline (LoRA) and ours.

Qwen2.5-VL-Instruct) using radar charts. As shown in Figure 14, PCLR achieves significant baseline improvements. Notably, PCLR-LwF incorporates LwF (Li & Hoiem, 2017) which is regularized to reduce the compression demand, achieving improved computational efficiency while further improving the performance of PCLR. Thus, PCLR enables compatibility with other continual learning algorithms, offering a practical solution for deployment in real-world applications.

## H  THREE TYPES OF TUNING ORDER SEQUENCES

To evaluate robustness under different Continual Instruction Tuning orders, we employ three task ordering strategies:

1. **Original Order**: Tasks are presented in the default sequence: ScienceQA, TextVQA, ImageNet, GQA, VizWiz, Grounding, VQAv2, OCRVQA.

2. **Reverse Order**: The sequence is reversed relative to the original: OCRVQA, VQAv2, Grounding, VizWiz, GQA, ImageNet, TextVQA, ScienceQA.

3. **Alphabetical Order**: Tasks are sorted alphabetically: GQA, Grounding, ImageNet, OCRVQA, ScienceQA, TextVQA, VizWiz, VQAv2.

## I  THREE TYPES OF INSTRUCTION TEMPLATES

To evaluate robustness under different instruction templates, we adopt three types of instruction templates. Detailed examples of these templates are provided in Table 20.

1. **Original Instruction Template**: Each task is associated with a single instruction, and multiple tasks may share identical instruction formats. This setting reflects minimal prompt diversity.

2. **Diverse Instruction Template**: Each task uses a single, uniquely designed instruction. Instructions are carefully tailored to reflect the semantic and structural characteristics of individual tasks, maximizing prompt diversity.

3. **10Type Instruction Template**: Each task is assigned approximately ten distinct instruction variants. While instructions are diverse within each task, certain templates may be shared across related tasks, simulating a balanced scenario between diversity and generalization.

## J  DETAILED SETTINGS FOR DIFFERENT COMPRESSION STRATEGIES

All settings are maintained at total rank = 256, activated rank = 64, and share rank = 32.

Aggressive: The compression retention rate is fixed at 75%, and the number of new rank experts is fixed at 64.

Conservative: The compression retention rate is fixed at 87.5%, and the number of new rank experts is fixed at 64.

Reverse: The compression retention rate changes from 87.5% → 75%, with the number of new rank experts fixed at 32 → 64.

Centralized: The first eight tasks are not compressed, and after all tasks are learned, a 50% retention rate compression is applied (forgetting is calculated based on 8 tasks). In terms of integration data, compared to the Ours (integrating on the current dataset after each learning cycle), it is relatively singular (the last integration only shows OCRVQA data).

Ours: The compression retention rate changes from 75% → 87.5%, with the number of new rank experts fixed at 64 → 32.

The dynamic adjustment strategy we adopt is based on the distillation loss during the integration phase: if, within the 50th–70th training steps of the integration process, at least 25% of the fake queries (i.e., 25% of the learned tasks) exhibit KL divergence losses exceeding a predefined threshold (set to 0.05 in this paper), it is determined that the current compression intensity is too high, causing significant knowledge conflicts between tasks. This occurs because the optimization objectives of different tasks generate severe discrepancies over the same rank components, leading to interference in task-specific information.

In such cases, the system increases the retention rate and re-executes the compression-integration process, reducing the degree of information fusion between tasks to mitigate the loss of historical knowledge. Simultaneously, during the learning phase of subsequent tasks, the number of newly added ranks is correspondingly reduced (e.g., from 64 to 32) to maintain the total parameter capacity constant.

## K  TASK GROUPING OF PCLR-LWF

**Task Grouping for LLaVA-1.5-hf**:

    Group 1: 1-ArxivQA, 2-GeoChat, 3-IconQA
    Group 2: 4-ClevrMath
    Group 3: 5-CodeQA, 6-ImageNet
    Group 4: 7-Flickr30k, 8-DocVQA, 9-TextVQA
    Group 5: 10-MathQA, 11-ChartQA, 12-PathVQA
    Group 6: 13-Grounding, 14-ScienceQA, 15-WikiQA

Using a compression retention ratio of 66.67%, the final PCLR consists of 256 expert ranks.

**Task Grouping for Qwen2.5-VL-Instruct**:

    Group 1: 1-ArxivQA, 2-GeoChat, 3-IconQA
    Group 2: 4-ClevrMath, 5-CodeQA, 6-ImageNet
    Group 3: 7-Flickr30k, 8-DocVQA, 9-TextVQA
    Group 4: 10-MathQA, 11-ChartQA, 12-PathVQA
    Group 5: 13-Grounding, 14-ScienceQA, 15-WikiQA

Using a compression retention ratio of 80%, the final PCLR consists of 256 expert ranks.

This different grouping strategy is motivated by the observation that the LLaVA-1.5-hf after training on ImageNet performs worse on ClevrMath. To address this, we place the ClevrMath task in a

dedicated group. The memory budget can be mitigated by either reducing the compression retention ratio or increasing the number of compression times.

## L COMPARISON OF TRAINING COST WITH MOELORA

Table 12: The Training Cost of LLaVA about **different numbers of experts**.

| Number | Training Cost on Each Task (Unit: Minutes) | | | | | | | | |
|---|---|---|---|---|---|---|---|---|---|
| | 1-ScienceQA | 2-TextVQA | 3-ImageNet | 4-GQA | 5-VizWiz | 6-Grounding | 7-VQAV2 | 8-OCRVQA | Average |
| 1 | 10.05 | 33.55 | 91.22 | 118.12 | 17.78 | 79.58 | 93.44 | 135.68 | 72.43 |
| 2 | 17.73 | 50.97 | 130.61 | 173.58 | 27.94 | 115.71 | 136.53 | 188.86 | 105.24 |
| 4 | 21.25 | 68.14 | 189.05 | 245.97 | 35.25 | 169.12 | 205.52 | 286.92 | 152.65 |
| **Ours** | **8.86** | **34.43** | **94.82** | **115.49** | **19.85** | **93.68** | **121.87** | **177.12** | **83.27** |

As discussed in the main paper, all computations in our fine-grained LRP, are fully parallelizable. On LLaVA-13B, we compare LRP (total rank 256, activated rank 64, shared rank 32) with MoELoRA (total and activated rank of 128) variants having 1, 2, and 4 experts, as summarized in Table 12.

The forward computation of MoELoRA is defined as:

$$y = xW + \beta \sum_{i=1}^{t} s(x)_i x A_i B_i^T,$$ (11)

where $s(x) \in \mathbb{R}^t$ denotes the expert routing scores, $t$ is the number of total experts, $k$ is the number of activated experts, and $A_i, B_i$ are the LoRA weights of the $i$-th expert.

The proposed LRP forward computation in the main paper is formulated as:

$$y = xW + \beta_s x A_s B_s^T + \beta_m F\left(x A_m, gate\left(Kq, r\right)^T\right) B_m^T,$$ (12)

where $A_s, B_s$ represent the globally shared components, and $A_m, B_m$ represent the mixture expert components. The operator $F(U, v)$ broadcasts $v \in \mathbb{R}^b$ to match the shape of $U \in \mathbb{R}^{a \times b}$, followed by an element-wise multiplication. Here, $K \in \mathbb{R}^{n \times d}, q \in \mathbb{R}^d$ are the L2-regularized key pool and query, $n$ is the number of total rank experts, $r$ is the number of activated rank experts.

MoELoRA consistently activates all ranks and performs serial summation across experts, whereas LRP avoids serial computation by directly integrating expert scores into the LoRA formulation. Specifically, we set the number of experts to match the LoRA rank to align the expert score dimension with the LoRA rank. This design enables parallel integration of expert scores into the LoRA forward computation. This yields efficiency comparable to MoELoRA with one expert because GPU matrix multiplication is faster than serial multiplication and summation over smaller matrices at the same total rank.

## M INSPIRATIONS FROM ADALORA AND THE L2P SERIES

AdaLoRA (Zhang et al., 2023) converts each LoRA adapter into a SVD-style representation, enabling dynamic rank adjustment under a fixed parameter budget and thereby improving computational allocation efficiency. The L2P series (Wang et al., 2022c;b; Smith et al., 2023b) decomposes conventional prompts or prefixes into subcomponent representations paired with learnable keys. It further introduces a dynamic parameter allocation mechanism, markedly improving performance and efficiency in continual learning for the vision domain (Krizhevsky et al., 2009; Peng et al., 2019).

Inspired by AdaLoRA, we likewise decompose LoRA updates into rank vectors but remove explicit rank constraints, and adopt an L2P-like key–value matching scheme, we associate each rank vector with a learnable key and allow dynamic extension during CIT. This simultaneously (i) extends the L2P series to the more parameter-efficient LoRA paradigm, facilitating adaptation to LMMs, and (ii) generalizes adaptive rank allocation principle of AdaLoRA to a Mixture-of-Experts (MoE) structure tailored for continual learning. In summary, we introduce the LoRA Rank Pool (LRP), an extremely fine-grained MoE architecture that offers maximum flexibility for knowledge employment and editing to enhance stability and plasticity.

# N  EXPLANATION OF RELATIVE REDUNDANCY RANK

In this section, we provide a detailed definition of the relative redundancy rank, namely the relative redundancy rank of $M$ to $N$ for the LoRA pair $\langle M, N \rangle$. We select an arbitrary identically named weight pairs $\{(A_M, B_M), (A_N, B_N)\}_i$ and perform an orthogonal triangle decomposition on $A_N$:

$$QR = A_N, \tag{13}$$

where $A_M, A_N \in \mathbb{R}^{d_{in} \times r}$, $Q \in \mathbb{R}^{d_{in} \times r}$, and $Q$ is a set of orthogonal vectors of $A_N$. Subsequently, we apply L2 normalization to every column of $A_M$, producing $\widetilde{A_M}$. For each $a \in \mathbb{R}^{d_{in} \times 1}$ in $\widetilde{A_M}$, we use $Q$ to reconstruct:

$$a^* = QQ^T a, \tag{14}$$

Therefore, we can obtain the reconstruction loss for the part $\langle a, A_N \rangle$:

$$\mathcal{L}_a = 1 - \|a^*\|_2^2. \tag{15}$$

$\mathcal{L}_a \in [0, 1]$ reflects the degree of linear dependence between $a$ and the vectors in $A_N$, a smaller value indicates a higher degree of linear dependence. When $\mathcal{L}_a = 0$, $a$ and $A_N$ are linearly dependent. When $\mathcal{L}_a = 1$, the inner product between $a$ and every $v \in A_N$ is zero.

Similarly, we can obtain the reconstruction loss for the part of $\langle b, B_N \rangle$, $\mathcal{L}_b \in [0, 1]$. Next, we define the reconstruction loss for the rank $(a, b) \in (A_M, B_M)$ with respect to LoRA $(A_N, B_N)$:

$$\mathcal{L} = \mathcal{L}_a * \mathcal{L}_b, \tag{16}$$

$\mathcal{L} \in [0, 1]$ reflects the degree of linear dependence between the rank component $(a, b) \in (A_M, B_M)$ and the LoRA weight pair $(A_N, B_N)$. According to the theoretical insights from LoRA (Hu et al., 2022) and AdaLoRA (Zhang et al., 2023), rank components that are linearly dependent are redundant, and they carry knowledge similar to other rank components, which is the knowledge redundancy referred to in the main text.

We define the threshold $\sigma = 0.001$. For each rank component $(a, b) \in (A_M, B_M)$ relative to the LoRA weight $(A_N, B_N)$, if its score $\mathcal{L} < \sigma$, this component is deemed a redundant rank. Next, we obtain the redundancy rank ratio of $(A_M, B_M)$ relative to $(A_N, B_N)$ (the corresponding LoRA weight pairs $\{(A_M, B_M), (A_N, B_N)\}_i$). Finally, by computing $\mathcal{L}$ of all attention modules, we obtain the relative redundancy rank ratio for $\langle M, N \rangle$ ($M$ relative to $N$).

# O  THE EFFECT OF EPOCH ON PCLR

To verify the impact of epoch count on PCLR's performance and forgetting resistance, we conducted experiments on the LLaVA-1.5-7b model and the CoIN benchmark, setting training cycles of 1, 3, and 5 epochs to compare the performance of PCLR and PGP:

Table 13: Results of LLaVA-1.5-7B on **PGP and PCLR methods with different epochs**.

| Methods-Epoch | Final Accuracy on Each Task | | | | | | | | Overall Results | | |
|---|---|---|---|---|---|---|---|---|---|---|---|
| | ScienceQA | TextVQA | ImageNet | GQA | VizWiz | Grounding | VQAv2 | OCRVQA | Avg.ACC | Forgetting | New.ACC |
| PGP-1 | 85.17 | 56.85 | 32.26 | 61.74 | 49.43 | 32.74 | 65.74 | 62.20 | 55.77 | 12.94 | 67.09 |
| PGP-3 | 85.38 | 57.14 | 32.59 | 62.15 | 49.68 | 33.12 | 66.06 | 62.43 | 56.07 | 12.76 | 67.24 |
| PGP-5 | 85.50 | 57.43 | 32.75 | 62.31 | 49.88 | 33.46 | 66.22 | 62.67 | 56.28 | 12.63 | 67.33 |
| PCLR-1 | 78.33 | 58.24 | 86.08 | 58.14 | 57.61 | 33.04 | 64.17 | 61.92 | 62.19 | 3.39 | 65.16 |
| PCLR-3 | 84.56 | 59.73 | 91.66 | 57.31 | 58.12 | 35.65 | 65.03 | 62.06 | 64.27 | 2.75 | 66.68 |
| PCLR-5 | 86.04 | 59.24 | 93.29 | 55.64 | 57.81 | 36.82 | 64.41 | 61.69 | 64.37 | 2.36 | 66.43 |

Note: To ensure fairness, the integration data usage (IDU) for PCLR-1, PCLR-3, and PCLR-5 remains the same.

PGP shows limited improvement over longer epochs: Although PGP restricts updates through gradient constraints, during longer epoch training, the trainable parameters gradually skew toward the optimal solution for the current task, leading to overwriting of historical knowledge and maintaining high forgetting rates ($12.94 \rightarrow 12.76 \rightarrow 12.63$).

PCLR's CIL mechanism demonstrates stronger performance over longer epochs: During training with 3 and 5 epochs, PCLR's forgetting rate remains significantly lower than PGP (2.75 *v.s.* 12.94,

2.36 *v.s.* 12.63), and the Avg.ACC. continues to improve (62.19 → 64.27 → 64.37), the Forgetting continues to reduce (3.39 → 2.75 → 2.36).

We can analyze the reasons for the performance improvement from the perspective of CIL. Learning phase: New tasks only update newly added rank experts, avoiding interference with historical knowledge, so increasing training epochs does not lead to forgetting. Integration phase: A well-learned set of parameters has less noise, and it is easier for compatible representations to emerge across different tasks, which is beneficial for integration. Knowledge distillation merges new and old knowledge, ensuring stability improvement. The time spent in this phase has a positive feedback relationship with the effectiveness of forgetting resistance.

In summary, PCLR's design not only adapts to tasks with varying training lengths but also excels in multi-epoch training, ensuring efficient knowledge integration and low forgetting rates.

## P    MORE ABLATION EXPERIMENTS

In this section, we discuss how to balance CIT performance and memory efficiency from two perspectives: knowledge density and memory scale. For more intuitive comparisons, we use **total rank experts + shared ranks** to measure the static storage parameters (SSP) and **activated rank experts + shared ranks** to measure the inference activated parameters (IAP).

Table 14: Results of LLaVA-1.5-7B on **different knowledge density**.

| Method | Rank Allocation | | Final Accuracy on Each Task | | | | | | | | Overall Results | | |
|---|---|---|---|---|---|---|---|---|---|---|---|---|---|
| | SSP | IAP | ScienceQA | TextVQA | ImageNet | GQA | VizWiz | Grounding | VQAv2 | OCRVQA | Avg.ACC | Forgetting | New.ACC |
| Dense Space | 192+32 | 64+32 | 72.01 | 53.68 | 88.32 | 58.19 | 55.20 | 30.82 | 64.38 | 63.40 | 60.75 | 4.83 | 64.98 |
| Sparse Space | 320+32 | 64+32 | 80.71 | 59.36 | 90.81 | 56.82 | 56.05 | 31.32 | 65.13 | 62.50 | 62.84 | 2.65 | 65.16 |
| Dynamic Space | 384+32 | 64+32 | 81.54 | 59.37 | 93.05 | 56.44 | 57.05 | 30.80 | 64.46 | 62.80 | 63.19 | 2.06 | 65.00 |
| Ours | 256+32 | 64+32 | 78.33 | 58.24 | 86.08 | 58.14 | 57.61 | 33.04 | 64.17 | 61.92 | 62.19 | 3.39 | 65.16 |

**Dense Space:** With total rank experts = 192, the initial compression retention rate is adjusted to 66.67%, and the compression strategy is 66.67% → 88.33%, with new ranks changing from 64 → 32. 1/3 of the experts are activated during inference, resulting in dense knowledge compression and significant overlap among rank experts. Consequently, Avg.ACC decreases (60.75), Forgetting increases (4.83), and SSP reduces by 22.22% compared to ours.

**Sparse Space:** With total rank experts = 320, the initial compression retention rate is adjusted to 80%, and the compression strategy is 80% → 90%, with new ranks changing from 64 → 32. 1/5 of the experts are activated during inference, with weaker sharing among rank experts. Consequently, Avg.ACC increases (62.84), Forgetting decreases (2.65), and SSP rises by 22.22% compared to ours.

**Dynamic Space:** Total rank experts dynamically change from 256 → 384, with 32 ranks compressed and 64 ranks added at each step. This setting achieves the highest Avg.ACC (63.19), the lowest Forgetting (2.06), and the largest final SSP.

The data shows that forgetting largely depends on the **knowledge density** of the PCLR parameter space (activated rank experts / total rank experts). Higher values tend to favor knowledge integration, while lower values favor expert specialization. Knowledge density is a flexible parameter designed to balance memory efficiency and continual learning capability. It can be adjusted based on memory constraints (there is no universally optimal setting; it depends on memory conditions).

In summary, the **optimal compression strategy** involves a transition from **aggressive to conservative**, and the **initial compression rate** can be **adjusted as needed**. We do not recommend treating it as a fixed parameter.

Table 15: Results of LLaVA-1.5-7B on **different memory scale**.

| Method | Rank Allocation | | Final Accuracy on Each Task | | | | | | | | Overall Results | | |
|---|---|---|---|---|---|---|---|---|---|---|---|---|---|
| | SSP | IAP | ScienceQA | TextVQA | ImageNet | GQA | VizWiz | Grounding | VQAv2 | OCRVQA | Avg.ACC | Forgetting | New.ACC |
| Small-Scale | 128+0 | 32+0 | 78.87 | 58.23 | 82.16 | 56.77 | 53.81 | 30.76 | 63.86 | 60.14 | 60.58 | 4.13 | 64.19 |
| No Shared | 256+0 | 64+0 | 78.59 | 58.09 | 88.57 | 56.93 | 55.29 | 31.96 | 64.41 | 62.40 | 62.03 | 3.63 | 65.21 |
| Ours | 256+32 | 64+32 | 78.33 | 58.24 | 86.08 | 58.14 | 57.61 | 33.04 | 64.17 | 61.92 | 62.19 | 3.39 | 65.16 |

**Small-Scale Parameter Space:** Compared to **No Shared**, memory usage is halved, but performance declines across the board: Avg.ACC (62.03 → 60.58), Forgetting (3.63 → 4.13), and New.ACC (65.21 → 64.19). This reflects insufficient plasticity and severe parameter contention, leading to increased forgetting.

**No Shared Ranks:** Compared to ours, the lack of shared ranks during integration prevents the absorption of global knowledge, resulting in higher forgetting (3.39 → 3.63).

The data in the table indicates that the scale of the parameters is positively correlated with model performance. **Small-Scale**, despite having the same total parameter space as regularization-based methods, only utilizes 1/4 of the activated/trained parameters, yet it still outperforms PGP (55.77) and SEFE (58.57), which are among the best regularization-based methods. Both **No Shared** and **Ours**, with fewer activated parameters and a fixed memory budget, surpass the extension-based method Eproj (60.79) in performance.

## Q    MEMORY EFFICIENCY COMPARISON

Using the LLaVA-1.5-7b model on the CoIN benchmark, we further compared PGP, Eproj, PCLR-small (0 shared, 32 active, 128 total), and PCLR-ours (32 shared, 64 active, 256 total), trainable parameters, and final performance (all statistics exclude the base model and focus solely on the adapter components).

Table 16: PGP, parameters and accuracy across tasks.

| Metric | Tasks | | | | | | | |
|---|---|---|---|---|---|---|---|---|
| | ScienceQA | TextVQA | ImageNet | GQA | VizWiz | Grounding | VQAv2 | OCRVQA |
| Total Params | 340.80M | 340.80M | 340.80M | 340.80M | 340.80M | 340.80M | 340.80M | 340.80M |
| Activated Params | 340.80M | 340.80M | 340.80M | 340.80M | 340.80M | 340.80M | 340.80M | 340.80M |
| Trainable Params | 340.80M | 340.80M | 340.80M | 340.80M | 340.80M | 340.80M | 340.80M | 340.80M |
| Final Accuracy | 85.17 | 56.85 | 32.26 | 61.74 | 49.43 | 32.74 | 65.74 | 62.20 |

Table 17: Eproj, parameters and accuracy across tasks.

| Metric | Tasks | | | | | | | |
|---|---|---|---|---|---|---|---|---|
| | ScienceQA | TextVQA | ImageNet | GQA | VizWiz | Grounding | VQAv2 | OCRVQA |
| Total Params | 340.80M | 660.61M | 980.43M | 980.43M | 980.43M | 1300.24M | 1300.24M | 1300.24M |
| Activated Params | 340.80M | 340.80M | 340.80M | 340.80M | 340.80M | 340.80M | 340.80M | 340.80M |
| Trainable Params | 340.80M | 340.80M | 340.80M | 340.80M | 340.80M | 340.80M | 340.80M | 340.80M |
| Final Accuracy | 78.51 | 57.53 | 92.35 | 55.93 | 44.67 | 36.59 | 63.74 | 57.00 |

Table 18: PCLR-small, parameters and accuracy across tasks.

| Metric | Tasks | | | | | | | |
|---|---|---|---|---|---|---|---|---|
| | ScienceQA | TextVQA | ImageNet | GQA | VizWiz | Grounding | VQAv2 | OCRVQA |
| Total Params | 85.79M | 171.58M | 257.37M | 343.16M | 343.16M | 343.16M | 343.16M | 343.16M |
| Activated Params | 85.79M | 85.79M | 85.79M | 85.79M | 85.79M | 85.79M | 85.79M | 85.79M |
| Trainable Params | 85.79M | 85.79M | 85.79M | 85.79M | 85.79M | 85.79M | 42.90M | 42.90M |
| Final Accuracy | 78.87 | 58.23 | 82.16 | 56.77 | 53.81 | 30.76 | 63.86 | 60.14 |

Table 19: PCLR-ours, parameters and accuracy across tasks.

| Metric | Tasks | | | | | | | |
|---|---|---|---|---|---|---|---|---|
| | ScienceQA | TextVQA | ImageNet | GQA | VizWiz | Grounding | VQAv2 | OCRVQA |
| Total Params | 171.58M | 343.16M | 514.74M | 686.31M | 766.70M | 766.70M | 766.70M | 766.70M |
| Activated Params | 171.58M | 171.58M | 171.58M | 171.58M | 251.97M | 251.97M | 251.97M | 251.97M |
| Trainable Params | 171.58M | 171.58M | 171.58M | 171.58M | 171.58M | 171.58M | 85.79M | 85.79M |
| Final Accuracy | 78.33 | 58.24 | 86.08 | 58.14 | 57.61 | 33.04 | 64.17 | 61.92 |

PCLR imposes a strict upper bound on total ranks, halting parameter growth after early expansion via compression-integration. In contrast, Eproj expands continuously, risking parameter explosion. PCLR outperforms both PGP and Eproj in average accuracy (**62.19** *v.s.* **55.77 / 60.79**) and forgetting (**3.39** *v.s.* **12.94 / 5.42**), demonstrating its ability to control memory while maintaining strong continual learning performance.

When the total parameter count is constrained to a level similar to PGP, PCLR still achieves competitive performance with an Avg.ACC of **60.58**, close to Eproj (**60.79**) and significantly higher than PGP (**55.77**). These results demonstrate that PCLR can effectively utilize limited resources, providing robust performance even under strict parameter budget constraints.

## R  MODELS

**LLaVA-1.5** (Liu et al., 2024): LLaVA-1.5 is a foundational Large Multimodal Model (LMM) that integrates Vicuna (Chiang et al., 2023) as its large language model (LLM) backbone and CLIP-ViT (Radford et al., 2021) as the visual encoder. The cross-modal bridge module employs linear projection layers to align visual features with linguistic representations. Specifically, CLIP-ViT extracts image embeddings, which are then projected through a linear transformation to match the hidden dimension of Vicuna. This design enables efficient fusion of visual and textual information for downstream tasks such as visual question answering (VQA) and image captioning. In this paper, we use LLaVA-1.5-7B[1], LLaVA-1.5-13B[2], and LLaVA-1.5-7B-hf[3] (a fine-tuned version of LLaVA-1.5-7B).

**Qwen-VL** (Bai et al., 2023b): Qwen-VL[4] is a Visual-Large Language Model (VLLM) developed by Alibaba Cloud, combining QwenLM (Bai et al., 2023a) as its LLM backbone and a Vision Transformer (ViT) for visual encoding. Its cross-modal bridge module utilizes Q-Former (a transformer-based architecture) designed for multi-modal feature fusion. The Q-Former attends to visual and textual tokens, enabling context-aware interaction between modalities.

**Qwen2.5-VL** (Bai et al., 2025): Qwen2.5-VL is an advanced version of Qwen-VL (Bai et al., 2023b), featuring Qwen2.5-LM (Yang et al., 2025) as its LLM backbone and a cross-modal architecture. Its cross-modal bridge module works as follows: RMSNorm first normalizes the visual features, followed by a three-layer MLP that projects them into the linguistic space. This replaces the Q-Former in Qwen-VL with a lightweight yet powerful structure, enabling more efficient parameter allocation. Qwen2.5-VL-Instruct[5] builds upon Qwen2.5-VL through supervised fine-tuning on curated, high-quality instruction data, improving its adaptability to downstream tasks.

## S  METHODS

**LoRA (Base)** (Hu et al., 2022): LoRA is a parameter-efficient fine-tuning method that introduces low-rank decomposition matrices into the weight matrices of pretrained models. By freezing the original model weights and using low-rank matrices to capture task-specific information, LoRA significantly reduces the number of trainable parameters while preserving the model's generalization ability. It is widely applied in natural language processing and computer vision due to its low memory consumption, fast training speed, and support for multi-task adaptability.

**MoELoRA** (Chen et al., 2024a): MoELoRA combines the Mixture-of-Experts (MoE) mechanism with Low-Rank Adaptation (LoRA) to enhance adaptability in dynamic task environments. It dynamically allocates expert resources and adjusts modality-specific adaptation weights, achieving improved multi-task performance. In our CIT setting, the number of experts is set to 2 per MoE layer. It is static-structure-based.

**Learning Without Forgetting (LwF)** (Li & Hoiem, 2017): LwF addresses catastrophic forgetting through knowledge distillation. A pretrained model guides the current model to retain historical task knowledge without requiring access to old datasets. It enforces alignment between the output distribution of the current model and the teacher model by combining next-token prediction loss and distillation loss with a weighted sum of two losses of two losses. It is regularization-based.

**Elastic Weight Consolidation (EWC)** (Kirkpatrick et al., 2017): EWC introduces a regularization term based on the Fisher Information Matrix (FIM) to protect critical parameters from previous tasks. By computing parameter importance (diagonal elements of FIM) and imposing constraints during new task training, EWC balances old and new task performance. It is regularization-based.

**Gradient Episodic Memory (GEM)** (Lopez-Paz & Ranzato, 2017): GEM prevents forgetting by maintaining episodic memory of historical task samples and enforcing gradient constraints. It projects the current task gradient into a subspace compatible with all historical gradients via

---

[1]https://huggingface.co/liuhaotian/llava-v1.5-7b

[2]https://huggingface.co/liuhaotian/llava-v1.5-13b

[3]https://huggingface.co/llava-hf/llava-1.5-7b-hf

[4]https://huggingface.co/Qwen/Qwen-VL

[5]https://huggingface.co/Qwen/Qwen2.5-VL-3B-Instruct

quadratic programming, ensuring new task updates do not degrade old task performance. It is regularization-based and replay-based. We set 100 samples per task for replay.

**Experience Replay** (Rolnick et al., 2019): Experience Replay mitigates catastrophic forgetting by replaying subsets of historical data during new task training. A replay buffer stores representative samples, and the model alternates between learning new data and reinforcing old knowledge through mixed training. It is replay-based. We set 100 samples per task for replay.

**Prompt Gradient Projection (PGP)** (Qiao et al., 2025b): PGP proposes a gradient projection approach that enforces model parameter updates to be orthogonal to the previous feature subspace, thereby preserving historical knowledge and enabling adaptation to new tasks. It is regularization-based.

**Model Tailor (MT)** (Zhu et al., 2024): MT restricts training to the critical parameters while compensating for variations in the trainable parameters. It is regularization-based.

**Eproj** (He et al., 2023): Eproj is an advanced dynamic model adaptation method. It groups high-conflict tasks and handles low-conflict tasks via regularization. It is extension-based.

**Dynamic EMA (CIA\*)** (Qiao et al., 2025a): CIA* derives optimal balance weights from the trade-off premise and EMA update, satisfying plasticity-stability conditions. The weights are adaptively determined by gradients and learned parameters. It is regularization-based.

**Dynamic EMA + Instruction Grouping (CIA)** (Qiao et al., 2025a): CIA extends CIA* by introducing instruction grouping to avoid high-conflict tasks. Based on instruction semantic similarity, it determines whether to retrain or extend parameters and allocates the most suitable parameters for testing instances. It is extension-based.

**PCLR (Ours):** PCLR introduces two key innovations to address the challenges of continual instruction tuning (CIT) in Large Multimodal Models (LMMs). First, we decompose LoRA weights into LoRA Rank Pool (LRP), enabling fine-grained control over expert ranks and achieving flexible parameter allocation. This design supports arbitrary compression ratios while preserving task-specific adaptability. Then, we propose the Compression-Integration-Learning (CIL) pipeline, which balances plasticity and stability through three stages:

(1) Compression: It prunes rank experts to reserve space for new task learning.

(2) Integration: It fuses knowledge from similar experts via distillation to enhance synergy.

(3) Learning: It trains new experts in the released space without memory explosion.

**PCLR-LwF (A Simplified Variant of PCLR)** To further enhance the performance of PCLR on long-term CIT while minimizing unnecessary performance degradation and training overhead, we propose PCLR-LwF, a simplified variant that combines PCLR with LwF (Li & Hoiem, 2017). We group consecutive tasks into clusters, we apply LwF within each cluster for continual instruction tuning. This approach preserves the core advantages of PCLR and significantly reducing the cost of integration.

## T  DATASETS

**ScienceQA** (Lu et al., 2022): ScienceQA is a multimodal science question-answering dataset designed to evaluate the ability of models to perform reasoning by integrating visual and textual information. The training dataset comprises 12,726 samples, with 6,218 instances in the image-text modality and 6,508 in the text-only modality. The testing dataset contains 4,241 samples, distributed as 2,017 image-text instances and 2,224 text-only instances.

**TextVQA** (Singh et al., 2019): TextVQA targets text recognition in visual question-answering. The dataset includes real-world images with diverse text formats (handwritten, printed). The training dataset comprises 34,602 samples, all of which belong to the image-text modality. The testing dataset contains 5,000 samples, all of which are in the image-text modality.

**ImageNet** (Russakovsky et al., 2015): ImageNet is a large-scale image classification dataset. The training dataset comprises 129,833 samples, all of which belong to the image-text modality. The testing dataset contains 5,050 samples, all of which are in the image-text modality.

**GQA** (Hudson & Manning, 2019): GQA emphasizes real-world visual reasoning. It evaluates understanding of object relationships and perform multi-step inference. The dataset includes both synthetic and real-world images with scene graphs for structured reasoning. The training dataset comprises 72,140 samples, all of which belong to the image-text modality. The testing dataset contains 12,578 samples, all of which are in the image-text modality.

**VizWiz** (Gurari et al., 2018): VizWiz is a visual question-answering dataset for visually impaired users. The dataset focuses on practical, everyday visual queries. The training dataset comprises 20,523 samples, all of which belong to the image-text modality. The testing dataset contains 4,319 samples, all of which are in the image-text modality.

**Grounding** (Mao et al., 2016): The Grounding tests the ability of models to align natural language instructions with objects in images. It includes image-text pairs where the text describes object locations or attributes, and the task is to predict the corresponding bounding boxes. The training dataset comprises 55,885 samples, all of which belong to the image-text modality. The testing dataset contains 30,969 samples, all of which are in the image-text modality.

**VQAv2** (Goyal et al., 2017): VQAv2 is a foundational visual question-answering dataset. It emphasizes balanced answer distributions and diverse topics. The training dataset comprises 82,783 samples, all of which belong to the image-text modality. The testing dataset contains 214,354 samples, all of which are in the image-text modality.

**OCRVQA** (Mishra et al., 2019): OCRVQA combines optical character recognition with visual question-answering. It evaluates models' ability to parse text from images and generate answers based on the extracted content. The training dataset comprises 165,348 samples, all of which belong to the image-text modality. The testing dataset contains 99,926 samples, all of which are in the image-text modality.

**ArXivQA** (Li et al., 2024b): ArXivQA is a multi-modal dataset for scientific paper analysis. It includes images, formulas, and text from arXiv papers, with questions requiring cross-modal reasoning for tasks such as figure interpretation or explaining derivations. The training dataset comprises 90,000 samples, all of which belong to the image-text modality. The testing dataset contains 10,000 samples, all of which are in the image-text modality.

**GeoChat** (Kuckreja et al., 2024): GeoChat focuses on geospatial reasoning using maps. It includes map images and natural language questions about locations, terrain features, or symbols. The training dataset comprises 25,362 samples, all of which belong to the image-text modality. The testing dataset contains 3,000 samples, all of which are in the image-text modality.

**IconQA** (Lu et al., 2021): IconQA targets abstract icon understanding. It contains icon-text pairs where models must match icons to their descriptions, testing semantic parsing of symbolic visual elements. The training dataset comprises 29,859 samples, all of which belong to the image-text modality. The testing dataset contains 3,000 samples, all of which in the image-text modality.

**ClevrMath** (Lindström & Abraham, 2022): ClevrMath includes synthetic images and math problems (geometry, arithmetic) and requires models to perform compositional logic and computation. The training dataset comprises 40,000 samples, all of which belong to the image-text modality. The testing dataset contains 3,000 samples, all of which are in the image-text modality.

**CodeQA** (Liu & Wan, 2021): CodeQA is a programming question-answering dataset. It pairs code snippets with questions, testing the ability of models to understand and explain code logic across multiple programming languages. The training dataset comprises 150,896 samples, all of which belong to the text-only modality. The testing dataset contains 18,997 samples, all of which in the text-only modality.

**Flickr30k** (Plummer et al., 2015): Flickr30k is an image captioning dataset. It supports fine-grained visual-linguistic alignment tasks. The training dataset comprises 30,000 samples, all of which belong to the image-text modality. The testing dataset contains 1,783 samples, all of which are in the image-text modality.

**DocVQA** (Mathew et al., 2021): DocVQA focuses on document understanding. It includes scanned documents with questions that require text localization and semantic interpretation in complex layouts. The training dataset comprises 39,463 samples, all of which belong to the image-text modality. The testing dataset contains 5,349 samples, all of which are in the image-text modality.

**MathQA** (Amini et al., 2019): MathQA is a mathematical problem solving dataset with text-based questions. It emphasizes interpretability and logical reasoning for algebraic and geometric problems. The training dataset comprises 29,837 samples, all of which belong to the text-only modality. The testing dataset contains 2,985 samples, all of which are in the text-only modality.

**ChartQA** (Masry et al., 2022): ChartQA evaluates quantitative reasoning on charts. Questions involve trend analysis and numerical comparisons. The training dataset comprises 28,299 samples, all of which belong to the image-text modality. The testing dataset contains 2,500 samples, all of which are in the image-text modality.

**PathVQA** (He et al., 2020): PathVQA is a medical pathology dataset. It focuses on disease identification and cellular structure analysis for clinical applications. The training dataset comprises 19,654 samples, all of which belong to the image-text modality. The testing dataset contains 6,719 samples, all of which are in the image-text modality.

**WikiQA** (Yang et al., 2015): WikiQA is an open-domain question-answering dataset with question-answer pairs from Wikipedia. The training dataset comprises 20,360 samples, all of which belong to the text-only modality. The testing dataset contains 633 samples, all of which are in the text-only modality.

Table 20: The list of instructions for each task.

| Task | Original | Diverse | 10Type |
|------|----------|---------|--------|
| ScienceQA | Answer with the option's letter from the given choices directly. | Answer with the option's letter from the given choices directly. | Answer with the option's letter from the given choices directly.
Select the correct answer from the given choices and respond with the letter of the chosen option.
Determine the correct option from the provided choices and reply with its corresponding letter.
Pick the correct answer from the listed options and provide the letter of the selected option.
Identify the correct choice from the options below and respond with the letter of the correct option.
From the given choices, choose the correct answer and respond with the letter of that choice.
Choose the right answer from the options and respond with its letter.
Select the correct answer from the provided options and reply with the letter associated with it.
From the given choices, select the correct answer and reply with the letter of the chosen option.
Identify the correct option from the choices provided and respond with the letter of the correct option.
From the given choices, pick the correct answer and respond by indicating the letter of the correct option. |
| TextVQA | Answer the question using a single word or phrase. | Capture the essence of your response in a single word or a concise phrase. | Answer the question with just one word or a brief phrase.
Use one word or a concise phrase to respond to the question.
Answer using only one word or a short, descriptive phrase.
Provide your answer in the form of a single word or a brief phrase.
Use a single word or a short phrase to respond to the question.
Summarize your response in one word or a concise phrase.
Respond to the question using a single word or a brief phrase.
Provide your answer in one word or a short, descriptive phrase.
Answer the question with a single word or a brief, descriptive phrase.
Capture the essence of your response in one word or a short phrase.
Capture the essence of your response in a single word or a concise phrase. |
| ImageNet | Give the object in the image in the form of a single word or a concise phrase. | Express the object in the image in a single word or a short, descriptive phrase. | Summarize the object in the image in a single word or a brief phrase.
Provide the object in the image using a single word or a brief phrase.
Give the object in the image in the form of a single word or a concise phrase.
Express the object in the image with one word or a short, descriptive phrase.
Identify the type of content in the image using one word or a concise phrase.
Respond to the object in the image with a single word or a short, descriptive phrase.
Describe the content of the image using one word or a concise phrase.
Express the object in the image in a single word or a short, descriptive phrase.
Use a single word or a short phrase to categorize the image content.
Classify the image content using only one word or a brief phrase.
Use one word or a short phrase to classify the content of the image. |
| GQA | Answer the question using a single word or phrase. | Respond to the question briefly, using only one word or a phrase. | Respond to the question with a single word or a short phrase.
Respond to the question using only one word or a concise phrase.
Answer the question with a single word or a brief phrase.
Respond with one word or a short phrase.
Provide your answer in the form of a single word or a concise phrase.
Respond to the question with just one word or a brief phrase.
Answer the question using a single word or a concise phrase.
Provide your response using only one word or a short phrase.
Respond to the question with a single word or a brief phrase.
Respond to the question using just one word or a concise phrase.
Answer the question with one word or a short phrase. |
| VizWiz | Answer the question using a single word or phrase. | Provide a succinct response with a single word or phrase. | Answer the question using only one word or a concise phrase.
Respond to the question using only one word or a concise phrase.
Respond to the question with a single word or a brief phrase.
Provide your answer using just one word or a short phrase.
Respond with one word or a concise phrase.
Answer the question with just one word or a brief phrase.
Use a single word or a short phrase to answer the question.
Provide your answer in the form of one word or a brief phrase.
Reply to the question using one word or a concise phrase.
Answer with a single word or a short phrase.
Use one word or a brief phrase to answer the question. |
| Grounding | Please provide the bounding box coordinate of the region this sentence describes. | Please provide the bounding box coordinate of the region this sentence describes. | Identify and provide the bounding box coordinates that match the description given in this sentence.
Extract and provide the bounding box coordinates based on the region described in the sentence.
Please provide the bounding box coordinate of the region this sentence describes.
Find and provide the bounding box coordinates for the region mentioned in the sentence.
Provide the coordinates of the bounding box that correspond to the region described in the sentence.
Give the bounding box coordinates as described in the sentence.
Determine and provide the bounding box coordinates based on the description in the sentence.
Identify and provide the coordinates of the bounding box described in the sentence.
Provide the coordinates for the bounding box based on the region described in the sentence.
Extract and provide the coordinates for the bounding box described in the sentence.
Identify and give the coordinates of the bounding box as described by the sentence. |
| VQAv2 | Answer the question using a single word or phrase. | Answer the question using a single word or phrase. | Answer the question using a single word or phrase.
Answer the question with a single word or a brief phrase.
Use one word or a short phrase to respond to the question.
Answer the question using just one word or a concise phrase.
Provide your answer to the question using only one word or a brief phrase.
Use a single word or phrase to answer the question.
Provide an answer using only one word or a brief phrase.
Answer the question succinctly with one word or a brief phrase.
Answer the question with just one word or a short phrase.
Respond to the question using a single word or a concise phrase. |
| OCRVQA | Answer the question using a single word or phrase. | Condense your answer for each question into a single word or concise phrase. | Respond to the question with a single word or a short phrase.
Answer the question using a single word or a concise phrase.
Provide your response using only one word or a short phrase.
Use one word or a brief phrase to answer the question.
Reply to the question using one word or a concise phrase.
Use a single word or a short phrase to answer the question.
Use a single word or phrase to answer the question.
Provide an answer using only one word or a brief phrase.
Provide your answer to the question using only one word or a brief phrase.
Respond to the question using a single word or a concise phrase.
Answer the question using a single word or phrase. |

## U   ALGORITHM

---

**Algorithm 1: PCLR**

---

**Input:** Pretrained LMM $F$, number of task $N$, training set $\{\{x_i^t, y_i^t\}_{i=1}^{n_t}\}_{t=1}^{T}$, learning rate $\eta_1$, $\eta_2$, learning loss function $\mathcal{L}_1$, integration loss function $\mathcal{L}_2$, progressive compression retention ratio function $\alpha$, preset total rank number $r_m$.

**Output:** LRP module $M$.

**Initialize:** LRP module $M = \varnothing$, fake query pool $T_K = \varnothing$.

**for** $t = 1, ..., N$ **do**
    1. $M = \text{L}(F, \{x_i^t, y_i^t\}_{i=1}^{n_t}, M, \eta_1, \mathcal{L}_1)$.      # Learning (L).
    2. Collect all trained keys and calculate their mean $q_k$.
    3. $T_K = T_K \cup \{q_k\}$.
    **if** $rank(M) \geq r_m$ **then**
        $M = \text{CI}(F, \{x_i^t, y_i^t\}_{i=1}^{n_t}, T_K, M, \alpha(t), \eta_2, \mathcal{L}_2)$.      # Compression and Integration (CI).
    **end**
**end**

---

**Algorithm 2: Learning** (L)

---

**Input:** Pretrained LMM $F$, current task dataset $D$, LRP module $M_{old}$, learning rate $\eta_1$, learning loss function $\mathcal{L}_1$ according to Eq.(4).

**Output:** LRP module $M$.

**Initialize:** $M_{new}$.

**if** $M_{old} \neq \varnothing$ **then**
    1. Sample a subset from $D$ and calculate their average query $q$.
    2. Compute the cosine similarity scores between the keys in $M_{old}$ and the query $q$, select the top-$r$ indices based on the scores (where $r$ represents the rank of $M_{new}$), and $M_{new}$ is set to the value of the corresponding vector set.
**end**

Extend rank space $M = cat[M_{old}, M_{new}]$, freeze the old part, and activate the new part.

**for** $epoch = 1$ **do**
    **for** $t = 1, ..., T$ **do**
        1. Draw a mini-batch $B = \{x_i^t, y_i^t\}_{i=1}^{b}$.
        2. Calculate $l = \mathcal{L}_1(F, M, B)$.
        3. Using $l$ to backward.
        4. Update $M$ with learning rate $\eta_1$.
    **end**
**end**

---

**Algorithm 3: Compression and Integration** (CI)

---

**Input:** Pretrained LMM $F$, visible datasets $D$, task key pool $K$, original LRP $M$, retention ratio $\alpha$, learning rate $\eta_2$, integration loss function $\mathcal{L}_2$ according to Eq.(5).

**Output:** post-integration LRP $\widetilde{M}$.

**Initialize:** Load $K$ as fake query pool, initialize probability table $P$ of fake query, freeze $F$ and $M$, using $\alpha$ to get post-compression $M$ to initialize $\widetilde{M}$.

**for** $epoch = 1$ **do**
    **for** $t = 1, ..., T$ **do**
        1. Draw a mini-batch $B = \{x_i^t, y_i^t\}_{i=1}^{b}$.
        2. Sample $q$ from $K$ with probability table $P$.
        3. Calculate integration Loss $l = \mathcal{L}_2(F, M, \widetilde{M}, B, q)$.
        4. Using $l$ to update $P$ by Eq.(6).
        5. Using $l$ to backward.
        6. Update $\widetilde{M}$ with learning rate $\eta_2$.
    **end**
**end**

---

---

**Algorithm 4: PCLR-LwF**

---

**Input:** Pretrained LMM $F$, number of task $N$, training set $\{\{x_i^t, y_i^t\}_{i=1}^{n_t}\}_{t=1}^{T}$, learning rate $\eta_1$, $\eta_2$, learning loss function $\mathcal{L}_1$, integration loss function $\mathcal{L}_2$, progressive compression retention ratio function $\alpha$, preset total rank number $r_m$, LwF weight $\beta$, Conflict detection function $E$.

**Output:** LRP module $M$.

**Initialize:** LRP module $M = \varnothing$, fake query pool $T_K = \varnothing$, $n = 0$.

**for** $t = 1, ..., N$ **do**

    **if** $E(F, M)$ **then**

        1. $M = \mathrm{L}(F, \{x_i^t, y_i^t\}_{i=1}^{n_t}, M, \eta_1, \mathcal{L}_1)$.     # Learning (L).

        2. $n = 0$.

    **end**

    **else**

        1. $M = \mathrm{L_{LwF}}(F, \{x_i^t, y_i^t\}_{i=1}^{n_t}, M, \eta_1, \mathcal{L}_1, n, \beta)$.     # Learning with LwF ($\mathrm{L_{LwF}}$).

        2. $n = n + 1$.

    **end**

    1. Collect all trained keys and calculate their mean $q_k$.

    2. $T_K = T_K \cup \{q_k\}$.

    **if** $M \geq r_m$ **then**

        $M = \mathrm{CI}(F, \{x_i^t, y_i^t\}_{i=1}^{n_t}, T_K, M, \alpha(t), \eta_2, \mathcal{L}_2)$.     # Compression and Integration (CI).

    **end**

**end**

---

**Algorithm 5: Learning ($\mathrm{L_{LwF}}$)**

---

**Input:** Pretrained LMM $F$, current task dataset $D$, LRP module $M$, learning rate $\eta_1$, learning loss function $\mathcal{L}_1$ according to Eq.(4), continual LwF number $n$, LwF weight $\beta$.

**Output:** LRP module $M$.

**Initialize:** Copy $M$ as $M_{old}$, and freeze it, set KLD Loss $\mathcal{L}_{KL}$, L2 regularization function $l_2$.

**for** $epoch = 1$ **do**

    **for** $t = 1, ..., T$ **do**

        1. Draw a mini-batch $B = \{x_i^t, y_i^t\}_{i=1}^{b}$.

        2. Calculate $l = \mathcal{L}_1(F, M, B) + \beta \mathcal{L}_{KL}(F, M, M_{old}, B)$.

        3. Using $l$ to backward.

        4. Update $M$ with learning rate $\eta_1$.

    **end**

**end**

1. Take out all the keys $K_{old}$ of $M_{old}$ and all the keys $K$ of $M$.

2. For all $(k_{old,i}, k_i) \in (K_{old}, K)$, $k_i = l_2(n * k_{old,i} + k_i)$.

3. Assign all $k_i \in K$ back.

---

