# OpenReview forum: "PCLR: Progressively Compressed LoRA for Multimodal Continual Instruction Tuning"
_ICLR.cc/2026/Conference — ICLR 2026 Poster_

### Official Review · Reviewer_pBRi · 2025-10-27

**Soundness:** 3
**Presentation:** 3
**Contribution:** 3
**Rating:** 4
**Confidence:** 3

**Summary:**

This paper addresses catastrophic forgetting and unbounded memory growth in Multimodal Continual Instruction Tuning (CIT) for Large Multimodal Models (LMMs) by proposing Progressively Compressed LoRA (PCLR). Through integrating the fine-grained LoRA Rank Pool (LRP) architecture and Compression–Integration–Learning (CIL) pipeline (inspired by human memory consolidation), the authors demonstrate that PCLR balances stability, plasticity, and memory efficiency: LRP decomposes LoRA into rank vectors with learnable keys for flexible knowledge reuse, while CIL prunes redundant experts, fuses knowledge via distillation, and trains new experts in released memory. Experiments demonstrate that PCLR outperforms state-of-the-art baseline methods in terms of average accuracy and forgetting rate, and its variant PCLR-LwF further enhances the performance of long-term CIT.

**Strengths:**

1. It innovatively proposes the LRP architecture, decomposing LoRA into fine-grained rank vector experts with shared components and dynamic gating, which enables flexible knowledge reuse and reduces redundancy, addressing the memory explosion issue of traditional extension methods.
2. The CIL pipeline, inspired by human memory consolidation, balances catastrophic forgetting and memory efficiency through compression-pruning redundant experts, integration-compensating performance loss via distillation, and learning-allocating capacity for new tasks, achieving a stable-plastic-memory balance.

**Weaknesses:**

1. The paper lacks sufficient analysis on why the progressive learning process adjusts the number of new ranks and compression retention rate in specific ways (e.g., 75% to 87.5%), with no clear theoretical or experimental justification for these hyperparameter choices.
2. It fails to compare PCLR with more multimodal continual instruction tuning methods published in 2024 and 2025, such as SEFE[1], HiDe-LLaVA[2], ProgLoRA[3].

[1] SEFE: Superficial and Essential Forgetting Eliminator for Multimodal Continual Instruction Tuning, ICML 2025

[2] HiDe-LLaVA: Hierarchical Decoupling for Continual Instruction Tuning of Multimodal Large Language Model, ACL 2025

[3] Progressive LoRA for Multimodal Continual Instruction Tuning, ACL 2025

**Questions:**

1. In Section 4.4, Progressive Learning mentions that "as the number of tasks increases, fewer new ranks are allocated and a lighter, high-retention compression scheme is adopted", but the quantitative criteria for adjustment are not clearly defined. Is "a certain degree of task increase" based on a task quantity threshold or a model knowledge density indicator? What is the basis for reducing the rank from 64 to 32 in the Experimental Setup Section?
2. When comparing with the LoRA baseline—where LoRA uses a rank of 128, while PCLR adopts a "total rank of 256 and activated rank of 64"—the authors claim to "balance parameter efficiency," yet fail to align the "effective trainable parameter scale" between the two methods. Since PCLR involves fewer parameters actually participating in computation, a critical question arises: Does the "shared expert reusing historical knowledge" implicitly increase the effective parameter count? It is necessary to supplement direct comparisons of parameter scale and computational cost between the two methods.
3. In the experiments, all tasks were trained with 1 epoch, and the impact of the number of epochs on PCLR was not verified. If a new task is highly complex and requires multi-epoch training, should the compression-integration frequency of CIL be adjusted? Will multi-epoch training exacerbate the forgetting of old tasks, and will PCLR's anti-forgetting ability still be superior to that of baseline methods?

---

> ### Author Response · Authors · 2025-11-21
>
> Dear Reviewer pBRi,
>
> We sincerely appreciate the effort you have devoted to reviewing our manuscript and the valuable comments you have provided. Below is our response:

---

> ### Author Response · Authors · 2025-11-21
> **Response to Weakness 1**
>
> **Weakness 1: Experimental and Theoretical Analysis of the Progressive Learning Process.**
>
> **Response:**
>
> Thank you to the reviewer for pointing out this critical issue. We have supplemented the experimental analysis and theoretical explanation of the hyperparameter scheduling strategy (increasing the retention rate from 75% to 87.5% and reducing the number of new ranks from 64 to 32) and clarified why a two-stage dynamic adjustment was adopted instead of a fixed configuration.
>
> ### **Experimental Analysis**
> We compared four configurations on the CoIN benchmark (8 tasks) with the base model LLaVA-1.5-7b, activated rank experts of 64, and shared ranks of 32:
> ||ScienceQA|TextVQA|ImageNet|GQA|VizWiz|Grounding|VQAv2|OCRVQA|Avg.ACC|Forgetting|New.ACC|
> |:-:|:-:|:-:|:-:|:-:|:-:|:-:|:-:|:-:|:-:|:-:|:-:|
> |Setting1|76.02|57.03|82.61|55.84|57.24|30.71|63.86|62.92|60.78|5.09|65.23|
> |Setting2|81.37|59.23|92.28|58.77|54.55|25.88|64.16|59.03|61.91|1.77|63.43|
> |Setting3|81.54|59.37|93.05|56.44|57.05|30.80|64.46|62.80|63.19|2.06|65.00|
> |Setting4|78.33|58.24|86.08|58.14|57.61|33.04|64.17|61.92|62.19|3.39|65.16|
> - **Setting1 (High Plasticity, Low Stability):** The total number of rank experts is 256, with compression always removing 64 rank experts (75% retention rate) and adding 64 new rank experts. Strong task learning ability (high New.ACC), but forgetting is severe (Forgetting = **5.09**), and the performance of early tasks degraded significantly.
> - **Setting2 (Low Plasticity, High Stability):** The total number of rank experts is 256, with compression always removing 32 rank experts (87.5% retention rate) and adding 32 new rank experts. Forgetting is minimal (**1.77**), but the ability to learn new tasks is limited (New.ACC only **63.43**), indicating insufficient parameter updates.
> - **Setting3 (High Plasticity, High Stability, Low Memory Efficiency):** The total number of rank experts transitions from 256 to 384, with compression always removing 32 rank experts and adding 64 new rank experts. Although performance is optimal, this configuration exceeds the fixed memory constraint, violating the design goal of PCLR for "bounded resources."
> - **Setting4 (Our Dynamic Strategy):** The total number of rank experts is 256. The number of rank experts removed during compression transitions from 64 to 32 (retention rate 75% → 87.5%), and the number of new rank experts added transitions from 64 to 32. Without increasing the total parameter count, our strategy achieves the best balance between forgetting control and new task adaptability.
>
> In summary, under a fixed parameter budget, optimal continual learning requires dynamically adjusting compression based on knowledge accumulation. Our strategy leverages this insight: early on, low knowledge density permits aggressive compression (75% retention) and more new ranks to accelerate learning. As tasks progress, increasing knowledge density and more enriched shared structures necessitate a higher knowledge retention rate (87.5%) to prevent substantial loss of reusable knowledge, and fewer new ranks are allocated to maintain a fixed capacity. This balances adaptability and retention without increasing parameters, outperforming fixed strategies. This design is not the result of empirical hyperparameter tuning but rather a rational modeling of the knowledge evolution dynamics in continual learning.
>
> ### **Theoretical Analysis**
> 1. **Parameter Space Saturation**
>    In a learning system with fixed capacity, the ability to retain knowledge is inherently limited. As learning progresses continuously, the "knowledge density" within the system steadily increases; if the compression intensity remains constant, the loss of information during later stages of learning will significantly escalate.
> 2. **Optimization Conflicts in Joint Distillation Objectives**
>    During the integration phase, multiple tasks' distillation losses need to be minimized simultaneously:
>    $$
>    \min_{\theta} \sum_{t=1}^{T} P(q_{t}) \left[ D_{\text{KL}}\left( p_t(x) \parallel p_\theta(x) \right) \right],
>    $$
>    where $p_t$ represents the teacher model's output for task $t$. As $T$ increases, distillation losses for different tasks produce competing gradients on the same set of shared rank components, and optimizing for one task may harm others. In such cases, maintaining a low retention rate (encouraging task information fusion) exacerbates this interference.
> ### **Conclusion**
> PCLR’s dynamic strategy adaptively addresses the plasticity–stability trade-off driven by evolving knowledge density under a fixed memory budget. Early on, sparse and less reusable knowledge favors high plasticity for rapid learning; later, denser knowledge and richer shared structures demand high stability to mitigate compression-induced loss. Ignoring this evolution, as our experiments show, leads to significant performance degradation.

---

> ### Author Response · Authors · 2025-11-21
> **Response to Weakness 2**
>
> **Weakness 2: Lack of comparison with more multimodal continual instruction-tuning methods.**
>
> **Response:**
>
> ### **Experiments**
>
> Based on your suggestions, we have supplemented a systematic comparison on the CoIN benchmark with the latest multimodal continual learning methods from 2024–2025, including SEFE [1], HiDe-LLaVA [2], and ProgLoRA [3]. The complete results are shown in the table below (all methods are based on the LLaVA-1.5-7b backbone, epoch=1):
>
> |         | ScienceQA | TextVQA | ImageNet |  GQA  | VizWiz | Grounding | VQAv2 | OCRVQA | Avg.ACC | Forgetting | New.ACC |
> | :-----: | :-------: | :-----: | :------: | :---: | :----: | :-------: | :---: | :----: | :-----: | :--------: | :-----: |
> | Zero-shot   | 49.91    | 2.88    | 0.33     | 2.08  | 0.90   | 0.00      | 0.68  | 0.17   | 7.12    | -          | -       |
> | Multi-Task  | 56.77    | 49.35   | 95.55    | 56.65 | 53.90  | 30.09     | 59.50 | 55.65  | 57.18   | -          | -       |
> | LoRA        | 21.26    | 28.74   | 10.25    | 36.78 | 32.45  | 0.83      | 42.50 | 57.08  | 28.74   | 37.29      | 61.36   |
> | LwF         | 63.14    | 39.60   | 8.90     | 34.83 | 14.53  | 2.48      | 40.67 | 62.35  | 33.31   | 22.32      | 52.58   |
> | EWC         | 67.41    | 40.41   | 8.18     | 35.05 | 37.88  | 2.67      | 41.27 | 61.02  | 36.74   | 20.51      | 54.68   |
> | MoELoRA     | 58.92    | 38.59   | 8.85     | 37.10 | 44.25  | 2.45      | 41.40 | 55.35  | 35.86   | 25.71      | 58.36   |
> | AdaLoRA     | 73.40    | 51.29   | 35.47    | 44.53 | 46.75  | 0.93      | 55.86 | 62.03  | 46.28   | 23.99      | 63.27   |
> | MT          | 79.63    | 55.47   | 35.64    | 58.70 | 44.37  | 32.20     | 62.21 | 61.59  | 53.73   | 14.03      | 66.00   |
> | PGP         | 85.17    | 56.85   | 32.26    | 61.74 | 49.43  | 32.74     | 65.74 | 62.20  | 55.77   | 12.94      | 67.09   |
> | CIA$^{∗}$   | 75.63    | 54.47   | 43.64    | 60.70 | 43.37  | 36.00     | 65.21 | 63.59  | 55.33   | 7.04       | 61.49   |
> | HiDe-LLaVA  | 20.37    | 33.20   | 56.16    | 42.96 | 23.82  |  5.52     | 52.64 | 50.53  | 35.65   | 24.74      | 57.30   |
> | SEFE        | 75.35    | 58.66   | 83.10    | 54.25 | 48.85  | 16.75     | 65.35 | 66.25  | 58.57   | 11.94      | 69.02   |
> | ProgLoRA    | 74.84    | 51.83   | 83.90    | 49.93 | 53.87  | 31.19     | 62.71 | 64.44  | 59.09   | 7.53       | 65.68   |
> | EProj       | 78.51    | 57.53   | 92.35    | 55.93 | 44.67  | 36.59     | 63.74 | 57.00  | 60.79   | 5.42       | 65.54   |
> | **PCLR**    | **78.33**|**58.24**|**86.08**|**58.14**|**57.61**|**33.04**|**64.17**|**61.92**|**62.19**|**3.39**|**65.16**|
>
> PCLR not only outperforms classical baselines but also demonstrates stronger comprehensive continual learning capabilities in comparison with the latest methods. We will update the relevant sections in the revised manuscript and include the complete results in the main table to enhance the timeliness and completeness of the manuscript.
>
> ### **References**
>
> [1] SEFE: Superficial and Essential Forgetting Eliminator for Multimodal Continual Instruction Tuning, ICML 2025.
>
> [2] HiDe-LLaVA: Hierarchical Decoupling for Continual Instruction Tuning of Multimodal Large Language Model, ACL 2025.
>
> [3] Progressive LoRA for Multimodal Continual Instruction Tuning, ACL 2025.

---

> ### Author Response · Authors · 2025-11-21
> **Response to Question 1**
>
> **Question 1: Quantitative criteria and configuration of the progressive learning adjustment strategy.**
>
> **Response:**
>
> ### **Adjustment Basis for Progressive Learning Strategy**
>
> The dynamic adjustment strategy we adopt is based on the distillation loss during the integration phase: if, within the 50th–70th training steps of the integration process, at least 25% of the fake queries (i.e., 25% of the learned tasks) exhibit KL divergence losses exceeding a predefined threshold (set to 0.05 in this manuscript), it is determined that the current compression intensity is too high, causing significant knowledge conflicts between tasks. This occurs because the optimization objectives of different tasks generate severe discrepancies over the same rank components, leading to interference in task-specific information.
>
> In such cases, the system increases the retention rate and re-executes the compression-integration process, reducing the degree of information fusion between tasks to mitigate the loss of historical knowledge. Simultaneously, during the learning phase of subsequent tasks, the number of newly added ranks is correspondingly reduced (e.g., from 64 to 32) to maintain the total parameter capacity constant.
>
> ### **Basis for Reducing New Ranks from 64 to 32**
>
> This decision is primarily based on two points:
>
> **1. Capacity Conservation:** Given that the total rank pool is capped at 256, if 64 new ranks are added per round in the early stages, 64 ranks must be deleted to maintain capacity. In the later stages, continuing to add 64 ranks would force the system to excessively compress existing knowledge. Therefore, we halve the new ranks to 32 and simultaneously reduce the deletion amount from 64 to 32 (i.e., increasing the retention rate from 75% to 87.5%). This significantly reduces the information loss caused by compression without increasing memory overhead.
>
> **2. Best Practices in the LoRA Community:** In instruction fine-tuning of multimodal large models (e.g., LLaVA, Qwen-VL ), **r = 8, 16, 32, 64 are the most widely adopted standard rank configurations** (refer to the official LLaVA repository, HuggingFace PEFT documentation, etc.). Therefore, we align the design of new ranks with this practice: using $r = 64$ in the initial stages to ensure sufficient expressive capacity for independent learning of new tasks, and reducing to $r = 32$ in later stages. This approach not only retains sufficient trainable freedom but also aligns with community-validated effective configurations. Additionally, using powers of 2 for reduction facilitates divisibility while ensuring that the resulting rank values remain within the commonly used and efficient empirical range.

---

> ### Author Response · Authors · 2025-11-21
> **Response to Question 2 (part1)**
>
> **Question 2: Comparison of parameter scale and computational cost between PCLR and baseline methods.**
>
> **Response-part1:**
>
> ### **Does "Shared Experts Reusing Historical Knowledge" Implicitly Increase the Number of Effective Parameters?**
>
> Thank you to the reviewer for highlighting the details of parameter configuration.
>
> Before answering the question, it is important to clarify that shared experts and shared ranks are distinct concepts. Shared ranks are generated upon the first integration to capture common features across global tasks, frozen during the Learning phase, and adjusted during the Integration phase. Shared experts refer to the identical rank components activated across different tasks, meaning the knowledge of these ranks is shared by multiple tasks. I will revise the sections of the manuscript where these terms were uncleared. Below, I provide two versions of the response:
>
> **1. Shared Ranks**
>    As described in **Section 4.2, Formula 2** of the manuscript, shared experts are not included in the calculation of activated rank experts. Therefore, the number of ranks involved in forward computation is the sum of shared ranks and activated rank experts, meaning shared ranks do increase the number of effective parameters. However, during the Learning phase, shared ranks are frozen, thus not contributing to gradient computation or optimizer overhead.
>
> **2. Shared Experts**
>    For a single response, only 64 ranks are fixedly activated within the rank experts for forward propagation, regardless of whether the experts are shared. The phenomenon of "Shared Experts Reusing Historical Knowledge" occurs after the initial compression rate adjustment, where the number of new ranks is less than the activated ranks. This results in some ranks being trainable while others are frozen during the Learning phase. Compared to settings where all activated ranks are trainable, this reduces memory overhead.
>
> ### **Comparison of Parameter Usage and Memory Consumption**
>
> Based on the LLaVA-1.5-7b model on the CoIN benchmark, we further compared **PGP (regularization-based and LoRA-based strong baseline)**, **EProj (extension-based and LoRA-based strong baseline)**, and PCLR under various configurations in terms of total parameter count, activated parameter count (involved in forward/backward propagation), trainable parameter count (during the Learning phase), integrated parameter count (during the Integration phase), and final performance (all statistics exclude the base model and focus solely on the adapter components). Key results are summarized below:
>
>
> ### **PGP (maintains a fixed total parameter count but lacks flexibility in resource allocation):**
> || ScienceQA|TextVQA|ImageNet|GQA|VizWiz|Grounding|VQAv2|OCRVQA|
> |:-:|:-:|:-:|:-:|:-:|:-:|:-:|:-:|:-:|
> |  Total Params   | 340.80M | 340.80M | 340.80M | 340.80M |  340.80M |   340.80M  |  340.80M | 340.80M |
> | Activated Params| 340.80M | 340.80M | 340.80M | 340.80M |  340.80M |   340.80M  |  340.80M | 340.80M |
> | Trainable Params| 340.80M | 340.80M | 340.80M | 340.80M |  340.80M |   340.80M  |  340.80M | 340.80M |
> |  Final Accuracy | 85.17 | 56.85 | 32.26 | 61.74 |  49.43  |   32.74    |  65.74  | 62.20  |
>
> **Avg.ACC is 55.77, Forgetting is 12.94, New.ACC is 67.09.**
>
> ### **Eproj (expands based on task similarity but risks continuous growth with longer task sequences):**
> || ScienceQA|TextVQA|ImageNet|GQA|VizWiz|Grounding|VQAv2|OCRVQA|
> |:-:|:-:|:-:|:-:|:-:|:-:|:-:|:-:|:-:|
> |  Total Params   | 340.80M | 660.61M | 980.43M | 980.43M |  980.43M |  1300.24M  | 1300.24M | 1300.24M |
> | Activated Params| 340.80M | 340.80M | 340.80M | 340.80M |  340.80M |   340.80M  |  340.80M |  340.80M |
> | Trainable Params| 340.80M | 340.80M | 340.80M | 340.80M |  340.80M |   340.80M  |  340.80M |  340.80M |
> |  Final Accuracy | 78.51 | 57.53 | 92.35 | 55.93 |  44.67  |   36.59    |  63.74  | 57.00  |
>
> **Avg.ACC is 60.79, Forgetting is 5.42, New.ACC is 65.54.**

---

> ### Author Response · Authors · 2025-11-21
> **Response to Question 2 (part2)**
>
> **Question 2: Comparison of parameter scale and computational cost between PCLR and baseline methods.**
>
> **Response-part2:**
>
> ### **PCLR (0 shared ranks, 64 active rank experts, 256 total rank experts):**
> || ScienceQA|TextVQA|ImageNet|GQA|VizWiz|Grounding|VQAv2|OCRVQA|
> |:-:|:-:|:-:|:-:|:-:|:-:|:-:|:-:|:-:|
> |Total Params|171.58M|343.16M|514.74M|686.31M|686.31M|686.31M|686.31M|686.31M|
> |Activated Params|171.58M|171.58M|171.58M|171.58M|171.58M|171.58M|171.58M|171.58M|
> |Trainable Params|171.58M|171.58M|171.58M|171.58M|171.58M|171.58M|85.79M|85.79M|
> |Integrated Params|-|-|-|-|514.74M|514.74M|600.54M|600.54M|
> |Final Accuracy|78.59|58.09|88.57|56.93|55.29|31.96|64.41|62.40|
>
> **Avg.ACC is 62.03, Forgetting is 3.63, New.ACC is 65.21.**
>
> ### **PCLR (0 shared ranks, 32 active rank experts, 128 total rank experts):**
> ||ScienceQA|TextVQA|ImageNet|GQA|VizWiz|Grounding|VQAv2|OCRVQA|
> |:-:|:-:|:-:|:-:|:-:|:-:|:-:|:-:|:-:|
> |Total Params|85.79M|171.58M|257.37M|343.16M|343.16M|343.16M|343.16M|343.16M|
> |Activated Params|85.79M|85.79M|85.79M|85.79M|85.79M|85.79M|85.79M|85.79M|
> |Trainable Params|85.79M|85.79M|85.79M|85.79M|85.79M|85.79M|42.90M|42.90M|
> |Integrated Params|-|-|-|-|257.38M|257.38M|300.28M|300.28M|
> |Final Accuracy|78.87|58.23|82.16|56.77|53.81|30.76|63.86|60.14|
>
> **Avg.ACC is 60.58, Forgetting is 4.13, New.ACC is 64.19.**
>
> ### **PCLR (32 shared ranks, 64 active rank experts, 256 total rank experts):**
> || ScienceQA|TextVQA|ImageNet|GQA|VizWiz|Grounding|VQAv2|OCRVQA|
> |:-:|:-:|:-:|:-:|:-:|:-:|:-:|:-:|:-:|
> |Total Params|171.58M|343.16M|514.74M|686.31M|766.70M|766.70M|766.70M|766.70M|
> |Activated Params|171.58M|171.58M|171.58M|171.58M|251.97M|251.97M|251.97M|251.97M|
> |Trainable Params|171.58M|171.58M|171.58M|171.58M|171.58M|171.58M|85.79M|85.79M|
> |Integrated Params|-|-|-|-|595.12M|595.12M|680.92M|680.92M|
> |Final Accuracy|78.33|58.24|86.08|58.14|57.61|33.04|64.17|61.92|
>
> **Avg.ACC is 62.19, Forgetting is 3.39, New.ACC is 65.16.**
>
> PCLR expands in its early stages, but this does not affect long-term CIT memory usage. We set the total rank count to 256 and the activated rank count to 64. When the rank pool is not fully utilized, compression is unnecessary, meaning that for the first four tasks, 64 ranks are added for each learning process. After completing the fourth task, compression and integration are initiated. This approach avoids wasting parameter resources and prevents parameter expansion during subsequent learning.
>
> Although the total parameter count of PCLR is higher than that of PGP, its trainable parameter count gradually decreases as tasks progress, ultimately requiring only approximately **85.79M** trainable parameters and **251.97M** activated parameters, while achieving significantly better performance (Avg.ACC **62.19** v.s. PGP **55.77** / EProj **60.79**) and lower forgetting (**3.39** v.s. **12.94** / **5.42**). This indicates that PCLR, through dynamic expansion and compression, effectively enhances continual learning ability without sacrificing long-term memory control.
>
> While the integration phase involves adjustments to a larger number of parameters, this process is fully controllable and converges rapidly, keeping computational overhead within an acceptable range.
>
> PCLR’s total rank count is set to a fixed upper limit (e.g., 256 + 32). After moderate expansion in the early stages, growth is halted through the compression-integration mechanism, ensuring strict upper bounds for long-term memory usage. In contrast, EProj has a probability of expanding after learning each task, posing a risk of parameter explosion. Experiments demonstrate that PCLR achieves superior performance compared to both PGP and EProj while maintaining long-term memory control, effectively balancing memory efficiency and continual learning performance.
>
> Under the PCLR configuration (0 shared ranks, 64 activated rank experts, 256 total rank experts), despite the total parameter count being significantly reduced to approximately **52.83%** of EProj, PCLR still outperforms EProj in performance (**62.19** v.s. **60.79**), demonstrating stronger continual learning capabilities. When reducing the total parameter count to a level similar to PGP: under the PCLR configuration (0 shared ranks, 32 activated rank experts, 128 total rank experts). Despite the substantial reduction in parameters, PCLR maintains strong competitiveness, achieving an Avg.ACC of **60.58**, close to EProj (**60.79**) and far surpassing PGP (**55.77**). This demonstrates that even under strict parameter budget constraints, PCLR can effectively utilize limited resources to deliver outstanding performance.

---

> ### Author Response · Authors · 2025-11-21
> **Response to Question 3**
>
> **Question 3: Verify whether the number of epochs impacts PCLR's learning.**
>
> **Response:**
>
> ### **Should the compression frequency in CIL be adjusted?**
>
> No. PCLR triggers compression and integration operations only after the training of each task is fully completed, and the compression frequency is not adjusted based on the number of epochs.
>
> ### **Our reasoning is**:
>
> 1. The goal of compression and integration is to achieve offline consolidation of cross-task knowledge, which requires the learning representation of the current task to be stable and sufficient.
> 2. Whether a task uses 1 epoch or multiple epochs, PCLR consistently follows the principle of "integration occurs after full learning", ensuring that each task participates in knowledge integration only after reaching its optimal state.
>
> This design allows PCLR to adapt flexibly to tasks of varying complexity: simpler tasks can be quickly completed and enter the integration phase, while complex tasks are fully optimized through multiple epochs before triggering integration, thereby ensuring learning quality and maintaining the stability of long-term memory.
>
> ### **Experimental Setup and Results**
>
> To verify the **impact of epoch count on PCLR's performance and forgetting resistance**, we conducted experiments on the **LLaVA-1.5-7b** model and the CoIN benchmark, setting training cycles of **1, 3, and 5 epochs** to compare the performance of PCLR and PGP:
>
> | Methods-Epoch | ScienceQA  | TextVQA  | ImageNet  |  GQA  | VizWiz | Grounding | VQAv2 | OCRVQA | Avg.ACC | Forgetting | New.ACC |
> | :-----: | :--------: | :------: | :-------: | :---: | :----: | :-------: | :---: | :----: | :-----: | :--------: | :-----: |
> | PGP-1| 85.17 | 56.85 | 32.26 | 61.74  |  49.43  |   32.74    |  65.74  | 62.20  |  55.77 |  12.94 |  67.09 |
> |PCLR-1| 78.33 | 58.24 | 86.08 | 58.14  |  57.61  |   33.04    |  64.17  | 61.92  |  62.19 |   3.39 |  65.16 |
> | PGP-3| 85.38 | 57.14 | 32.59 | 62.15  |  49.68  |   33.12    |  66.06  | 62.43  |  56.07 |  12.76 |  67.24 |
> |PCLR-3| 84.56 | 59.73 | 91.66 | 57.31  |  58.12  |   35.65    |  65.03  | 62.06  |  64.27 |   2.75 |  66.68 |
> | PGP-5| 85.50 | 57.43 | 32.75 | 62.31  |  49.88  |   33.46    |  66.22  | 62.67  |  56.28 |  12.63 |  67.33 |
> |PCLR-5| 86.04 | 59.24 | 93.29 | 55.64  |  57.81  |   36.82    |  64.41  | 61.69  |  64.37 |   2.36 |  66.43 |
>
> **Note: To ensure fairness, the integration data usage (IDU) for PCLR-1, PCLR-3, and PCLR-5 remains the same.**
>
> **PGP shows limited improvement over longer epochs**: Although PGP restricts updates through gradient constraints, during longer epoch training, the trainable parameters gradually skew toward the optimal solution for the current task, leading to overwriting of historical knowledge and maintaining high forgetting rates (**12.94** -> **12.76** -> **12.63**).
>
> **PCLR's CIL mechanism demonstrates stronger performance over longer epochs**: During training with 3 and 5 epochs, PCLR's forgetting rate remains significantly lower than PGP (**2.75** v.s. **12.94**, **2.36** v.s. **12.63**), and the Avg.ACC. continues to improve (**62.19** -> **64.27** -> **64.37**), the Forgetting continues to reduce (**3.39** -> **2.75** -> **2.36**).
>
> **Reasons from CIL analysis**:
> 1. **Learning phase**: New tasks only update newly added rank experts, avoiding interference with historical knowledge, so increasing training epochs does not lead to forgetting.
> 2. **Integration phase**: A well-learned set of parameters has less noise, and it is easier for compatible representations to emerge across different tasks, which is beneficial for integration. Knowledge distillation merges new and old knowledge, ensuring stability improvement. The time spent in this phase has a positive feedback relationship with the effectiveness of forgetting resistance.
>
> In summary, PCLR's design not only adapts to tasks with varying training lengths but also excels in multi-epoch training, ensuring efficient knowledge integration and low forgetting rates.

---

> ### Author Response · Authors · 2025-11-21
>
> Thank you once again for your thoughtful feedback and dedicated efforts, your input has greatly enhanced the quality and clarity of our paper. We sincerely appreciate the time and care you devoted to helping us improve this work.

---

> ### Author Response · Authors · 2025-11-24
> **Supplementary Answer-part1 (Weakness 1 & Question 1)**
>
> Dear Reviewer pBRi,
>
> Thank you for your concern regarding the hyperparameters. We have conducted multi-dimensional ablation experiments, and please allow me to organize and present them systematically. This will include some results that have already been mentioned earlier, but this step is necessary for a comprehensive and structured explanation. We appreciate your understanding. This will be very helpful in answering your questions about compression retention ratio, compression strategy, and memory usage. It will also help you better understand the key settings of PCLR that balance stability, flexibility, and memory efficiency.
>
> We comprehensively demonstrate the effects of factors such as compression retention rate, the number of new ranks, total ranks, activated ranks, and shared ranks on memory and performance (these results will be included in the appendix). For more intuitive comparisons, we use **total rank experts + shared ranks** to measure the static storage parameters (SSP) and **activated rank experts + shared ranks** to measure the inference activated parameters (IAP).
>
> **Note: Our method adopts total rank experts = 256, activated rank experts = 64, shared ranks = 32, with a compression retention rate of 75% → 87.5%, and a setting for new ranks corresponding to 64 → 32.**
>
> ### **Fixed total rank experts = 256, activated rank experts = 64, shared ranks = 32: Evaluating the impact of compression strategies**
>
> ||SSP|IAP|ScienceQA|TextVQA|ImageNet|GQA|VizWiz|Grounding|VQAv2|OCRVQA|Avg.ACC|Forgetting|New.ACC|
> |:-:|:-:|:-:|:-:|:-:|:-:|:-:|:-:|:-:|:-:|:-:|:-:|:-:|:-:|
> |Setting1|256+32|64+32|76.02|57.03|82.61|55.84|57.24|30.71|63.86|62.92|60.78|5.09|65.23|
> |Setting2|256+32|64+32|81.37|59.23|92.28|58.77|54.55|25.88|64.16|59.03|61.91|1.77|63.43|
> |Setting3|256+32|64+32|79.13|58.79|87.41|58.51|53.39|25.67|64.44|62.76|61.26|3.21|64.07|
> |Setting4|256+32|64+32|77.72|53.78|88.93|60.05|55.71|28.71|62.93|60.16|61.00|3.38|64.37|
> |Ours|256+32|64+32|78.33|58.24|86.08|58.14|57.61|33.04|64.17|61.92|62.19|3.39|65.16|
>
> - **Setting1 (high plasticity, low stability):** The compression retention rate is fixed at 75%, and the number of new rank experts is fixed at 64. While early-stage learning for new tasks is fast (New.ACC is high), forgetting is severe (Forgetting = 5.09), resulting in significant early-stage forgetting.
> - **Setting2 (low plasticity, high stability):** The compression retention rate is fixed at 87.5%, and the number of new rank experts is fixed at 64. Forgetting is minimized (1.77), but the ability to learn new tasks is limited (New.ACC = 63.43), indicating insufficient parameter updates.
> - **Setting3 (reverse variation):** The compression retention rate changes from 87.5% → 75%, with the number of new rank experts fixed at 32 → 64. Both plasticity and forgetting are constrained, and overall performance (Avg.ACC = 61.26) is worse than Setting2.
> - **Setting4 (concentrated compression):** The first eight tasks are not compressed, and after all tasks are learned, a 50% retention rate compression is applied (forgetting is calculated based on 8 tasks). In terms of integration data, compared to the **Ours** (integrating on the current dataset after each learning cycle), it is relatively singular (the last integration only shows OCRVQA data), resulting in lower overall performance.
>
> The data in the table shows that under a fixed initial compression retention rate, reducing the retention rate gradually and smoothly is the most effective learning strategy. This aligns with the understanding of continual learning systems: early tasks have low knowledge density and fewer reusable components, so the information loss caused by compression is minimal, enabling rapid learning. As the task sequence progresses, knowledge density increases, and shared structures across historical tasks become more prominent, making conservative compression more suitable.
>
> The initial compression retention ratio should be set to **1 - activated rank experts / total rank experts**. This allows for rapid learning in the initial tasks, avoiding insufficient adaptability. We will adhere to this setting going forward.

---

> ### Author Response · Authors · 2025-11-24
> **Supplementary Answer-part2 (Question 1 & Question 2)**
>
> ### **Fixed activated rank experts, shared ranks: Evaluating the impact of total rank experts (corresponding changes in initial compression retention rate)**
>
> ||SSP|IAP|ScienceQA|TextVQA|ImageNet|GQA|VizWiz|Grounding|VQAv2|OCRVQA|Avg.ACC|Forgetting|New.ACC|
> |:-:|:-:|:-:|:-:|:-:|:-:|:-:|:-:|:-:|:-:|:-:|:-:|:-:|:-:|
> |Setting5|192+32|64+32|72.01|53.68|88.32|58.19|55.20|30.82|64.38|63.40|60.75|4.83|64.98|
> |Setting6|320+32|64+32|80.71|59.36|90.81|56.82|56.05|31.32|65.13|62.50|62.84|2.65|65.16|
> |Setting7|384+32|64+32|81.54|59.37|93.05|56.44|57.05|30.80|64.46|62.80|63.19|2.06|65.00|
> |Ours|256+32|64+32|78.33|58.24|86.08|58.14|57.61|33.04|64.17|61.92|62.19|3.39|65.16|
>
> - **Setting5 (dense space):** With total rank experts = 192, the initial compression retention rate is adjusted to 66.67%, and the compression strategy is 66.67% → 88.33%, with new ranks changing from 64 → 32. 1/3 of the experts are activated during inference, resulting in dense knowledge compression and significant overlap among rank experts. Consequently, Avg.ACC decreases (60.75), Forgetting increases (4.83), and SSP reduces by 22.22% compared to ours.
> - **Setting6 (sparse space):** With total rank experts = 320, the initial compression retention rate is adjusted to 80%, and the compression strategy is 80% → 90%, with new ranks changing from 64 → 32. 1/5 of the experts are activated during inference, with weaker sharing among rank experts. Consequently, Avg.ACC increases (62.84), Forgetting decreases (2.65), and SSP rises by 22.22% compared to ours.
> - **Setting7 (dynamic space):** Total rank experts dynamically change from 256 → 384, with 32 ranks compressed and 64 ranks added at each step. This setting achieves the highest Avg.ACC (63.19), the lowest Forgetting (2.06), and the largest final SSP.
>
> The data shows that forgetting largely depends on the **knowledge density** of the PCLR parameter space (**activated rank experts / total rank experts**). Higher values tend to favor knowledge integration, while lower values favor expert specialization. **Knowledge density** is a flexible parameter designed to balance memory efficiency and continual learning capability. It can be adjusted based on memory constraints (there is no universally optimal setting; it depends on memory conditions).
>
> In summary, the **optimal compression strategy** involves a transition from **aggressive to conservative**, and the **initial compression rate** can be **adjusted as needed**. We do not recommend treating it as a fixed parameter.
>
> ### **Fixed the ratio of activated rank experts / total rank experts (initial compression rate), compression strategy: Evaluating the impact of shared ranks and parameter space**
>
> ||SSP|IAP|ScienceQA|TextVQA|ImageNet|GQA|VizWiz|Grounding|VQAv2|OCRVQA|Avg.ACC|Forgetting|New.ACC|
> |:-:|:-:|:-:|:-:|:-:|:-:|:-:|:-:|:-:|:-:|:-:|:-:|:-:|:-:|
> |Setting8|128+0|32+0|78.87|58.23|82.16|56.77|53.81|30.76|63.86|60.14|60.58|4.13|64.19|
> |Setting9|256+0|64+0|78.59|58.09|88.57|56.93|55.29|31.96|64.41|62.40|62.03|3.63|65.21|
> |Ours|256+32|64+32|78.33|58.24|86.08|58.14|57.61|33.04|64.17|61.92|62.19|3.39|65.16|
>
> - **Setting8 (small-scale parameter space):** Compared to Setting9, memory usage is halved, but performance declines across the board: Avg.ACC (62.03 → 60.58), Forgetting (3.63 → 4.13), and New.ACC (65.21 → 64.19). This reflects insufficient plasticity and severe parameter contention, leading to increased forgetting.
> - **Setting9 (no shared ranks):** Compared to ours, the lack of shared ranks during integration prevents the absorption of global knowledge, resulting in higher forgetting (3.39 → 3.63).
>
> The data in the table indicates that the scale of the parameters is positively correlated with model performance. Setting8, despite having the same total parameter space as regularization-based methods, only utilizes 1/4 of the activated/trained parameters, yet it still outperforms PGP (55.77) and SEFE (58.57), which are among the best regularization-based methods. Both Setting9 and Ours, with fewer activated parameters and a fixed memory budget, surpass the extension-based method Eproj (60.79) in performance.

---

> ### Author Response · Authors · 2025-11-24
>
> Thank you again for your efforts in helping us improve our work!

---

> ### Author Response · Authors · 2025-11-27
> **Further Discussion Request**
>
> Dear Reviewer pBRi,
>
> We appreciate your time and apologize for the interruption. If you have any questions about our response or any other concerns, we are glad to have further discussion with you.
>
> Thank you for your time in reviewing our paper.

---

> ### Comment · Reviewer_pBRi · 2025-11-28
> **Comment after rebuttal**
>
> Thank you to the authors for their clarification. The authors demonstrated the necessity of progressive learning and clearly explained its specific implementation mechanism. Additionally, the authors validated the performance of PCLR under different training settings. Given the innovation of their method and the sufficiency of their experiments, I believe this is work has a significant contribution, thus I decide to raise my score to 6.

---

> > ### Author Response · Authors · 2025-11-29
> >
> > I'm glad to have resolved your issue, and I sincerely thank you for your positive feedback on our work.

---

### Official Review · Reviewer_U4Ez · 2025-10-29

**Soundness:** 3
**Presentation:** 3
**Contribution:** 3
**Rating:** 6
**Confidence:** 3

**Summary:**

This paper proposes a Compression–Integration–Learning (CIL) pipeline and a LoRA Rank Pool (LRP) architecture for multimodal continual instruction tuning. The method releases model capacity by compressing redundant rank vectors, integrates old knowledge through knowledge distillation, and incrementally learns new tasks, thereby mitigating catastrophic forgetting in multi-task continual learning. Experiments demonstrate that PCLR significantly reduces forgetting on LLaVA and Qwen-VL models, outperforming existing methods on two benchmarks.

**Strengths:**

- The performance is impressive, achieving state-of-the-art results on both datasets.
- The CIL process exhibits a high degree of innovation.

**Weaknesses:**

- The capacity limit of the expert pool may constrain the method’s effectiveness in ultra-long continual learning sequences. When experts from earlier tasks are excessively compressed, complete prevention of forgetting becomes difficult. Therefore, the method may not be well-suited for extremely long task sequences.
- The Integration stage requires additional training, which could lead to longer overall training time compared to existing methods.

**Questions:**

- What is the retention rate used during the Compression stage? What is the rationale behind this choice?
- For the first task, are all experts trained, and is compression applied only after the second task begins?

---

> ### Author Response · Authors · 2025-11-21
>
> Dear Reviewer U4Ez,
>
> We sincerely appreciate the effort you have devoted to reviewing our manuscript and the valuable comments you have provided. Below is our response:

---

> ### Author Response · Authors · 2025-11-21
> **Response to Weakness 1 (part1)**
>
> **Weakness 1: The limitation of fixed capacity and concerns about long-term performance.**
>
> **Response-part1:**
>
> Although PCLR sets a fixed upper limit on the total rank (e.g., 256), its compression-consolidation mechanism does not simply discard parameters. Instead, it transfers early task information to shared/preserved rank subspaces through knowledge distillation. This design significantly alleviates forgetting caused by hard truncation.
>
> ### **On the Impact of "Expert Pool Capacity Limitation" on Continual Learning Performance**
>
> We clarify that the fixed capacity is a design choice of PCLR rather than a strict limitation. The core objective is to prevent "memory explosion" in resource-constrained scenarios, but the framework inherently supports dynamically adjusting the capacity boundary via hyperparameters.
>
> For example, we tested a more flexible configuration: introducing 64 new ranks for each new task, removing only 32 redundant ranks per compression round, and imposing no fixed upper limit on the total rank.
>
> |  Metrics   | ScienceQA  | TextVQA  | ImageNet  |  GQA  | VizWiz | Grounding | VQAv2 | OCRVQA | Avg  |
> | :-----: | :--------: | :------: | :-------: | :---: | :----: | :-------: | :---: | :----: | :-----: |
> | Instant Accuracy| 83.66 | 61.16 | 96.50 |  59.91  |  58.58  |   32.21    |  65.14  |  62.80  |  65.00  |
> |  Final Accuracy | 81.54 | 59.37 | 93.05 |  56.44  |  57.05  |   30.80    |  64.46  |  62.80  |  63.19  |
>
> **63.19/2.06/65.00 (Avg.ACC/Forgetting/New.ACC)**
>
> Under this configuration, the average accuracy (Avg.ACC) reached **63.19**, forgetting rate was **2.06**, and new task accuracy (New.ACC) was **65.00**. In comparison, the default PCLR settings achieved **62.19** (Avg.ACC), **3.39** (Forgetting), and **65.16** (New.ACC). These results indicate that moderately relaxing the capacity constraint significantly improves long-term performance, particularly in reducing forgetting.
>
> In summary, PCLR demonstrates excellent scalability and can flexibly adjust capacity constraints based on practical needs, achieving optimal continual learning performance across different application scenarios. This characteristic further validates PCLR's practical value in large-scale multimodal continual instruction tuning.

---

> ### Author Response · Authors · 2025-11-21
> **Response to Weakness 1 (part2)**
>
> **Weakness 1: The limitation of fixed capacity and concerns about long-term performance.**
>
> **Response-part2:**
>
> ### **Long-Term Performance Validation**
>
> Thank you for your insight. Indeed, in extremely long task sequences, performing compression-consolidation individually for each task may cause early knowledge to gradually dilute due to repeated compression. To address this, we propose a task grouping learning strategy as a natural extension of the PCLR framework: grouping multiple semantically or structurally similar tasks, accumulating learning within the group, and then performing unified compression and consolidation.
>
> To ensure low representational conflicts within a task group, after completing the learning of each new task, we use a small number of samples retained during the task's training (100) to quickly evaluate the performance of the currently optimized parameter set on other tasks within the group. Specifically, if the model's accuracy on the validation set of any existing task drops to 95% of its instant performance, we determine that the current task significantly interferes with the group. We immediately terminate the learning of the current task group: taking the parameter state at the end of the previous task as the ranks for the group's Compression-Integration, and using the parameters obtained from the current task as the starting point of a new group for the next round of cumulative learning. The validation set cache is then cleared.
>
> We developed a PCLR-LwF variant, enabling LwF within groups and maintaining CIL between groups. This was validated on the Continual-Next benchmark (a total of 15 continual learning tasks, currently the longest known multimodal continual instruction tuning benchmark):
>
> ### **Base model: LLaVA-1.5-7b-hf**
>
> |Task|New Accuracy (LoRA)|Final Accuracy (LoRA)|**Instant Accuracy (PCLR-LwF)**|**Final Accuracy (PCLR-LwF)**|
> |:-:|:-:|:-:|:-:|:-:|
> |ArxivQA|66.70|53.99|**66.78**|**61.16**|
> |GeoChat|97.00|92.23|**97.30**|**96.37**|
> |IconQA|67.93|47.23|**69.23**|**62.40**|
> |ClevrMath|77.86|44.86|**79.23**|**64.70**|
> |CodeQA|9.37|4.36|**10.70**|**8.68**|
> |ImageNet|97.64|67.84|**97.39**|**97.46**|
> |Flickr30k|21.31|17.16|**20.02**|**20.63**|
> |DocVQA|23.63|16.47|**23.59**|**19.84**|
> |TextVQA|66.31|47.70|**66.10**|**63.97**|
> |MathQA|41.64|33.80|**41.34**|**37.22**|
> |ChartQA|23.36|18.04|**22.96**|**19.84**|
> |PathVQA|60.44|50.98|**60.73**|**58.42**|
> |Grounding|73.54|69.52|**74.25**|**62.83**|
> |ScienceQA|89.64|89.46|**90.30**|**83.09**|
> |WikiQA|22.27|22.27|**33.49**|**33.49**|
> |Average|55.91|45.06|**56.89**|**52.67**|
>
> Task Grouping:
> - Group 1: 1-ArxivQA, 2-GeoChat, 3-IconQA
> - Group 2: 4-ClevrMath
> - Group 3: 5-CodeQA, 6-ImageNet
> - Group 4: 7-Flickr30k, 8-DocVQA, 9-TextVQA
> - Group 5: 10-MathQA, 11-ChartQA, 12-PathVQA
> - Group 6: 13-Grounding, 14-ScienceQA, 15-WikiQA
>
> ### **Base model: Qwen2.5-VL-Instruct (7B)**
>
> |Task|New Accuracy (LoRA)|Final Accuracy (LoRA)|**Instant Accuracy (PCLR-LwF)**|**Final Accuracy (PCLR-LwF)**|
> |:-:|:-:|:-:|:-:|:-:|
> |ArxivQA|73.43|71.39|**74.69**|**74.43**|
> |GeoChat|92.16|89.00|**93.77**|**92.23**|
> |IconQA|88.70|73.40|**92.87**|**88.60**|
> |ClevrMath|99.60|96.80|**99.40**|**97.80**|
> |CodeQA|7.89|2.05|**8.63**|**8.93**|
> |ImageNet|96.77|78.77|**96.85**|**88.42**|
> |Flickr30k|22.37|20.92|**22.94**|**20.75**|
> |DocVQA|87.01|85.04|**86.93**|**86.19**|
> |TextVQA|72.18|68.94|**81.92**|**80.38**|
> |MathQA|46.90|34.81|**54.14**|**45.16**|
> |ChartQA|76.36|73.88|**77.28**|**76.08**|
> |PathVQA|55.98|7.96|**57.90**|**44.43**|
> |Grounding|87.08|86.60|**87.76**|**85.53**|
> |ScienceQA|94.25|80.08|**94.53**|**88.38**|
> |WikiQA|4.87|4.87|**9.32**|**9.32**|
> |Average|67.04|58.30|**69.26**|**65.78**|
>
>
> Task Grouping:
> - Group 1: 1-ArxivQA, 2-GeoChat, 3-IconQA
> - Group 2: 4-ClevrMath, 5-CodeQA, 6-ImageNet
> - Group 3: 7-Flickr30k, 8-DocVQA, 9-TextVQA
> - Group 4: 10-MathQA, 11-ChartQA, 12-PathVQA
> - Group 5: 13-Grounding, 14-ScienceQA, 15-WikiQA
>
> The experimental results demonstrate that PCLR-LwF effectively mitigates the forgetting problem in continual learning over extremely long task sequences. On the LLaVA-1.5-7b-hf and Qwen2.5-VL-Instruct backbones, the model achieves an average final accuracy of **52.67** and **65.78** across 15 heterogeneous multimodal tasks, with corresponding forgetting rates of only **4.57** and **3.74**, significantly outperforming the baseline (**52.67/65.78** vs. **45.06/58.30**), particularly in terms of early-stage performance differences (e.g., IconQA: **62.40/88.60** vs. **47.23/73.40**). Additional baseline comparisons are presented in detail in the main text and appendix of the manuscript.
>
> In summary, we validate that by reasonably delaying compression and accumulating learning within low-conflict task groups, the issue of knowledge dilution can be effectively alleviated. This enables the successful extension of the PCLR framework to longer task sequences, offering a scalable and practical solution for large-scale continual instruction tuning.

---

> ### Author Response · Authors · 2025-11-21
> **Response to Weakness 2**
>
> **Weakness 2: The integration phase introduces extra training overhead.**
>
> **Response:**
>
> Although the integration phase in PCLR does introduce additional computational overhead, this design is essential for efficient knowledge transfer and long-term stability. Similar to how humans consolidate memories during sleep through offline replay and synaptic pruning, our integration phase compresses and distills the LoRA Rank Pool during a "post-training resting period," enabling structured cross-task knowledge fusion.
>
> ### **Empirical Analysis of the Efficiency-Performance Trade-off**
>
> To quantify the impact of integration overhead, we conducted an ablation study on Integration Data Usage (IDU) using the **LLaVA-1.5-13b** model with 4 GPUs (80GB each). IDU refers to the number of randomly sampled samples from the current task's training set used to drive distillation optimization. "0" indicates that only parameter compression is performed without knowledge integration. "Origin" represents using the full training data of the current task for integration.
>
> | IDU  | Avg.ACC | Forgetting | New.ACC | Avg.Cost |
> |:----:|:-------:|:----------:|:-------:|:--------:|
> |  0   | 59.40   | 8.86       | 67.15   | 0        |
> |  5k  | 64.50   | 3.11       | 67.22   | 11min    |
> | 10k  | 64.92   | 2.61       | 67.20   | 20min    |
> | 20k  | 65.14   | 2.27       | 67.13   | 38min    |
> |Origin| 65.51   | 2.08       | 67.33   | 115min   |
>
> The integration phase indeed introduces additional computational overhead, but empirical results show significant benefits. The ablation study on **LLaVA-1.5-13b** demonstrates that using only 5k samples (approximately 11 minutes) for integration increases the average accuracy from **59.40** to **64.50**, while the forgetting rate drops significantly from **8.86** to **3.11**.
>
> Further increasing the data volume yields diminishing returns: from 10k to full training data, performance improves by only about **0.6**, while computational cost increases nearly fivefold. This indicates that lightweight integration is already sufficiently efficient.
>
> Therefore, while the integration phase is not cost-free, its overhead is controllable, convergence is rapid, and the results are significant. It is a critical component for achieving long-term stability in continual learning, rather than a redundant burden.

---

> ### Author Response · Authors · 2025-11-21
> **Response to Question 1**
>
> **Question 1: Retention settings and selection principles in the compression phase.**
>
> **Response:**
>
> The PCLR framework enables the regulation of plasticity, stability, and memory efficiency:
> 1. **Plasticity**: The number of newly added ranks determines the model's capacity to absorb new task information.
> 2. **Stability**: The retention rate controls the extent of historical knowledge preservation.
> 3. **Memory Efficiency**: The ratio of activated ranks to total ranks affects memory usage during inference and training.
>
> In our experiments, we adopt a dynamic adjustment strategy rather than a fixed value to accommodate the evolving knowledge demands in continual learning.
>
> In the main experiments on the CoIN benchmark:
>
> ||GQA|VizWiz|Grounding|VQAv2|
> |:-:|:-:|:-:|:-:|:-:|
> |Retention rate|75%|75%|87.5%|87.5%|
>
> In the main experiments on the Continual-NeXT benchmark:
>
> ||ClevrMath|CodeQA|ImageNet|Flickr30k|DocVQA|TextVQA|MathQA|ChartQA|PathVQA|Grounding|ScienceQA|
> |:-:|:-:|:-:|:-:|:-:|:-:|:-:|:-:|:-:|:-:|:-:|:-:|
> |Retention rate|75%|75%|75%|75%|75%|75%|87.5%|87.5%|87.5%|87.5%|87.5%|
>
> In PCLR-LwF:
>
> ||Group-4|Group-5|
> |:-:|:-:|:-:|
> |Retention rate|75%|75%|
>
> **Note**: Compression is not required after learning the final task. This setting enhances immediate performance (plasticity).
>
> Our design is based on the principle that as the task sequence grows, the incremental knowledge gain diminishes, and the system should gradually shift towards "conservative integration," reducing the risk of forgetting caused by aggressive compression.
>
> We validated the impact of different retention rate strategies through experiments (base model: LLaVA-1.5-7b, total rank experts: 256, activated rank experts: 64, shared ranks: 32).
>
> ||ScienceQA|TextVQA|ImageNet|GQA|VizWiz|Grounding|VQAv2|OCRVQA|Avg.ACC|Forgetting|New.ACC|
> |:-:|:-:|:-:|:-:|:-:|:-:|:-:|:-:|:-:|:-:|:-:|:-:|
> |Setting1|76.02|57.03|82.61|55.84|57.24|30.71|63.86|62.92|60.78|5.09|65.23|
> |Setting2|81.37|59.23|92.28|58.77|54.55|25.88|64.16|59.03|61.91|1.77|63.43|
> |Setting3|78.33|58.24|86.08|58.14|57.61|33.04|64.17|61.92|62.19|3.39|65.16|
>
> - **Setting1 (High plasticity, low stability)**: Always deletes 64 rank experts during compression (75% retention rate), and the number of newly added rank experts is always 64.
>
> - **Setting2 (Low plasticity, high stability)**: Always deletes 32 rank experts during compression (87.5% retention rate), and the number of newly added rank experts is always 32.
>
> - **Setting3 (Our method)**: The number of deleted rank experts transitions from 64 to 32 (retention rate 75% -> 87.5%), and correspondingly, the number of newly added rank experts transitions from 64 to 32.
>
> Experimental results show that fixed retention rates struggle to balance stability and plasticity: Setting1 (75% retention) suffers from severe forgetting (**5.09**), while Setting2 (87.5% retention) achieves the lowest forgetting (**1.77**) but limits new task learning ability (New.ACC drops to **63.43**). In contrast, the dynamic retention rate strategy adopted by PCLR (Setting3) achieves the best balance between the two, with an average accuracy of **62.19**, moderate forgetting (**3.39**), and high adaptability to new tasks (New.ACC = **65.16**).
>
> **Timing of retention rate adjustments**
>
> The dynamic adjustment strategy we adopt is based on the distillation loss during the integration phase: if, within the 50th–70th training steps of the integration process, at least 25% of the fake queries (i.e., 25% of the learned tasks) exhibit KL divergence losses exceeding a predefined threshold (set to 0.05 in this manuscript), it is determined that the current compression intensity is too high, causing significant knowledge conflicts between tasks. This occurs because the optimization objectives of different tasks generate severe discrepancies over the same rank components, leading to interference in task-specific information.
>
> In such cases, the system increases the retention rate and re-executes the compression-integration process, reducing the degree of information fusion between tasks to mitigate the loss of historical knowledge. Simultaneously, during the learning phase of subsequent tasks, the number of newly added ranks is correspondingly reduced (e.g., from 64 to 32) to maintain the total parameter capacity constant.
>
> **On the Adjustment Strategy**
>
> The number of added/removed ranks decreases in powers of 2 (64 → 32 → 16). Correspondingly, the retention rate is adjusted to 75% → 87.5% → 93.75%. This approach is inspired by the commonly used rank settings in efficient parameter tuning based on LoRA for multimodal tasks.

---

> ### Author Response · Authors · 2025-11-21
> **Response to Question 2**
>
> **Question 2: For the first task, are all experts trained, and is compression applied only after the second task begins?**
>
> **Response:**
>
> The first task only trained 64 newly added rank experts.
>
> **On the Timing of Compression**
>
> Taking the settings of the main experiment as an example: total rank experts = 256, activated rank experts = 64.
>
> For the first task, the system only initializes and optimizes 64 ranks (i.e., one activation block), while the remaining ranks remain unallocated. Subsequently, during the 2nd, 3rd, and 4th tasks, we sequentially expand the rank pool by adding 64 trainable ranks each time (cumulatively reaching 256). The first instance of compression and integration occurs after the fourth task is completed, at which point the LoRA Rank Pool (LRP) reaches the predefined upper limit.
>
> This design avoids resource wastage, as early tasks do not allocate computation and memory to parameters that may not be needed in the future.
>
> Overall, compression is only activated when the rank pool reaches its capacity limit, aligning with the principles of efficient parameter learning.

---

> ### Author Response · Authors · 2025-11-21
>
> Thank you once again for your hard work and valuable feedback, which have significantly improved the quality of our paper.

---

> ### Author Response · Authors · 2025-11-24
> **Supplementary Answer (Question 1)**
>
> Dear Reviewer U4Ez,
>
> Thank you for your concern regarding the compression rate settings. My previous explanation primarily focused on the rationale behind the selection of compression strategies and the comparison of different strategies. Here, I have supplemented the analysis with the impact of choosing different **initial compression retention rates** (which is related to knowledge density) on performance and memory for your reference. For more intuitive comparisons, we use **total rank experts + shared ranks** to measure the static storage parameters (SSP) and **activated rank experts + shared ranks** to measure the inference activated parameters (IAP).
>
> We propose setting the initial compression rate as **1 - activated rank experts / total rank experts** to enable efficient early-stage learning and avoid insufficient plasticity.
>
> ||SSP|IAP|ScienceQA|TextVQA|ImageNet|GQA|VizWiz|Grounding|VQAv2|OCRVQA|Avg.ACC|Forgetting|New.ACC|
> |:-:|:-:|:-:|:-:|:-:|:-:|:-:|:-:|:-:|:-:|:-:|:-:|:-:|:-:|
> |Setting1|192+32|64+32|72.01|53.68|88.32|58.19|55.20|30.82|64.38|63.40|60.75|4.83|64.98|
> |Setting2|320+32|64+32|80.71|59.36|90.81|56.82|56.05|31.32|65.13|62.50|62.84|2.65|65.16|
> |Ours|256+32|64+32|78.33|58.24|86.08|58.14|57.61|33.04|64.17|61.92|62.19|3.39|65.16|
>
> - **Setting1 (dense space):** With total rank experts = 192, the initial compression rate is adjusted to 66.67%, and the compression strategy is 66.67% → 88.33%, with new ranks changing from 64 → 32. 1/3 of the experts are activated during inference, resulting in dense knowledge compression and significant overlap among rank experts. Consequently, Avg.ACC decreases (60.75), Forgetting increases (4.83), and SSP reduces by 22.22% compared to ours.
> - **Setting2 (sparse space):** With total rank experts = 320, the initial compression rate is adjusted to 80%, and the compression strategy is 80% → 90%, with new ranks changing from 64 → 32. 1/5 of the experts are activated during inference, with weaker sharing among rank experts. Consequently, Avg.ACC increases (62.84), Forgetting decreases (2.65), and SSP rises by 22.22% compared to ours.
>
> The data shows that forgetting largely depends on the **knowledge density** of the PCLR parameter space (**activated rank experts / total rank experts**). Higher values tend to favor knowledge integration, while lower values favor expert specialization. **Knowledge density** is a flexible parameter designed to balance memory efficiency and continual learning capability. It can be adjusted based on memory constraints (there is no universally optimal setting; it depends on memory conditions).

---

> ### Author Response · Authors · 2025-11-24
>
> Thank you again for your efforts in helping us improve our work!

---

> ### Author Response · Authors · 2025-11-27
> **Further Discussion Request**
>
> Dear Reviewer U4Ez,
>
> We appreciate your time and apologize for the interruption. If you have any questions about our response or any other concerns, we are glad to have further discussion with you.
>
> Thank you for your time in reviewing our paper.

---

### Official Review · Reviewer_v6DS · 2025-10-31

**Soundness:** 3
**Presentation:** 3
**Contribution:** 3
**Rating:** 6
**Confidence:** 3

**Summary:**

This paper proposes a method named PCLR for CIT of LMMs, introducing the LRP and a CIL pipeline, and demonstrates superior performance over existing approaches on the CoIN and Continual-NExT benchmarks. Despite the appealing experimental results, the work suffers from significant weaknesses in theoretical grounding and systematic efficiency analysis, which undermine its scientific rigor and practical deployment value.

**Strengths:**

1. The writing is fluent, the figures are clear, and the content is easy to understand.
2. PCLR introduces a progressive strategy that dynamically adjusts compression and learning strategies according to the learning stage, thereby optimizing long-term learning performance.

**Weaknesses:**

1. The theoretical foundation heavily relies on intuitive analogies and lacks formal justification. The paper extensively borrows the neuroscience concept of “memory consolidation during human sleep” and maps it onto the three-stage CIL pipeline. However, this analogy remains purely heuristic, with no accompanying verifiable computational model or theoretical guarantees.
2. Moreover, the paper lacks critical comparisons of efficiency and parameter counts, making it impossible to assess the true validity of its claimed memory advantages. Particularly given its emphasis on “solving memory explosion” the absence of parameter statistics and GPU memory usage curves constitutes a fatal flaw.

**Questions:**

1. Could you provide the theoretical basis for decomposing and obtaining the LoRA rank pool? The current method lacks relevant theoretical support. Although LoRA contains "linearly dependent redundant rank vectors," linear dependence does not equate to redundant knowledge: even if two rank vectors are mathematically linearly dependent, they may carry different information at the semantic or task representation level.
2. Why is the rank level the optimal granularity? Why not coarser or finer? It is recommended to include related ablation experiments.
3. Since this paper's method is based on AdaLoRA, should AdaLoRA be included as a baseline in the experiments?
4. In Table 2, the parameter count for LLaVA-1.5-hf is not reported—does this refer to LLaVA-1.5-7b-hf?

---

> ### Author Response · Authors · 2025-11-21
>
> Dear Reviewer v6DS,
>
> We sincerely appreciate the effort you have devoted to reviewing our manuscript and the valuable comments you have provided. Below is our response:

---

> ### Author Response · Authors · 2025-11-21
> **Response to Weakness 1 (part1)**
>
> **Weakness 1: The CIL pipeline lacks a clear computational model or theoretical guarantees.**
>
> **Response-part1:**
>
> Our goal in continual Instruction tuning (CIT) is to avoid knowledge conflict while removing redundant representations when multi-task knowledge is learned in the same parameter space. PCLR removes redundant representations during the compression phase and merges similar knowledge during the integration phase.
>
> ### **Experimental Evidence of Compression**
>
> To demonstrate the existence of redundant parameters in LoRA efficient parameter tuning, we conducted experiments on the LLaVA-1.5-7B model, analyzing the immediate learning performance under different rank configurations. The specific results are as follows:
>
> | Ranks Num | ScienceQA | TextVQA | ImageNet | GQA | VizWiz | Grounding | VQAv2 | OCRVQA | Average |
> |:---------:|:---------:|:-------:|:--------:|:---:|:------:|:---------:|:-----:|:------:|:-------:|
> |    64     |   83.57   |  61.22  |   97.47  | 60.36 |  59.92 |   34.81   |  67.30 |  63.61  |  66.03  |
> |   128     |   84.01   |  61.86  |   97.33  | 60.86 |  60.94 |   36.48   |  67.32 |  64.17  |  66.62  |
>
> As shown in the table, when the rank number is halved from 128 to 64 (i.e., the training parameters are reduced by 50%), the average accuracy drops by only **0.59** (**66.62** → **66.03**), and the performance of multiple tasks (e.g., ImageNet, VQAv2) remains almost unchanged. This demonstrates that there are a large number of redundant rank dimensions in LoRA. Therefore, we can compress the parameter groups learned for each task to remove redundant representations.
>
> ### **Theoretical Guarantee of Integration**
>
> Assume that task $a$ learns a parameter set $A = \\{ x_i^a , i = 1, 2, \dots, n \\}$, and task $b$ learns $B = \\{ x_i^b , i = 1, 2, \dots, n \\}$.
>
> During the compression phase, we prune the parameter sets of both tasks, retaining only part of the parameters, resulting in $\tilde{A} = \\{ x_i^a , i = 1, 2, \dots, m \\}$ and $\tilde{B} = \\{ x_i^b , i = 1, 2, \dots, m \\}$, where $m < n$.
>
> To compensate for the information loss caused by compression, we introduce cross-task shared parameters during the integration phase: extracting a portion of the parameters from $\tilde{A}$ and $\tilde{B}$ for sharing between them, resulting in $\bar{A} = \\{ x_i^a \\}, \bar{B} = \\{ x_i^b \\}, \bar{C} = \\{ x_i^{ab} \\}$, ensuring the parameter quantities satisfy: $|\bar{A}+\bar{C}| = |\bar{B}+\bar{C}| = |A| = |B|$.
>
> Correspondingly, our optimization objective is the joint optimization of KL divergence:
> \\begin{cases}D_{KL}(O(x|A)||O(x|\bar{A}\cup\bar{C})) \\\\
> D_{KL}(O(x|B)||O(x|\bar{B}\cup\bar{C}))\end{cases}\, where $O(x|A)$ represents the logits output generated by parameter set $A$ for the input sequence $x$. $D_{KL}$ is the commonly used KL divergence in distillation. Through joint optimization, the shared parameter set $\bar{C}$ can effectively capture the common knowledge between tasks $a$ and $b$, thereby mitigating the performance loss caused by compressed parameters.

---

> ### Author Response · Authors · 2025-11-21
> **Response to Weakness 1 (part2)**
>
> **Weakness 1: The CIL pipeline lacks a clear computational model or theoretical guarantees.**
>
> **Response-part2:**
>
> ### **Biological Computational Model**
>
> Our approach is inspired by the mechanism of "memory consolidation during sleep" in neuroscience. This process exhibits a highly structured circadian rhythm in the brains of humans and mammals: **experience acquisition during the day, and memory consolidation at night.** We systematically map this natural intelligence strategy onto the computational framework of Continual Instruction Tuning (CIT), constructing a bio-inspired **three-stage process of learning, compression, and integration.**
> ### **Daytime: Experience Acquisition → Learning Phase**
>
> ### **(a) Learning New Knowledge**
>
> - **Human Behavior**: The CA1/CA3 regions of the hippocampus record event sequences with high plasticity during wakefulness, forming transient but high-capacity "memory traces". However, these representations are unstable and prone to overwriting. [1]
> - **Our Approach**: LoRA Rank Pool (LRP) serves as a short-term memory buffer, dynamically expanding to accommodate knowledge from new tasks. Its parameters are updated only for the current task, avoiding interference with historical pathways.
>
> ### **Nighttime: Memory Consolidation → Compression and Integration Phase**
>
> ### **(b) Resource Reset**
>
> - **Human Behavior**: Related research indicates that following sharp-wave ripples (SWR), the hippocampal CA2 region triggers **BARR (Burst of Action Potential Reset)**, a strong inhibitory discharge that selectively suppresses CA1/CA3 neurons, achieving "neural reset" and releasing resources for subsequent learning. Blocking BARR leads to excessive neuronal synchronization and impaired memory performance. [2]
> - **Our Approach**: After learning, we perform **structured compression** on LRP, removing redundant components to free up space for future learning. This mimics the biological system's **synaptic pruning** and **resource recycling**, preventing unlimited parameter growth.
>
> ### **(c) Replay and Transfer**
>
> - **Human Behavior**: During non-rapid eye movement (NREM) slow-wave sleep, the hippocampus replays daytime experiences offline and gradually transfers information to the neocortex, forming long-term memory. Related institutions have confirmed through intracranial electroencephalography (iEEG) studies that this process involves two distinct "echo activations": the first driven by the hippocampus, and the second dominated by cortical spindle waves, ultimately completing semantic consolidation. [3]
> - **Our Approach**: We introduce **pseudo queries**, samples drawn from the input distribution of the current task, to **activate the rank subspaces related to historical tasks** without accessing historical data. Using knowledge distillation loss (KL divergence), we constrain the student model's output to be consistent with the teacher model's (pre-compression state) output, achieving **implicit replay and cross-task knowledge transfer.** This mechanism avoids storing raw data, ensuring privacy compliance while accurately simulating the brain's ability to replay autonomously without external input.
>
> To the best of our knowledge, PCLR is the first framework to systematically incorporate the mechanism of **"memory consolidation during sleep"** into continual instruction tuning. It not only functionally emulates the brain's memory management strategies but also significantly outperforms existing continual learning methods in terms of performance. This bio-inspired design enables PCLR to achieve long-term stability, efficiency, and privacy-friendly continual learning under a **fixed parameter budget** (analogous to the brain's limited neural resources).
>
> ### **References**
>
> [1] Buzsáki, G. Hippocampal sharp wave-ripple: A cognitive biomarker for episodic memory? Neuron, 88(4):684–697, 2015.
>
> [2] Oliva, A. et al. A hippocampal circuit mechanism for balancing memory reactivation during sleep. Science, 385(6708):123–130, 2024.
>
> [3] Wang, L. et al. Electrophysiological signatures underlying variability in human memory consolidation. Nature Communications, 16:2105, 2025.

---

> ### Author Response · Authors · 2025-11-21
> **Response to Weakness 2**
>
> **Weakness 2: Lack of a critical comparison of efficiency and parameter count.**
>
> **Response:**
>
> We agree on the necessity of conducting a quantitative analysis of memory efficiency and parameter scalability. To this end, we have supplemented detailed experimental data across multiple dimensions to systematically evaluate the performance of PCLR in terms of computational cost, parameter utilization, and memory usage.
>
> ### **Comparison of Learning Efficiency**
>
> Based on the LLaVA-1.5-13b model, we compared the training time (in minutes) during the Learning phase between PCLR and MoE-LoRA with varying numbers of experts on 4 × 80GB GPUs ("Experts Num = 1" corresponds to standard LoRA).
>
> |Experts Num|ScienceQA|TextVQA|ImageNet|GQA|VizWiz|Grounding|VQAv2|OCRVQA|Average|
> |:-:|:-:|:-:|:-:|:-:|:-:|:-:|:-:|:-:|:-:|
> |1|10.05|33.55|91.22|118.12|17.78|79.58|93.44|135.68|72.43|
> |2|17.73|50.97|130.61|173.58|27.94|115.71|136.53|188.86|105.24|
> |4|21.25|68.14|189.05|245.97|35.25|169.12|205.52|286.92|152.65|
> |ours|8.86|34.43|94.82|115.49|19.85|93.68|121.87|177.12|83.27|
>
> The results indicate that PCLR achieves training speeds comparable to standard LoRA (Experts Num = 1) and significantly outperforms MoE-LoRA (Experts Num ≥ 2). Despite introducing the LoRA Rank Pool mechanism, PCLR does not incur noticeable additional overhead during forward and backward propagation, validating its computational efficiency.
>
> ### **Comparison of Parameter Utilization and Memory Usage**
>
> Using the LLaVA-1.5-7b model on the CoIN benchmark, we further compared PGP, Eproj, and different configurations of PCLR in terms of total parameters, activated parameters (involved in forward/backward propagation), trainable parameters, and final performance (all statistics exclude the base model and focus solely on the adapter components).
>
> ### **PGP (maintains a fixed total parameter count but lacks flexibility in resource allocation):**
> || ScienceQA|TextVQA|ImageNet|GQA|VizWiz|Grounding|VQAv2|OCRVQA|
> |:-:|:-:|:-:|:-:|:-:|:-:|:-:|:-:|:-:|
> |Total Params|340.80M|340.80M|340.80M|340.80M|340.80M|340.80M|340.80M|340.80M|
> |Activated Params|340.80M|340.80M|340.80M|340.80M|340.80M|340.80M|340.80M|340.80M|
> |Trainable Params|340.80M|340.80M|340.80M|340.80M|340.80M|340.80M|340.80M|340.80M|
> |Final Accuracy|85.17|56.85|32.26|61.74|49.43|32.74|65.74|62.20|
>
> **Avg.ACC is 55.77, Forgetting is 12.94, New.ACC is 67.09.**
>
> ### **Eproj (expands based on task similarity but risks continuous growth with longer task sequences):**
> ||ScienceQA|TextVQA|ImageNet|GQA|VizWiz|Grounding|VQAv2|OCRVQA|
> |:-:|:-:|:-:|:-:|:-:|:-:|:-:|:-:|:-:|
> |Total Params|340.80M|660.61M|980.43M|980.43M|980.43M|1300.24M|1300.24M|1300.24M|
> |Activated Params|340.80M|340.80M|340.80M|340.80M|340.80M|340.80M|340.80M|340.80M|
> |Trainable Params|340.80M|340.80M|340.80M|340.80M|340.80M|340.80M|340.80M|340.80M|
> |Final Accuracy|78.51|57.53|92.35|55.93|44.67|36.59|63.74|57.00|
>
> **Avg.ACC is 60.79, Forgetting is 5.42, New.ACC is 65.54.**
>
> ### **PCLR (0 shared ranks, 32 active rank experts, 128 total rank experts):**
> ||ScienceQA|TextVQA|ImageNet|GQA|VizWiz|Grounding|VQAv2|OCRVQA|
> |:-:|:-:|:-:|:-:|:-:|:-:|:-:|:-:|:-:|
> |Total Params|85.79M|171.58M|257.37M|343.16M|343.16M|343.16M|343.16M|343.16M|
> |Activated Params|85.79M|85.79M|85.79M|85.79M|85.79M|85.79M|85.79M|85.79M|
> |Trainable Params|85.79M|85.79M|85.79M|85.79M|85.79M|85.79M|42.90M|42.90M|
> |Final Accuracy|78.87|58.23|82.16|56.77|53.81|30.76|63.86|60.14|
>
> **Avg.ACC is 60.58, Forgetting is 4.13, New.ACC is 64.19.**
>
> ### **PCLR (32 shared ranks, 64 active rank experts, 256 total rank experts):**
> ||ScienceQA|TextVQA|ImageNet|GQA|VizWiz|Grounding|VQAv2|OCRVQA|
> |:-:|:-:|:-:|:-:|:-:|:-:|:-:|:-:|:-:|
> |Total Params|171.58M|343.16M|514.74M|686.31M|766.70M|766.70M|766.70M|766.70M|
> |Activated Params|171.58M|171.58M|171.58M|171.58M|251.97M|251.97M|251.97M|251.97M|
> |Trainable Params|171.58M|171.58M|171.58M|171.58M|171.58M|171.58M|85.79M|85.79M|
> |Final Accuracy|78.33|58.24|86.08|58.14|57.61|33.04|64.17|61.92|
>
> **Avg.ACC is 62.19, Forgetting is 3.39, New.ACC is 65.16.**
>
> PCLR imposes a strict upper bound on total ranks, halting parameter growth after early expansion via compression-integration. In contrast, Eproj expands continuously, risking parameter explosion. PCLR outperforms both PGP and Eproj in average accuracy (**62.19** v.s. **55.77** / **60.79**) and forgetting (**3.39** v.s. **12.94** / **5.42**), demonstrating its ability to control memory while maintaining strong continual learning performance.
>
> When the total parameter count is constrained to a level similar to PGP (0 shared ranks, 32 active rank experts, 128 total rank experts), PCLR still achieves competitive performance with an Avg.ACC of **60.58**, close to Eproj (**60.79**) and significantly higher than PGP (**55.77**). These results demonstrate that PCLR can effectively utilize limited resources, providing robust performance even under strict parameter budget constraints.

---

> ### Author Response · Authors · 2025-11-21
> **Response to Question 1**
>
> **Question 1: Theoretical Basis for Decomposing the LoRA Rank Pool.**
>
> **Response:**
>
> ### **Theoretical Basis for Decomposing the LoRA Rank Pool**
>
> Our approach does not simply treat linearly dependent rank vectors as "invalid". Instead, it leverages the intrinsic degrees of freedom in the low-rank structure of matrices. By optimizing during the Compression-Integration phase, it identifies and compresses rank components that are functionally redundant under the current task sequence, without compromising representational capacity.
>
> ### **Equivalence Between Linear Dependency and Redundant Representations**
>
> Standard LoRA models weight updates as $\Delta W = AB$, where $A \in \mathbb{R}^{n \times r}$ and $B \in \mathbb{R}^{r \times m}$, satisfying $m, n \gg r$. Assuming $A$ is column full-rank and $B$ is row full-rank, we have $rank(\Delta W) = r$.
>
> Now, consider an over-parameterized extension: adding a column vector linearly dependent on the column space of $A$, resulting in $\tilde{A} \in \mathbb{R}^{n \times (r+1)}$ with $rank(\tilde{A}) = r$. Similarly, adding a row vector linearly dependent on the row space of $B$ gives $\tilde{B} \in \mathbb{R}^{(r+1) \times m}$ with $rank(\tilde{B}) = r$. In this case:
>
> \\[
> rank(\Delta \tilde{W}) \leq \min(rank(\tilde{A}), rank(\tilde{B})) = r.
> \\]
>
> According to the fundamental theorem of matrix rank decomposition, any matrix with a rank no greater than $r$ can be exactly reconstructed by a pair of factors with dimensions $n \times r$ and $r \times m$, respectively:
>
> \\[
> \Delta \tilde{W} = CD, \quad C \in \mathbb{R}^{n \times r}, \quad D \in \mathbb{R}^{r \times m}.
> \\]
>
> This indicates that the over-parameterized LoRA pair $(\tilde{A}, \tilde{B})$ with hidden dimension $r+1$ can always be optimized into a more compact rank-$r$ LoRA representation $(C, D)$.
>
> Thus, while $\tilde{A}$ and $\tilde{B}$ have higher dimensionality in parameter space, the function space they span remains unchanged. The newly added rank directions introduce no additional representational power but merely constitute redundant degrees of freedom in parameterization.
>
> ### **Generalization**
>
> More generally, let $A \in \mathbb{R}^{n \times x}$ and $B \in \mathbb{R}^{x \times m}$, satisfying $m, n \gg x$, $rank(A) = r_A$, and $rank(B) = r_B$. Let $r = \min(r_A, r_B)$, then $rank(AB) \leq r$. Thus, we can express:
>
> \\[
> AB = CD, \quad C \in \mathbb{R}^{n \times r}, \quad D \in \mathbb{R}^{r \times m}.
> \\]
>
> This means $(C, D)$ is a simplified form of $(A, B)$. Computationally, they are equivalent:
>
> \\[
> xW + xAB = xW + xCD, \quad x \in \mathbb{R}^{b \times n}, \quad W \in \mathbb{R}^{n \times m}.
> \\]
>
> This demonstrates that $(C, D)$ and $(A, B)$ are semantically equivalent.
>
> ### **Conclusion**
>
> In summary, our core insight is that the presence of linearly dependent rank components implies compressible degrees of freedom in the parameter space. However, in multi-objective optimization scenarios, the process of compression and adjustment should be guided by optimization goals rather than solely relying on algebraic relationships. In PCLR, the compression and integration phase uses knowledge distillation to project the high-dimensional rank pool onto the minimal effective subspace while maintaining joint performance across tasks. This process does not simply "remove a linearly dependent vector" but instead finds an equivalent yet more compact LoRA representation, enhancing memory efficiency without sacrificing representational capacity. In other words, we exploit the structural redundancy revealed by linear dependencies and achieve functionally equivalent compact representations through a learning-driven approach.
>
> Therefore, to identify and compress redundant rank components, decomposing LoRA to obtain the Rank Pool is absolutely essential.

---

> ### Author Response · Authors · 2025-11-21
> **Response to Question 2**
>
> **Question 2: The Fine-Grained Structure of Experts in Ablation Experiments.**
>
> **Response:**
>
> ### **Fine-Grained Ablation Experiments of Experts**
>
> We systematically conducted fine-grained ablation experiments on the expert granularity using the LLaVA-1.5-7b model and CoIN benchmark, comparing performance when experts were defined as 1-rank, 2-rank, and 4-rank:
>
> | Expert Granularity (ranks/expert) | ScienceQA | TextVQA | ImageNet | GQA  | VizWiz | Grounding | VQAv2 | OCRVQA | Avg.ACC | Forgetting | New.ACC |
> | :-------------------------------: | :-------: | :-----: | :------: | :--: | :----: | :-------: | :---: | :----: | :-----: | :--------: | :-----: |
> |    1    | 78.33 | 58.24 | 86.08 | 58.14  |  57.61  |   33.04    |  64.17  | 61.92  |  62.19  |  3.39  |  65.16  |
> |    2    | 78.33 | 58.30 | 86.57 | 58.88  |  57.24  |   30.96    |  64.36  | 62.07  |  62.09  |  3.29  |  64.97  |
> |    4    | 78.47 | 58.10 | 86.85 | 57.82  |  56.33  |   31.96    |  64.48  | 60.82  |  61.85  |  3.50  |  64.91  |
>
> From the results in the table, it can be observed that the expert granularity (1/2/4 ranks per expert) has minimal impact on final performance. This does not imply that granularity is unimportant for PCLR. As noted in our manuscript, extreme granularity provides maximum flexibility for knowledge editing. While 2-rank and 4-rank expert configurations remain applicable to CoIN tasks, their potential to handle long task sequences diminishes.
>
> ### **Reasons for Selecting 1-rank as the Base Granularity**
>
> ### **(1) Maximizing Resource Scheduling Flexibility**
>
> Our system sets an upper limit of 256 for the total rank pool. If the expert granularity is 4-rank, the system can accommodate at most 64 experts; for 2-rank, the limit is 128 experts; while 1-rank granularity supports 256 independent schedulable units. During long task sequences, the retention rate during the compression phase progressively increases (e.g., from 75% → 87.5% → 93.75%). In this case, a 4-rank configuration can retain at most 63/64 = 98.4% of experts, which prevents finer retention ratios (e.g., 99%). In contrast, a 1-rank configuration can flexibly support any retention ratio (e.g., 255/256 ≈ 99.6%), thereby minimizing knowledge loss.
>
> ### **(2) Avoiding the "All-or-Nothing" Risk for Task Knowledge**
>
> Consider a scenario where the algorithm fails due to the use of 32-rank experts. PCLR only one new expert (32 rank) is added during the 7th task. If this expert is entirely removed during subsequent compression, all newly learned knowledge for that task would be instantly lost, leading to catastrophic forgetting. In contrast, 1-rank granularity allows progressive retention or discarding of knowledge for the same task, significantly enhancing robustness.
>
> ### **(3) Reasons for Not Choosing Finer Granularity**
>
> We do not choose granularity finer than 1-rank because LoRA's semantic representation capability is fundamentally carried by its rank structure: each rank-1 component (i.e., the outer product of one column of matrix $A$ and one row of matrix $B$) constitutes the smallest, complete, and independently operable semantic unit for low-rank updates. In PCLR, whether adding ranks during the learning phase or removing ranks during the compression phase, the model consistently maintains valid LoRA weight structures, ensuring correctness of forward/backward computation and flexibility in knowledge editing. Editing at finer levels (e.g., incomplete vectors or embedding dimensions) would disrupt the matrix multiplication structure of LoRA, rendering weights invalid and potentially causing adapter failure. Therefore, selecting rank as the fundamental unit is theoretically sound and practically feasible for efficient parameter tuning based on LoRA.

---

> ### Author Response · Authors · 2025-11-21
> **Response to Question 3**
>
> **Question 3: Comparison with the AdaLoRA Baseline.**
>
> **Response:**
>
> First, we want to clarify that our method is not based on the architecture or algorithmic process of AdaLoRA but is inspired by its core idea of "dynamically adjusting LoRA parameters at the rank level to improve memory efficiency." AdaLoRA itself is designed for efficient optimization of parameters during single-task fine-tuning and does not model the continual learning scenario, especially lacking mechanisms for addressing catastrophic forgetting, knowledge integration, or long-term parameter management.
>
> Nevertheless, to comprehensively evaluate the performance of existing low-rank adaptive methods in Continual Instruction Tuning (CIT), we include AdaLoRA in our comparison and test it on the LLaVA-1.5-7b model and CoIN benchmark:
>
> | Metrics            | ScienceQA | TextVQA | ImageNet | GQA    | VizWiz | Grounding | VQAv2  | OCRVQA | Avg   |
> | :----------------: | :-------: | :-----: | :------: | :----: | :----: | :-------: | :----: | :----: | :---: |
> | Instant Accuracy   | 75.08     | 59.83   | 97.41    | 57.92  | 59.34  | 27.99     | 66.57  | 62.03  | 63.27 |
> | Final Accuracy     | 73.40     | 51.29   | 35.47    | 44.53  | 46.75  | 0.93      | 55.86  | 62.03  | 46.28 |
> | Supervised Fine-Tuning | 75.08 | 58.63   | 97.21    | 58.74  | 59.87  | 16.98     | 64.87  | 49.85  | 60.15 |
>
> **Forgetting is 23.99.** Supervised Fine-Tuning refers to instruction tuning for each task individually without a continual learning setup.
>
> The results in the table indicate that AdaLoRA exhibits significantly higher forgetting compared to methods specifically designed for continual learning, such as PCLR (**3.39**) and Eproj (**5.42**). Therefore, while AdaLoRA provides valuable inspiration, it is not suitable as a strong baseline for continual learning. We will use AdaLoRA as an early baseline in the main comparative experiment.

---

> ### Author Response · Authors · 2025-11-21
> **Response to Question 4**
>
> **Question 4: The parameter count for LLaVA-1.5-hf.**
>
> **Response:**
>
> Yes, the “LLaVA-1.5-hf” mentioned in the text refers to “LLaVA-1.5-7b-hf.” It is important to note that the model weights of LLaVA-1.5-7b and LLaVA-1.5-7b-hf are not the same. LLaVA-1.5-7b-hf is an optimized version based on the Hugging Face community, featuring more thorough multimodal alignment and instruction fine-tuning. This results in stronger zero-shot and generalization capabilities.

---

> ### Author Response · Authors · 2025-11-21
>
> Thank you once again for your hard work and valuable feedback, which have significantly improved the quality of our paper.

---

> ### Comment · Reviewer_v6DS · 2025-11-25
>
> Thank you for the author's clarification, but I still believe the theoretical basis relies too heavily on intuition. Therefore, I maintain my original score.

---

> ### Author Response · Authors · 2025-11-27
>
> Thank you for your reply. We would like to clarify that the CIL pipeline is inspired by “memory consolidation during human sleep” and is supported by well-established biological theories. Our paper is the first study to apply this framework to multimodal continual instruction tuning. Each stage of CIL is grounded in clear rationale.
>
> In the compression phase, we draw inspiration from sharp-wave ripples (SWRs), which facilitate a "neural reset" in the brain. We perform LoRA rank redundancy analysis and confirm that there indeed exists redundant representational capacity that can be released. We further validate the feasibility of this approach through empirical experiments.
>
> The integration phase draws inspiration from non-rapid eye movement (NREM) slow-wave sleep, and we implement it using knowledge distillation based on KL divergence. Analysis of the objective function confirms that it enables effective knowledge transfer and integration, and ablation studies further validate its feasibility.
>
> The learning phase follows standard instruction tuning.
>
> Thank you again for your suggestions and patient response.

---

### Author Response · Authors · 2025-12-01
**Rebuttal Summary**

We summarize the key discussion points raised by the three reviewers as follows:

**Rebuttal Abstract:** We have addressed the concerns raised by Reviewer pBRi, who has indicated they are willing to increase their score from 4 to 6. Reviewer v6DS has stated that the current score will be maintained, and we are still awaiting a response from Reviewer U4Ez.

**1. Comparative analysis of memory efficiency and parameter counts (from Reviewers v6DS and pBRi):** We present a comparison of the number of parameters and continuous learning performance under settings such as PGP(regularization-based and LoRA-based strong baseline), Eproj(extension-based and LoRA-based strong baseline), PCLR-small, and PCLR-ours, intuitively demonstrating that our algorithm has a memory advantage while possessing more powerful performance. The related content has been added to Appendix Q of the manuscript. In addition, we conducted more comprehensive ablation studies from two perspectives—parameter scale and activation ratio—to more completely demonstrate how PCLR balances memory efficiency and learning performance, and we have included the results in Appendix P. Conclusions: Sparse activation exhibits stronger anti-forgetting but poorer memory performance; dense activation is more memory-efficient but increases forgetting. Increasing the total parameter capacity benefits plasticity and stability, and shared spaces enhance stability.

**2. Rationale for compression strategies and the implementation of progression (from Reviewers U4Ez and pBRi):** Theoretically, we analyze the knowledge density increases during continual learning evolution and the optimization objective becomes more complex. Experimentally, we controlled the parameter space to be identical and set five compression strategies (aggressive, conservative, reversed, concentrated, and ours) to validate the effectiveness of PCLR’s progressive learning. We have added these experiments to the main text of the manuscript. Conclusions: A progressive evolution from low retention–high addition to high retention–low addition is essential for both plasticity and stability. We detail how the progression is implemented and include it in Appendix J. Furthermore, we present the impact of different initial compression rates on the algorithm’s performance as a supplement, addressing Reviewer U4Ez’s question.

**3. Ablation on expert granularity (from Reviewer v6DS):** Reviewer v6DS noted the lack of justification for choosing r=1 as the basic expert granularity. In the main text, we state that this choice enables maximal freedom for knowledge editing and scheduling. We further substantiate this experimentally with settings r=1, 2, and 4, confirming the performance and learning potential of r=1.

**4. Additional baseline comparisons (from Reviewers v6DS and pBRi):** Reviewer v6DS requested a comparison with AdaLoRA, and Reviewer pBRi requested comparisons with SEFE, HiDe-LLaVA, and ProgLoRA. We have implemented all of these and added the results to the main table of the manuscript.

**5. Baseline comparison with increased epochs (from Reviewer pBRi):** We compare PGP and our method at epochs = 1, 3, and 5, demonstrating that our approach maintains stronger plasticity and stability with more epochs, and that forgetting decreases as the number of epochs increases. These experiments have been added to Appendix O.

**6. Concerns about additional overhead during integration (from Reviewer U4Ez):** We compared algorithmic performance and overhead under different Integrated Data Usage (IDU) configurations and emphasized that IDU is a fast-converging, on-demand setting. With minimal additional cost, substantial improvements can be achieved, and higher expenditure generally yields further gains. For specific details, please refer to Section 5.6 of the manuscript and our responses to the reviewers.

**7. Concerns about PCLR’s long-term performance (from Reviewer U4Ez):** We introduce a setup that dynamically increases capacity during training, further improving resistance to forgetting. We also provide a more detailed description of the proposed improved PCLR-LwF, which executes compression and integration on-demand rather than maintaining CIL at all times, further validating the algorithm’s performance and scalability.

**8. Theoretical concerns (from Reviewer v6DS):** We present the theoretical foundations for factorizing and obtaining the LoRA rank pool, the theoretical basis and computational model for CIL, as well as corresponding experimental analyses.

**9. Requests for details (from Reviewers v6DS, U4Ez, and pBRi):** We have addressed the reviewers’ questions regarding the detailed aspects of the paper.

---

### Meta-Review · Area_Chair_imqq · 2026-01-07

**Summary:**

This paper proposes a novel progressive learning algorithm that dynamically adjusts compression and learning strategies within a Compression–Integration–Learning training pipeline. The reviewers’ main concerns include the following. First, the theoretical foundation of the method relies heavily on intuitive analogies and lacks formal justification, making the theoretical analysis insufficiently solid. Second, reviewers questioned why the progressive learning process adjusts the number of newly introduced ranks and the compression retention rate in specific ways, noting that these hyperparameter choices lack clear theoretical or experimental justification. In addition, concerns were raised about learning efficiency, as the Integration stage requires additional training, potentially leading to longer overall training time. Reviewers also noted the lack of critical comparisons of efficiency and parameter counts, which makes it difficult to assess the validity of the claimed memory advantages. Finally, the baseline comparisons were considered insufficiently up to date, particularly with respect to recent multimodal continual instruction tuning methods published in 2024 and 2025.

The authors attempted to provide a theoretical explanation for the proposed progressive learning process based on the Compression–Integration–Learning pipeline, but these explanations still fall short of providing fundamental theoretical guarantees. However, the additional experiments are fairly convincing: the authors validated the performance of PCLR under a variety of training settings, which led at least one reviewer to increase their score. Overall, I will tend to a weak accept. While the theoretical aspect is somewhat limited, the proposed PCLR algorithm demonstrates a reasonable degree of novelty in the context of multimodal continual instruction tuning, and the experimental results sufficiently support its effectiveness.

**Reviewer Concerns:**

I think the rebuttal addressed several of the reviewers’ technical concerns through additional experiments. In particular, the authors provided a comprehensive comparative analysis of memory efficiency, ablation studies on parameter scale and activation ratios, and comparisons with more recent multimodal continual instruction-tuning methods. They also reported the associated computational cost, demonstrating the claimed memory and efficiency advantages of PCLR. These experimental validations effectively respond to concerns regarding empirical performance and the practical effectiveness of the proposed method.

However, some concerns remain outstanding. Specifically, the theoretical guarantees of PCLR are still not fully established, and the rationale for selecting particular rank levels and compression retention rates  lacks a clear theoretical or experimental justification

**Reviewer Scores:**

Reviewer v6DS still challenges the theoretical guarantees of the PCLR method, and therefore tends to maintain a score of 6.

Reviewer U4Ez did not change their stance and continues to maintain a score of 6 despite the authors’ responses.

Reviewer pBRi believes that the additional experiments address part of their concerns, and has raised the score from 4 to 6.

---

### Decision · Program_Chairs · 2026-01-26

Accept (Poster)